# Structure and function of rice hybrid genomes reveal genetic basis and optimal performance of heterosis

Zhoulin Gu [1,6], Junyi Gong [2,6], Zhou Zhu[1,3,6], Zhen Li[1,4,6], Qi Feng [1], Changsheng Wang[1], Yan Zhao[1], Qilin Zhan[1], Congcong Zhou[1], Ahong Wang[1], Tao Huang [1], Lei Zhang[1], Qilin Tian[1], Danlin Fan[1], Yiqi Lu[1], Qiang Zhao[1], Xuehui Huang [5], Shihua Yang [2] ✉ & Bin Han [1] ✉

Exploitation of crop heterosis is crucial for increasing global agriculture production. However, the quantitative genomic analysis of heterosis was lacking, and there is currently no effective prediction tool to optimize cross-combinations. Here 2,839 rice hybrid cultivars and 9,839 segregation individuals were resequenced and phenotyped. Our findings demonstrated that *indica–indica* hybrid-improving breeding was a process that broadened genetic resources, pyramided breeding-favorable alleles through combinatorial selection and collaboratively improved both parents by eliminating the inferior alleles at negative dominant loci. Furthermore, we revealed that widespread genetic complementarity contributed to *indica–japonica* intersubspecific heterosis in yield traits, with dominance effect loci making a greater contribution to phenotypic variance than overdominance effect loci. On the basis of the comprehensive dataset, a genomic model applicable to diverse rice varieties was developed and optimized to predict the performance of hybrid combinations. Our data offer a valuable resource for advancing the understanding and facilitating the utilization of heterosis in rice.

Heterosis or hybrid vigor refers to the phenomenon that the heterozygous first filial generation (F$_1$) performs better than its parental inbred lines in target traits. With the development of the first commercial hybrid maize variety in the 1930s (ref. 1), and the development of rice hybrid varieties in the early 1970s in China[2], exploitation of heterosis in crop plants has achieved remarkable yield advantages over inbred lines, and remains a crucial approach to increase agricultural production for global food demand in response to rapidly increasing global population and changing climate[3,4].

Three non-mutually exclusive hypotheses—dominance, overdominance and epistasis—have been proposed to explain the genetic basis of crop heterosis[5–8], and several works have confirmed these hypotheses. Loci demonstrating overdominance or pseudo-overdominance effect, have been shown to drive single-locus heterosis in tomato and sorghum[9,10]. The genetic complementation (dominance) is observed to contribute to heterosis in *Arabidopsis*, maize and rice[11–13]. Large-scale genomic analysis based on commercial rice hybrids and associated populations, provides the support for partial dominance of heterozygous

[1]National Center for Gene Research, State Key Laboratory of Plant Molecular Genetics, CAS Center for Excellence in Molecular Plant Sciences, Institute of Plant Physiology and Ecology, Chinese Academy of Sciences, Shanghai, China. [2]State Key Laboratory of Rice Biology, China National Rice Research Institute, Chinese Academy of Agricultural Sciences, Hangzhou, China. [3]University of Chinese Academy of Sciences, Beijing, China. [4]College of Life Sciences, Anhui Normal University, Wuhu, China. [5]College of Life Sciences, Shanghai Normal University, Shanghai, China. [6]These authors contributed equally: Zhoulin Gu, Junyi Gong, Zhou Zhu, Zhen Li. ✉e-mail: yangshihua6308@163.com; bhan@ncgr.ac.cn

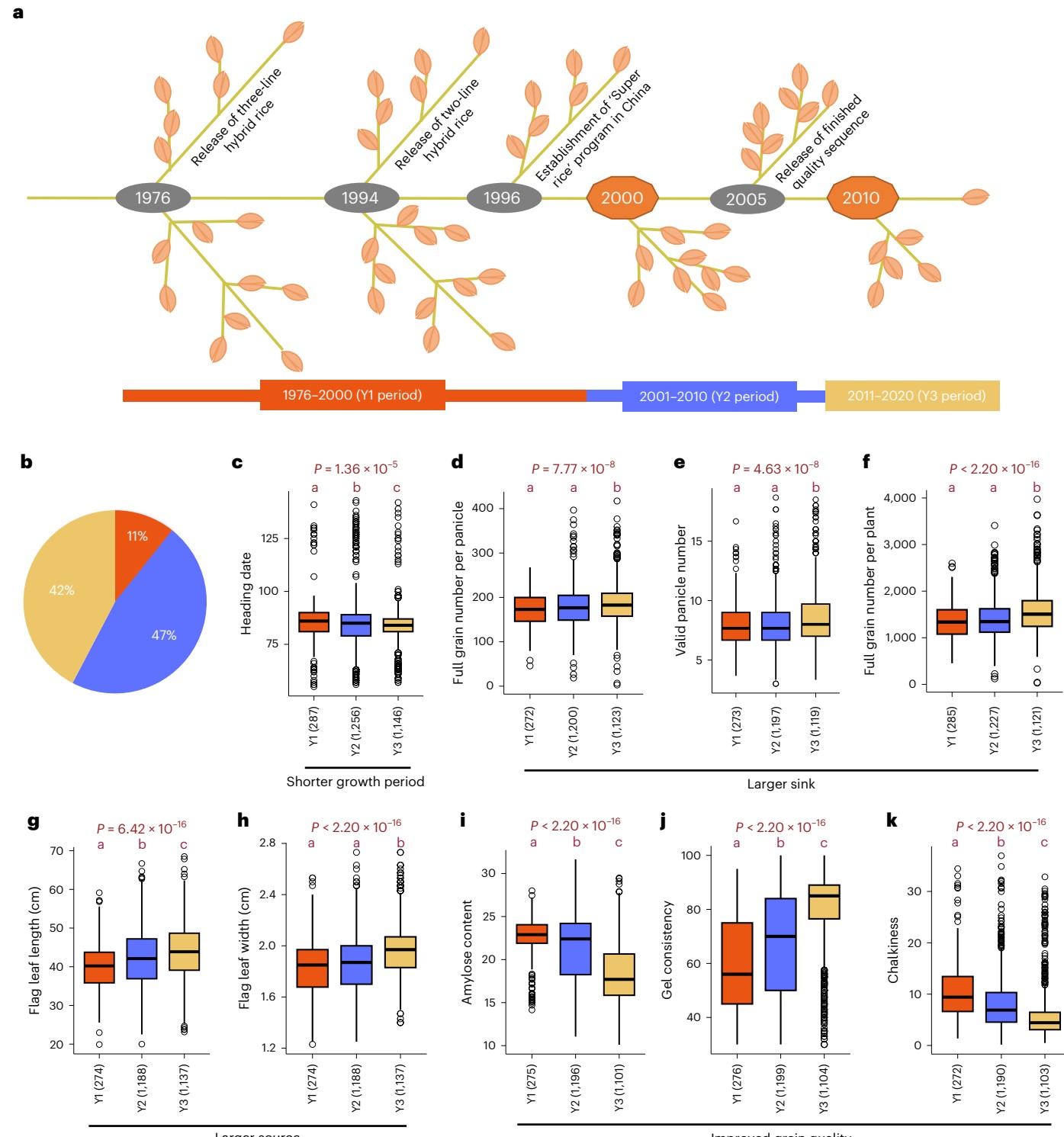

**Fig. 1 | The phenotypic change across improvement breeding for rice hybrids.** **a**, Three periods were divided on the basis of key developments of hybrid rice breeding. **b**, The proportion of hybrids from the three breeding periods. **c**–**k**, Box plots demonstrating phenotype distribution for hybrids from the three breeding periods: heading date (**c**); full grain number per panicle (**d**); valid panicle number (**e**); full grain number per plant (**f**); flag leaf length (**g**); flag leaf width (**h**); amylose content (**i**); gel consistency (**j**); chalkiness (**k**). Sample size is shown in parentheses. Significance test was conducted by one-way ANOVA for data with homoscedasticity distribution or Kruskal–Wallis test for data with heteroscedasticity distribution. Multiple comparison was further conducted by the least significant difference (LSD) method with 'Bonferroni' correction for homoscedasticity distribution or Nemenyi test for heteroscedasticity distribution. Different lowercase letters above the box plots represent significant phenotype differences ($P \leq 0.05$).

loci for yield-related traits, and the phenomenon of heterosis in rice agricultural production emerges from the nonlinear effects of multiple heterozygous genomic loci combined[14,15]. The observation provides

evidence that there is a dosage-sensitive component to heterosis[16]. Genomic studies have identified a number of quantitative trait loci (QTLs) that contribute to rice heterosis[14,17,18]. However, comprehensive

analysis that scans genome-wide breeding footprints in rice hybrids to identify and evaluate heterotic loci in a dynamic perspective is lacking.

In this Article, we performed a comprehensive genome analysis of 2,839 rice hybrid cultivars and 9,839 $F_2$ individuals derived from 18 elite hybrids, which represents the majority of commercially used rice hybrids in China over past 50 years. We systematically investigated the genetic basis of phenotypic changes for grain yield-related traits, flowering time (heading date) and grain quality, and explored how breeding history applied heterotic loci and shaped the genomic architecture in rice hybrids. We further investigated the genetic basis underlying strong heterosis in intersubspecific rice hybrids. Moreover, a genomic selection model was developed on the basis of the comprehensive dataset, to facilitate selection of hybrid combinations with high yield potential and design of crossing plans. The model has been validated with respect to its accuracy and broad applicability, including testing in intersubspecific rice hybrids.

## Results

### Phenotyping and resequencing

The experimental design of this study is shown in Extended Data Fig. 1a. To capture the genetic signatures for improvement breeding in rice hybrids, we collected 2,839 rice hybrid cultivars released from years 1976 to 2020 (Supplementary Table 1). All 2,839 rice hybrids were resequenced at an average depth of 35-fold (Supplementary Table 2). Clean paired-end reads were aligned against the *Oryza sativa* cv. Nipponbare IRGSP 1.0 reference genome[19] to identify polymorphisms. When compared with 100-fold sequencing data, 35-fold sequencing data could capture 96.70% of high-quality single-nucleotide polymorphisms (SNPs), with accuracy estimated at 95.21% and 94.72% for all and heterozygous loci (Supplementary Table 3). Thus, the polymorphic information provided in our study was of high accuracy and density. In total, 5,222,902 high-quality SNPs and 1,701,091 InDels (insertions and deletions with size ≤50 bp) were identified for the whole set of collections. Additionally, after alignment of clean paired-end reads against the graph-based rice genome[20], we also identified 22,555 high-quality SVs (structural variation with size ≥51 bp) for 964 representative hybrids.

Furthermore, 18 representative rice hybrids were selected to generate 9,839 $F_2$ progenies (Extended Data Table 1) to further map loci associated with heterotic effect and to provide more diverse genetic resource for model construction. $F_2$ individuals were sequenced at an average depth of 0.2-fold, and genotyping of $F_2$ individuals based on high-density SNPs from the parental lines was conducted largely following previous description[21].

Phenotypic investigation was conducted for hybrids and $F_2$ individuals, involving heading date, morphological characteristics and grain yield-related traits (Supplementary Table 4). Moreover, grain quality-related traits were investigated for hybrids. Among our

collections, 123 hybrids had both their parental lines collected. On the basis of the parent-hybrid trios, better parent heterosis (BPH) was evaluated. Most hybrids showed BPH in grain yield-related traits. However, with respect to grain quality, most hybrids did not exhibit heterosis (Extended Data Fig. 2b).

To facilitate access to the resource, we constructed a web-based tool (http://ricehybridresource.cemps.ac.cn/#/). Users can rapidly access the sample, phenotype, variant and genome-wide association analysis (GWAS) information of interest. The tool also demonstrates the allele frequency and potential impact of variants (according to annotation by SnpEff[22]), and provides haplotype analysis.

### Phenotypic change in rice hybrid across improvement breeding

Improvement breeding for rice hybrid during the past five decades has undergone policy steering to breed 'super rice', the proliferation of two-line hybrid varieties in 2000s, and the progress of breeding technology. The process can roughly be divided into three periods (further details in Methods): breeding periods of 1976–2000 (hereafter Y1), 2001–2010 (Y2) and 2011–2020 (Y3) (Fig. 1a). Hybrids from the three periods contributed 11%, 47% and 42% of the total collection, respectively (Fig. 1b).

According to previous report, hybrids hold five types of cytoplasm[23]: Wild-abortive (WA), Boro II (BT) and Honglian (HL) cytoplasms from three-line breeding system, as well as Twoline-Jap (TJ) and Twoline-Ind (TI) cytoplasms from two-line breeding system. Phylogenetic analysis based on nuclear polymorphisms implied that WA, TJ and BT hybrids were clearly separated with only a few exceptions and that TI hybrids were genetically mixed with WA and TJ hybrids (Extended Data Fig. 3a). These findings were in close agreement with previous reports[23] and indicated that different genetic resources were exploited by different breeding systems. HL hybrids were excluded from further analysis due to the small population size (17 HL hybrids in our collection).

Phenotypic comparison of hybrids from three breeding periods was conducted. Of involved traits, heading date, source and sink organ traits, and grain quality changed significantly (Fig. 1c–k and Extended Data Fig. 4a). Correlation analysis between phenotype and year showed a consistent change trend (Extended Data Fig. 4b). The variance of heading date became less and the heading date was slightly shortened for Y3 hybrids (3 days on average) when compared with Y1 hybrids. Y3 hybrids had higher valid panicle number, full grain number per panicle and full grain number per plant than Y1 and Y2 hybrids. Furthermore, the flag leaf became longer and wider across the breeding periods, indicating that the modern rice hybrids were improved with larger source organs to correspond with larger sink organs. Grain quality, including cooking and appearance quality, were significantly improved during improvement breeding. In addition, polished grain

**Fig. 2 | Genomic structure analysis for rice hybrids and genome-wide scanning of *japonica* introgression in Wild-abortive and Twoline-Jap *indica*–*indica* hybrids. a**, Comparison of nucleotide diversity was conducted by the least significant difference (LSD) method with 'Bonferroni' correction for hybrids from three breeding periods. A 200-kb-length sliding window was used to scan the genome-wide sequence diversity, and the number of genomic segments with the sequence diversity calculated is shown in parentheses. Different lowercase letters above the box plots represent significant phenotype differences ($P ≤ 0.05$). **b**, PCA plots for the first three PCs. Colored dots represent hybrids bred from three breeding periods. **c**, Admixture analysis with *K* from 2 to 5. Samples are clustered according to breeding periods. The five subgroups of hybrids classified by cytoplasmic type are also indicated at the bottom. Individuals bred from the two-line breeding system are marked by half-height bar to better distinguish from the three-line system hybrids. **d**, PCA plots for the first three PCs. Hybrids possessing Wild-abortive and Twoline-Jap organelles are colored in green and light blue, respectively, with the remaining samples in gray.

**e**,**f**, Distributions of *japonica*-origin segments in Wild-abortive (**e**) and Twoline-Jap (**f**) hybrids. Orange bands represent the heterozygous introgression, and blue bands represent homozygous introgression. Colored heat map indicates the fraction of hybrids containing the two forms of *japonica*-origin segments. **g**, Accumulated length of introgression fragments by chromosome. Introgression fragments, presenting in more than 20% Wild-abortive or Twoline-Jap hybrids, were counted. **h**, Four candidate genes (*GW3p6*, *NAL1*, *GS6* and *Waxy*) were involved in *japonica* introgression. The pie plots show the proportion of three types of segment (*indica/indica*, *indica/japonica* and *japonica/japonica*) in genomic region where the target genes reside; the bar plots demonstrate the genotypic frequencies of the four candidate genes. The putative causal polymorphisms marked at gene body were used to define three genotypes. The ratio of introgression was not completely equal to the frequency of breeding-favorable allele because not all of the *japonica* introgression was associated with breeding-favorable allele introduction.

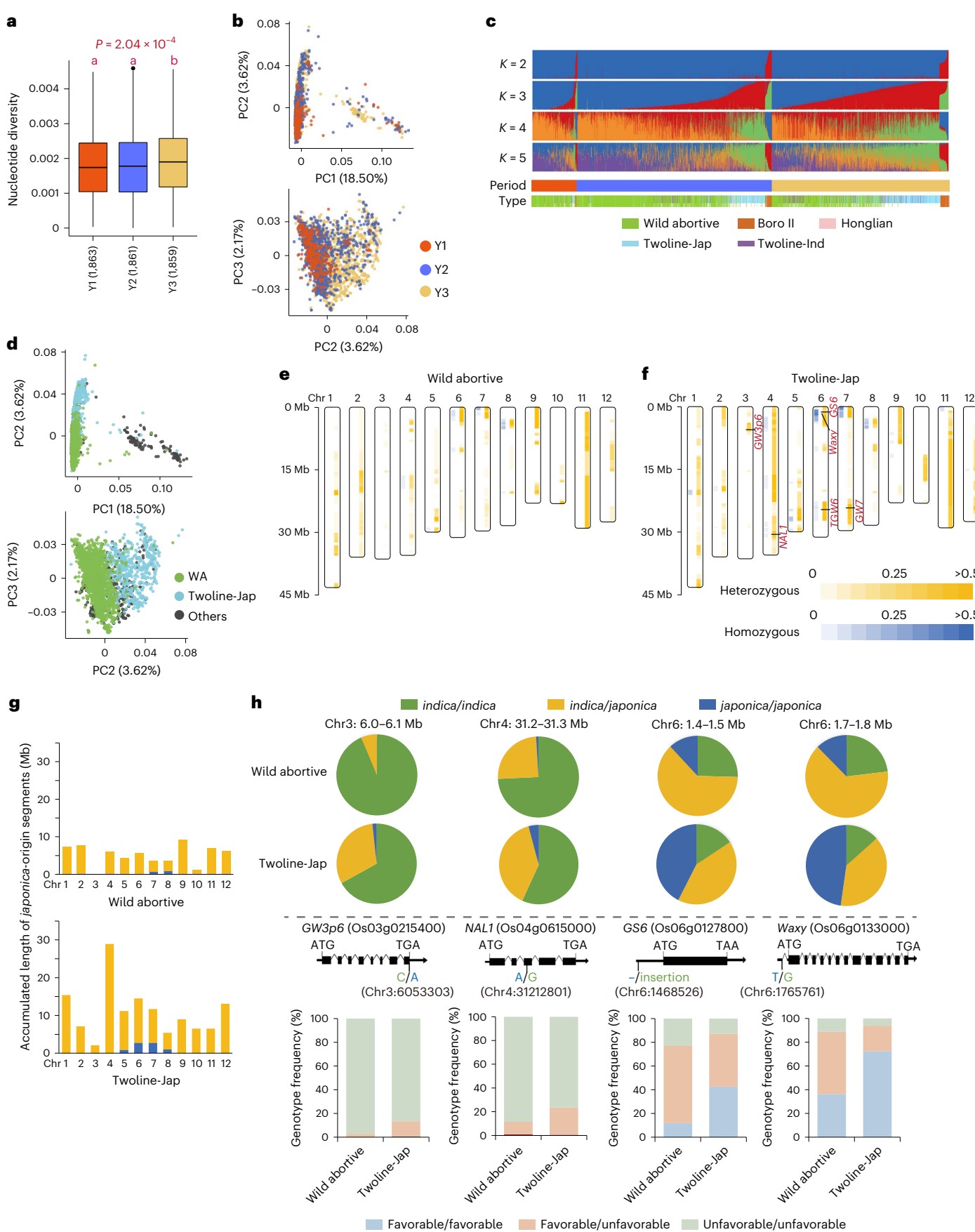

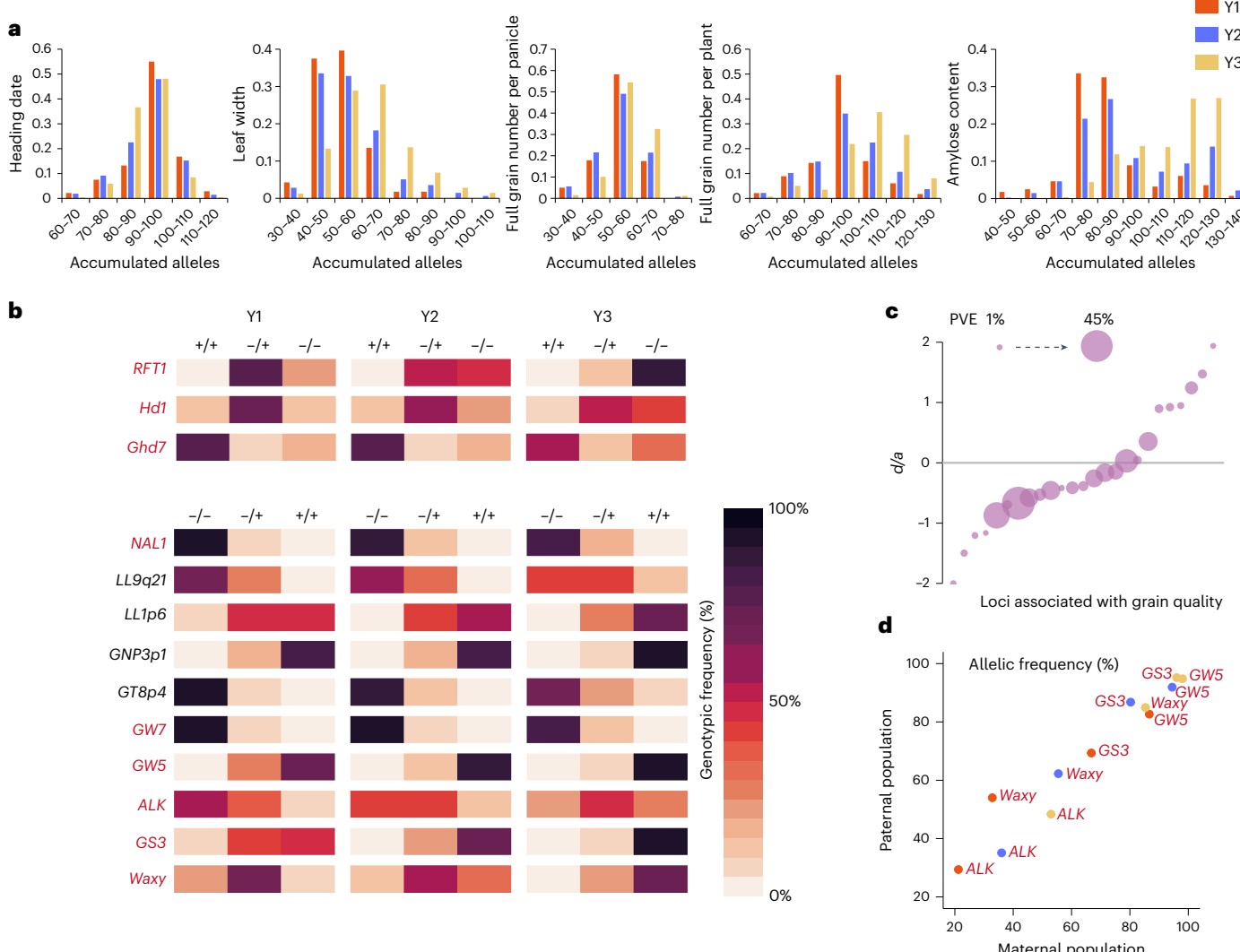

**Fig. 3 | Identification and analysis of loci associated with improvement breeding in *indica–indica* rice hybrids. a**, Distribution of the count of accumulated breeding-favorable alleles in hybrid individuals from three breeding periods for heading date, leaf width, full grain number per panicle, full grain number per plant and amylose content. The top-100 GWAS signals obtained from *indica–indica* rice hybrids were considered. A single locus contains two alleles, resulting in a cumulative count of 200 alleles across the aforementioned top-100 loci. **b**, Heat map demonstrating allelic frequency change within *indica–indica* rice hybrids across the three breeding periods. Symbol '+' indicates breeding-favorable alleles or longer heading date, while '−' represents breeding-unfavorable alleles or shorter heading date. **c**, Dominance-effect/additive-effect (*d/a*) for loci underlying grain quality-related traits and explaining larger than 1% phenotypic variance. **d**, Genotypic frequencies of four grain quality-related loci within both maternal and paternal populations of the *indica–indica* rice hybrids from Y1, Y2 and Y3 periods. Dots for Y1, Y2 and Y3 periods are marked by red, blue and yellow, respectively.

became increasingly elongated. Comparisons within WA, TJ, TI and BT hybrid subgroups were also conducted, revealing analogous change trends (Extended Data Fig. 3c).

Phenotypic changes demonstrated that the achievements of hybrid breeding in the past five decades were reflected in the substantial enlargement of source and sink organs, slight reduction of flowering time and improvement of grain quality.

**Enlarging genetic diversity by developing new germplasms**

By calculating genome-wide nucleotide diversity and analyzing genomic structure, we showed the rice hybrids became more diverse across the breeding process (Fig. 2a–c and Extended Data Figs. 1b and 3b). WA and TJ hybrids, as two predominant types of rice hybrid, exhibited differentiated genomic structure (Fig. 2d and Extended Data Fig. 3a). The number of TJ hybrids and the fraction of characteristic genomic components of TJ hybrids gradually increased from Y1 to Y3 (Fig. 2c). Given that most WA and TJ hybrids were *indica–indica*

hybrids but that TJ hybrids possessed *japonica* cytoplasms[23], we further surveyed the *japonica* introgression level. Across the whole genome, unequal levels of both heterozygous and homozygous introgression between WA and TJ hybrids were observed (the details for definition of heterozygous and homozygous introgression in Methods) (Fig. 2e–g), with higher introgressive level in TJ hybrids. We also identified several known genes involved in introgression events, such as *NAL1* (Os04g0615000) controlling leaf size and spikelet numbers[24,25], *Waxy* (Os06g0133000) associated with grain quality[26–28], *GW3p6* (Os03g0215400) (refs. 15,29,30) and *GS6* (Os06g0127800) (ref. 31) controlling grain size and weight (Fig. 2h). According to previous reports, the breeding-favorable alleles of the four loci were from *japonica* subspecies[15,24,25,28,31]. *Japonica*-origin introgression in *indica–indica* hybrids introduced key genetic resource. When comparing with WA hybrids, more TJ hybrids possessed the breeding-favorable alleles for the four genes. The result demonstrated that creation of new germplasm has promoted the widely utilization of favorable genetic resource.

## Identifying and quantitatively analyzing breeding signatures

GWAS was conducted to further capture genetic signatures of improvement breeding. We performed GWAS in 2,724 *indica–indica* hybrids (Extended Data Fig. 5a–e) to avoid strong population structure from the relatively small sample size of the *indica–japonica* and *japonica–japonica* hybrids. In addition, 4,497 F$_2$ individuals, derived from ten *indica–indica* hybrids, were subjected to GWAS separately to facilitate identification of additional heterotic loci (Extended Data Fig. 5f–h). On the basis of the peak SNPs of the top-100 associated loci identified in *indica–indica* hybrid rice for each trait, we analyzed the accumulation of breeding-favorable alleles in Y1, Y2 and Y3 rice hybrids. The number of Y3 individuals containing small (≤80) or large (≥100) count of alleles leading to short heading date was less than that in Y1 and Y2 periods (Fig. 3a). This result was consistent with the observation that the heading date became more concentrated in Y3 hybrids (Fig. 1c). Selecting suitable parental combinations to aggregate appropriate numbers of alleles helped keep the heading date within a reasonable range. The alleles leading to shortened heading date of gene *RFT1* (Os06g0157500) (ref. 32), *Hd1* (Os06g0275000) (ref. 33) and *Ghd7* (Os07g0261200) (ref. 34), increased from periods Y1 to Y3 (Fig. 3b). As for leaf size, full grain number per panicle/plant, yield per plant and grain quality, newly bred hybrid varieties tended to accumulate more breeding-favorable alleles than early-bred hybrids (Fig. 3a and Extended Data Fig. 6a). A newly identified locus *LL1p6* explained 3.75% phenotypic variance of leaf length in *indica–indica* hybrid population (Extended Data Figs. 6b and 7a,b). It was also identified by linkage mapping in a F$_2$ population and had *d/a* index estimated at 0.15, indicating that the magnitude of the dominance effect was modest and the effect of heterozygous genotype approached to the mid-parent value (Extended Data Fig. 7c). More Y3 hybrids from WA and TI subgroups held the advantageous homozygous genotype (Fig. 3b and Extended Data Fig. 6c). Another newly identified locus, *LL9q21*, controlling leaf length, had *d/a* estimated at 0.4 in a F$_2$ population, indicating a partial positive dominance effect (the performance of heterozygote was better than mid-parent value but was not as good as advantageous homozygote) (Extended Data Fig. 7d–f). The frequency of breeding-favorable allele increased during improvement breeding by promoting usage of heterozygous genotype in WA and TJ hybrids and advantageous homozygous genotype in TI hybrids. One association signal near the known gene *NAL1* (refs. 24,25) respectively explained 12.71% and 8.11% phenotypic variance for leaf width and grain number per panicle in hybrids, with *d/a* value estimated at 0.61 and 0.79. The number of WA and TI hybrids holding heterozygous genotypes slightly increased during improvement breeding. A newly identified locus *GNP3p1* explained 1.89% of phenotypic variance for full grain number per panicle in hybrids, and had its *d/a* value estimated at −0.14, indicating a negative dominance effect (the performance of heterozygote was worse than mid-parent value) (Extended Data Figs. 6b and 7g,h). The breeding-favorable homozygous genotype for *GNP3p1* increased in Y3 hybrids. A newly identified locus *GT8p4* explained 14.09% and 6.76% phenotypic variance for grain translucency and chalkiness in hybrids, with *d/a* value respectively estimated at 0.35 and 1.24 (indicating partial positive dominance and over-dominance effects) (Extended Data Figs. 6b and 7i–l). The heterozygous genotype

increased from Y1 to Y3 periods. Furthermore, the well-known genes, *Waxy* (refs. 26,27), *ALK* (Os06g0229800) (ref. 26), *GW5* (Os05g0187500) (refs. 35–38), *GS3* (Os03g0407400) (ref. 39) and *GW7* (Os07g0603300) (refs. 40,41), were identified as master loci controlling grain quality by GWAS. For those genes, frequencies of the breeding-favorable alleles increased during improvement breeding (Fig. 3b). Most quality-related loci with phenotypic variance explained (PVE) larger than 1% represented negative dominant effects (Fig. 3c and Supplementary Table 5). It was consistent with the observation that heterosis was uncommon for grain quality in rice hybrid cultivars (Extended Data Fig. 2). For four master loci representing negative dominance effects on grain quality (Extended Data Fig. 6b), comprising *Waxy*, *GW5*, *ALK* and *GS3*, we investigated the allelic frequency change in both parental populations. We found both parental lines were simultaneously equipped with breeding-favorable alleles to ensure the utilization of advantage homozygous genotypes in hybrids for improvement breeding (Fig. 3d). Through these overall findings, we conclude that constantly pyramiding the breeding-favorable alleles by optimizing heterotic combinations and improving parental lines to utilize homozygous genotype for genes exhibiting negative dominant or additive effects, contributes to the phenotypic improvement for rice hybrids.

It is worth mentioning that *GW5* was also identified to control kilo-grain weight. However, the allele of *GW5* with higher grain weight had a negative impact on grain quality (Extended Data Fig. 7m–t). Given that the heterozygous genotype of *GW5* displayed a negative dominance effect on grain quality, the allele with higher grain weight was discarded in favor of grain-quality improvement. This indicates that the moderate increase of yield per plant (Extended Data Fig. 4a) might be a result of pursuing a balance between grain yield and quality.

## Genetic basis of intersubspecific heterosis

Although the development of intersubspecific hybrid was still in initial stage and the subset of *indica–japonica* hybrids was small, it has garnered substantial attention due to its great yield potential and strong heterosis[4]. Here, on the basis of 68 *indica–japonica* F$_1$ hybrids and 5,342 F$_2$ individuals derived from eight elite *indica–japonica* rice hybrids, we explored the genetic basis of intersubspecific heterosis. Among the *indica–japonica* hybrids, 65.47% genome-wide segments were *indica/japonica* genotype (Fig. 4a). At the quantitative trait nucleotides (QTNs) in 17 agronomically important genes reported previously in rice[42], bi-parents for most *indica–japonica* hybrids contained differentiated genotypes (Fig. 4b). Out of the 64 significant signals identified by association analysis in *indica–japonica* F$_2$ population (Extended Data Fig. 5i–k), 55 loci were situated at the genomic region where *indica/japonica* genotypes were present in more than half of the intersubspecific hybrids (Fig. 4c), and included the well-known genes, *DEP1* (Os09g0441900), *Ghd7*, *Ghd8* (Os08g0174500) and *Hd1*, which were differentiated between *indica* and *japonica* subspecies according to previous report[43–46]. These results indicated genetic complementation was prevalent in intersubspecific hybrids. We further evaluated the effects of genetic complementation by F$_2$ individuals derived from an *indica–japonica* hybrid Quanjingyou No.1. In the hybrid, 0.98%, 59.22% and

**Fig. 4 | Genome-wide genetic complementation contributing to intersubspecific heterosis in rice hybrids. a**, Distribution of *indica*- and *japonica*-origin segments across the whole genome of 68 *indica–japonica* hybrids. The *indica–indica*, *indica–japonica* and *japonica–japonica* genomic sequence are indicated by red, yellow and blue, respectively. **b**, Scanning the genotypes for 19 QTNs in 17 agronomically important genes in maternal and paternal lines of the *indica–japonica* hybrids. Breeding-favorable and breeding-unfavorable genotypes are marked in red and blue, respectively. Rows represent hybrids and columns represent QTNs. Each strip is composed of the genotype of maternal (left) and paternal (right) lines of a hybrid individual. **c**, Proportion of three types of segment at the genomic region where the significant association

signals identified in *indica–japonica* F$_2$ populations reside. **d**, Distribution of three types of segment across the whole genome of Quanjingyou No.1. The gray area represents hypothetical chromosomes. In each gray-shaded region, the lower stripes represent SNPs along the hypothetical chromosome region, and the higher bar indicates genomic sequence with its genotype judged by the SNP markers distributed in it. **e**, The dominance-effect/additive-effect (*d/a*) of the four major loci. Loci with breeding-favorable genotype contributed by female and male lines are marked in red and blue, respectively. **f**, Distribution of grain number per plant of F$_2$ individuals containing heterozygous genotypes across all four loci. The performance of both parental lines is also indicated by double-height stripes.

39.80% genomic segments were *indica/indica*, *indica/japonica* and *japonica/japonica* genotypes, respectively (Fig. 4d). Four major loci, controlling total grain number per plant and having PVE exceeding 3%, all resided at the *indica/japonica* complementary genomic region (Fig. 4d). For loci at bin0357 (*sd1* (Os01g0883800) located nearby) and bin1687 (*Hd1* located nearby), breeding-favorable alleles were contributed by maternal line and their *d/a* were evaluated at 1.40 and 0.24,

respectively. Loci at bin0796 and bin2547 (*DEP1* (Os09g0441900) located nearby) were from paternal line, with *d/a* evaluated at 0.09 and 0.01 (Fig. 4e). Most (76%) $F_2$ individuals possessing the heterozygous genotypes across all the four loci had more grain number per plant than both parental lines (Fig. 4f). Genetic complementation of the four master loci largely accounted for transgressive segregation of grain number per plant in the $F_2$ population.

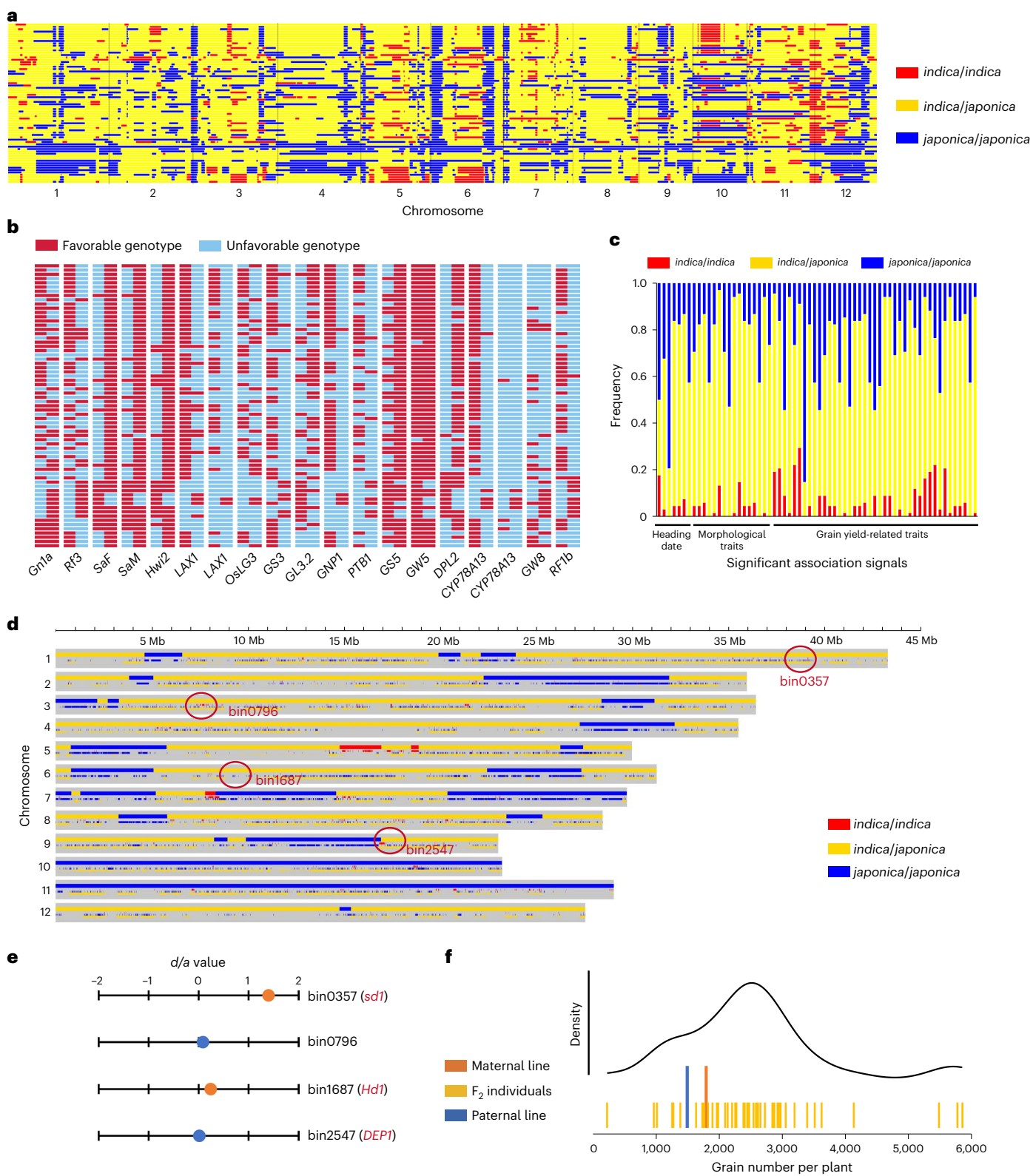

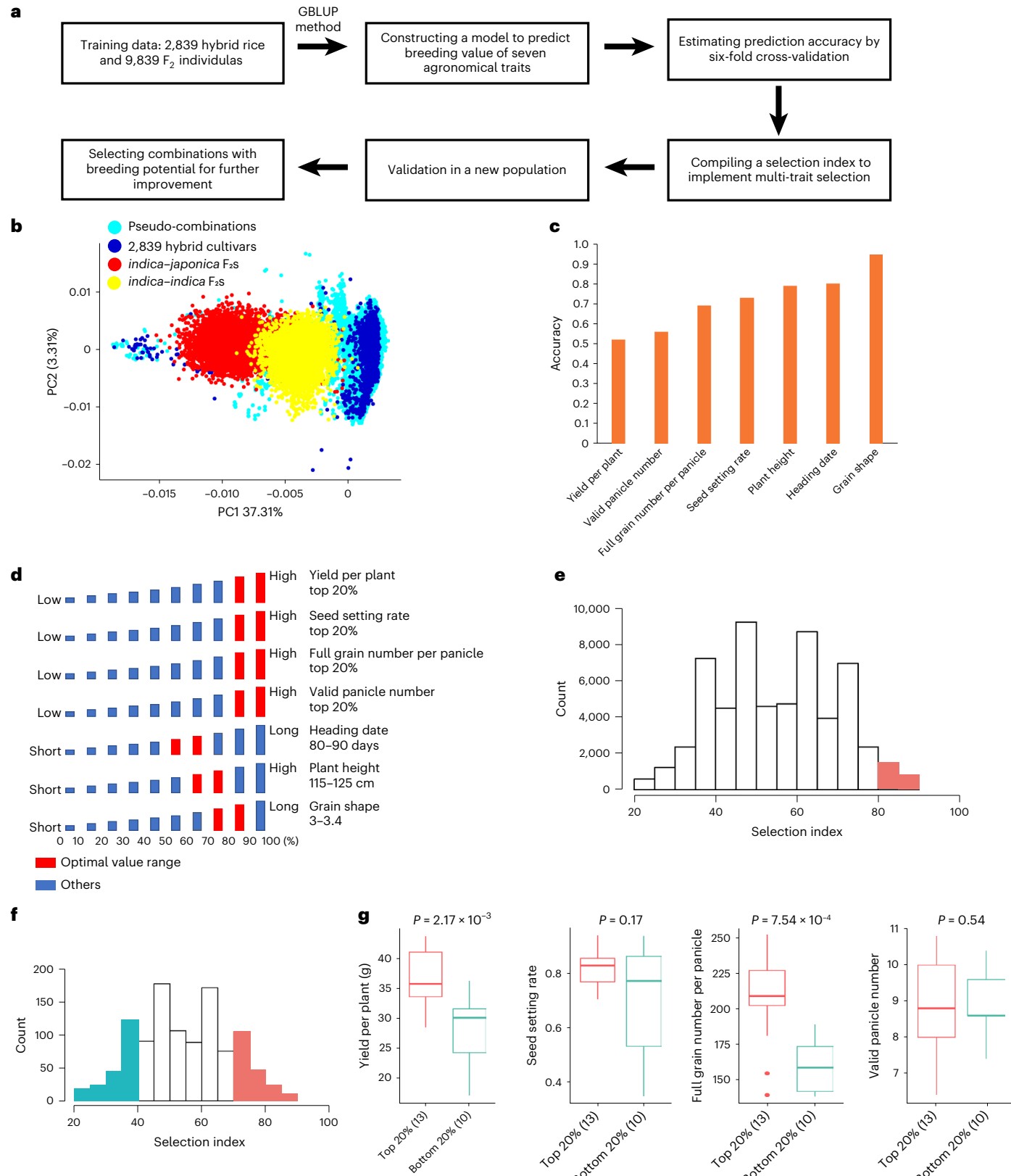

**Fig. 5 | Construction and validation of the genomic selection model for hybrid rice breeding. a**, Schematic diagram for model construction and validation. **b**, PCA plot for all materials used in the construction and validation of genomic selection model. **c**, Prediction accuracy for seven selected agronomic traits. **d**, Diagram demonstrating the optimal value range (red) of seven traits for selection index calculation. **e**, Distribution of the selection index for 58,353 pseudo-combinations. Combinations with high scores (≥80) are visually emphasized in red. **f**, Distribution of the selection index for 1,102 pseudo-combinations from the validation population. The combinations located within uppermost 20% and lowermost 20% are respectively indicated by red and blue. **g**, Comparison of four yield-related traits between high-scoring and low-scoring combinations in the validation population. Sample size is shown in parentheses. Significance test was conducted by one-sided (greater) Wilcoxon test.

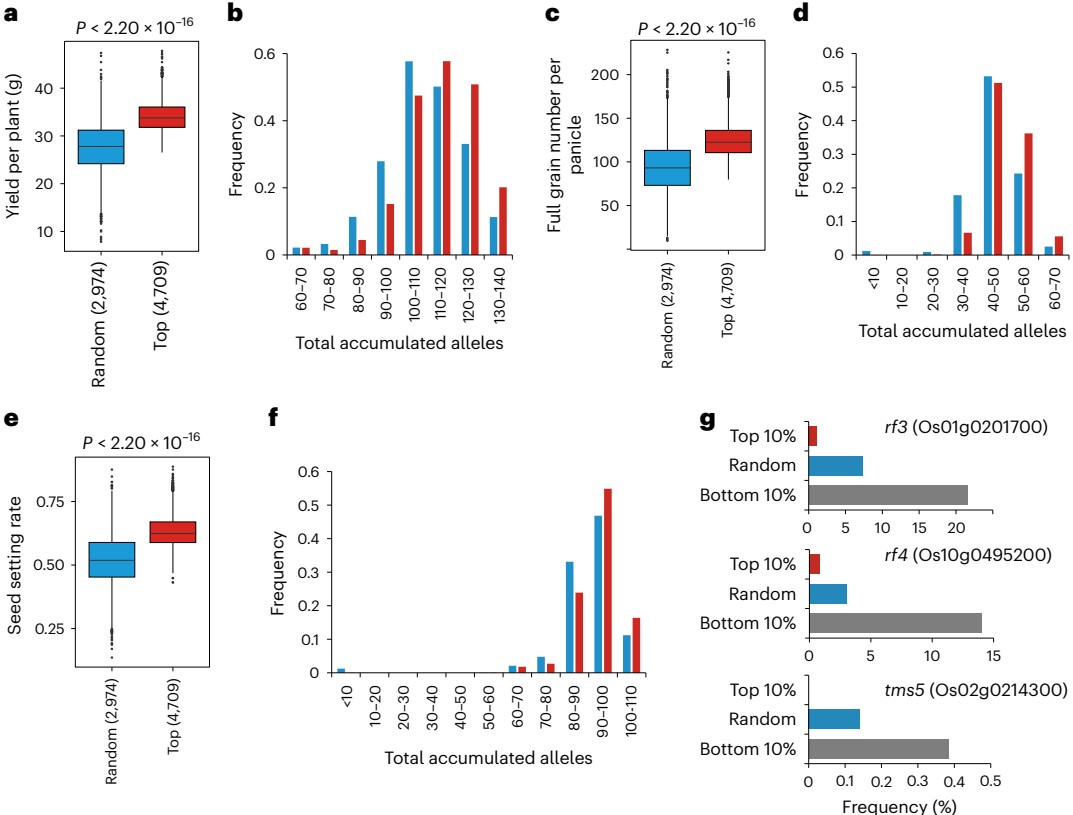

**Fig. 6 | Correlation between yield potential and the total accumulation of breeding-favorable loci. a,c,e**, GEBV of yield per plant (**a**), full grain number per panicle (**c**) and seed setting rate (**e**) for Wild-abortive hybrid combinations scored in the uppermost 10% or from random selection. Sample size is shown in parentheses. Significance test was conducted by one-sided (less) Wilcoxon test. **b,d,f**, Distribution of the count of accumulated breeding-favorable alleles in Wild-abortive hybrids combinations ranked in uppermost 10% or from random selection for yield per plant (**b**), full grain number per panicle (**d**) and seed setting rate (**f**). For each of the three traits, the top-100 GWAS signals and corresponding QTNs reported previously were used for analysis. **g**, Bar plot demonstrating the proportion of the combinations containing the genotypes causing pollen abortion for genes *rf3*, *rf4* and *tms5* in uppermost 10%, lowermost 10% or randomly selected subgroups.

Furthermore, for the yield-related traits exhibiting BPH in most hybrids (Extended Data Fig. 2b), we conducted linkage mapping in eight *indica–japonica* F$_2$ populations. On the basis of all identified QTLs, we evaluated the contribution of dominance and overdominance effects to intersubspecific heterosis. For yield per plant, full grain number per panicle, full grain number per plant and seed setting rate, the number of QTLs with partial dominance and overdominance effects was relatively balanced. However, with the exception of the seed setting rate, QTLs with partial dominance effects accounted for a relatively larger PVE (Extended Data Fig. 8). It was worth noting that the over-dominance loci mentioned here may also include the pseudo-overdominance loci, which was conferred by tightly linked genes in opposite phase. With respect to kilo-grain weight, whether in terms of count or contribution of phenotypic variation, QTLs with positive partial dominance effects played the leading role.

The overall findings indicate that genetic complementation for loci exhibiting dominance and over-dominance effects has greatly contributed to intersubspecific heterosis in yield traits, and loci with partial dominance effects assume more critical roles, attributable to their larger PVE.

## Constructing a genomic-selection model

We used the genomic and phenotypic data from the 2,839 rice hybrids and 9,839 F$_2$ individuals to construct a genomic selection model for optimizing rice hybrid breeding (Fig. 5a). The numerous F$_2$ individuals provided novel genotypes of transgressive segregations, which

frequently occurred in plant breeding[47]. Furthermore, the F$_2$ populations, especially the *indica–japonica* F$_2$ population, had obvious population stratification with the F$_1$ population (Fig. 5b and Extended Data Fig. 1c). Adding numerous *indica–japonica* F$_2$ individuals to the training model offset the lack of *indica–japonica* hybrids in the F$_1$ population. On the basis of genotype data of 88,909 SNPs and phenotype data of seven key agronomical traits from 12,678 individuals, we constructed a genomic selection model by genomic best linear unbiased prediction (GBLUP) method. The prediction accuracies for yield per plant, valid panicle number, full grain number per panicle, seed setting rate, plant height, heading date and grain shape were 0.518, 0.559, 0.689, 0.728, 0.790, 0.799 and 0.945, respectively (Fig. 5c). Because the training data containing adequate samples from four subgroups of hybrids, we found the prediction accuracies within subgroups were comparable to that of all hybrids (Extended Data Fig. 9b). We compiled a selection index integrating all of the above-list traits to implement multi-trait selection. Phenotypic observation of plant height, heading date and grain shape need to be controlled in an appropriate range in agricultural practice, and so the optimal value range was set on the basis of the phenotypic distribution in cultivated hybrid (Fig. 5d). For the rest of four grain yield-related traits, the best value range was set as the highest 20%. High score was allocated to genomic estimated breeding value (GEBV) located in the best value range, and the selection index was calculated by summing up all the scores of seven traits. We scored all 58,353 pseudo-combinations between 367 and 159 commercial parental and maternal lines[23]. Compared with hybrid cultivars,

pseudo-combinations had higher genomic diversity (Fig. 5b), and the high-scoring pseudo-combinations, which have not yet been selected by breeders, might have great breeding potential (Fig. 5e).

To further verify the accuracy of the model, all 1,102 possible combinations with no intersection with training data were generated by combining 58 inbred lines and 19 commercial female lines. *Indica*, *japonica* and intermediate inbred lines, and WA, TJ, TI and BT female lines, were all included to test the wide applicability of our model (Supplementary Table 6). The selection index was calculated for all combinations (Fig. 5f). Among them, 67 combinations were randomly selected and planted to investigate yield-related traits in an experimental field in Shanghai, China. According to the selection index, 13 materials were scored in the top 20% and 10 materials were scored in the bottom 20%. Phenotypic comparisons between high-scoring and low-scoring combinations were conducted (Fig. 5g). High-scoring combinations had better performance in yield per plant and full grain number per panicle. With respect to seed setting rate, although there was no obvious significant difference between high-scoring and low-scoring combinations, high-scoring combinations had lower phenotypic variation (Fig. 5g). Furthermore, a representative combination with a high score was selected to conduct genome-wide scanning for QTNs controlling key agronomical traits[42], and a customized project for improvement was proposed (Extended Data Fig. 9c,d). These results demonstrate that the prediction model is effective for genomic selection in a completely new population.

### Breeding potential linked to favorable allele pyramiding

On the basis of the selection index of 58,353 combinations described above, we further analyzed the associations between the number of accumulating breeding-favorable alleles and breeding potential. For yield per plant, grain number per panicle and seed setting date, the top-100 associated loci from hybrid GWAS analysis as well as reported QTNs[42] were used for analysis. For all three traits, compared with randomly selected WA combinations, WA combinations ranked in top 10% had significantly better GEBV (Fig. 6a,c,e), and tended to accumulate more breeding-favorable alleles (Fig. 6b,d,f). The result was consistent with the observation in rice hybrid improvement breeding (Fig. 3b–e): pyramiding more breeding-favorable alleles contributed to phenotypic improvement. This is also observed in maize hybrid[48]. Furthermore, incorporation of seed setting rate into the selection index allows the model to screen out the combinations with failure in fertility restoration. High-scoring WA combinations rarely possessed the homozygous genotype of *rf3* (Os01g0201700) (ref. 49) and *rf4* (Os10g0495200) (ref. 50), both of which were responsible for wild-abortive sterility. Moreover, there was no high-scoring TJ combination containing the homozygous genotype of *tms5* (Os02g0214300) (refs. 51,52), which contributed to environment-sensitive genic male sterility (Fig. 6g). However, a number of random selections or bottom-scoring combinations possessed the homozygous genotype of *rf3*, *rf4* and *tms5*.

### Discussion

In our study, a large set of rice hybrid varieties covering the entire breeding course were resequenced and phenotyped. This enabled a comprehensive genome analysis and quantitative genomics study to capture genetic footprints of improvement. Rice hybrid breeding is a complicated process with multiple rounds of genetic improvements to achieve high grain yield, high grain quality and adaptability to changed environments. In agricultural practice, the climatic condition is changeable in the late growth period of cultivated rice. Slightly shortened growth period observed in improvement breeding could aid stable production. However, the shortened growth period and the pursuit of better grain quality, pose challenges in maintaining grain yield. As part of the improvement breeding efforts, significantly enlarged source and sink organs, which are closely linked to rice grain yield[24,25], have helped

maintain a balance between grain yield and quality, particularly under the shortened growth period.

Different from the population bottleneck and genetic diversity reduction during domestication and improvement breeding in inbred cultivars[53,54], genetic diversity of rice hybrids increased during improvement breeding. The development of commercial hybrid rice began with the discovery of wild abortive resource in wild rice in 1970s, and this promoted the creation of the three-line system[55,56]. Subsequently, the discovery of environment-sensitive genic male sterility resource in a cultivated *japonica* rice facilitated the creation of the two-line system[57,58]. As reported in previous studies, two-line and three-line hybrids were genetically differentiated, which probably originated from the differentiation in founder cultivars of different breeding systems[23,59]. We found TJ hybrids from two-line system and WA hybrids from three-line system were genetically differentiated, and TJ hybrids had higher *japonica*-introgression levels than WA hybrids. During improvement breeding, the rapid increase of TJ hybrids broadened the genetic diversity. Introgression events also introduced breeding-favorable alleles from the *japonica* subpopulation. This result suggested that the development of different mating systems in hybrid production by exploiting the new types of male-sterility resource also diversified rice hybrid germplasms and enlarged gene pools for hybrid breeding.

The genomic selection model developed in this study presented a valuable tool for breeders to search efficiently for optimal cross-breeding combinations and accelerate breeding cycles while reducing costs. Given the significance of non-additive effects in rice hybrid heterosis[14], we used additive plus dominance model in genomic selection. However, the additive plus dominance model did not yield an improvement in prediction accuracy in comparison to the additive only model (Extended Data Fig. 9a). It has been reported that it is difficult to increase prediction accuracy by adding dominance in genomic selection models[60], and our work further supports this claim using our large-scale training data. The major causes might be: (1) the magnitude of dominance effects varied across different loci, posing a challenge for reliably estimating the dominance effects; (2) including dominance in model elevated complexity, hindering precise estimation of marker weights. Current models were inadequate in handling the non-additive effect, and the community will need more time to construct a new model to address this issue. The breeding potential predicted by our selection model could help identify core collections from extensive germplasm resources, thereby serving as backbone materials for further improvement. In the future, with the characterization of more quantitative trait genes and better understanding of genetic frameworks, molecular design can be implemented to fully develop the potential of heterosis.

### Online content

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

## Methods

Ethics approval was not required in this article.

### Plant materials and phenotypic investigation

All 2,839 rice hybrids were from the collections preserved at the China National Rice Research Institute in Hangzhou, China, comprising 1,495 rice hybrid accessions reported previously[13]. According to information recorded by the China Rice Data Center (https://www.ricedata.cn/), 2,715 out of 2,839 hybrids had registration information and were released from year 1985 to 2017. We performed Y1 to Y3 classification for 2,715 hybrids based on the following: (1) Policy steering in late 1990s and 2000s. China Ministry of Agriculture established a national mega project entitled 'Breeding and cultivation system of super rice in China' in 1996. The project encouraged combined use of heterosis and ideal plant type to increase yield, and they set standards for ideal plant type of super rice: erect panicle model, long erect leaves with delayed senescence model, early vigorous growth and heavy panicle model[4]. (2) Boom of two-line hybrids in 2000s (Fig. 2c and Extended Data Fig. 3b). (3) Progress of breeding technology. The finished quality sequence of rice genome was released in 2005 (ref. 61). With the reference sequence available and development of high-throughput sequencing and genotyping methods, rice molecular biology and functional genomics made great progress, which further accelerated the rational design and molecular breeding of hybrid super rice by marker assisted selection in the past decade[4,62].

All hybrid accessions were germinated and planted in an experimental field in Hangzhou, China in summer of 2020. All the $F_1$ lines were grown in the consecutive farmland with well-distributed soil status. For each accession, 15 samples were germinated and planted in three rows with row and column spacing setting to 20 and 26.7 cm, respectively. Plants in the middle of the plot were selected to investigate phenotype, involving morphological characteristics (leaf length, leaf width, plant height and tiller angle), yield components and relevant factors (full/total grain number per plant, valid panicle number, full grain number per panicle, kilo-grain weight, seed setting rate and yield per plant), grain quality-related traits (amylose content, gel consistency, chalkiness, chalky grain percentage, grain translucency and grain shape) and physiological feature (heading date). Among them, heading date and morphological characteristics were measured in the field. The heading date was recorded as the duration in days from the date of sowing to the emergence of first inflorescences above flag leaf sheath of five plants for each accession. The other field traits were investigated after heading, and three individuals were selected for phenotyping: the tallest panicle was selected to investigate plant height, flag leaf length and width; tiller angle was measured between tiller and the ground level. The collection of observations for grain-related traits was carried out in the laboratory after harvest, and they were investigated with three biological replicates for each sample. Full/total grain number per plant was counted manually; valid panicle number was directly counted after removing ineffective panicles; full grain number per panicle was estimated by dividing the full grain number per plant by the valid panicle number; seed setting rate was estimated by calculating the ratio of full grain number to total grain number per plant; kilo-grain weight was obtained by weighing 400 fully filled grains and then converting the value to 1,000-grain weight; yield per plant was recorded as the weight of all full grains per plant. Grain quality traits were investigated according to the Chinese national standard (NY/T 2334−2013 'Determination of head rice yield, grain shape, chalky grain percentage, chalkiness degree and translucency−an image analysis method'). Grain shape was measured as length-to-width ratio of polished grain, which was the grain having its husk and outer brown layers removed. Grains from mixed harvest of the same accession were randomly selected to investigate grain quality-related traits, with two replicates.

Eighteen hybrid accessions were chosen to construct $F_2$ populations, generating a total of 9,839 $F_2$ lines. The 18 hybrids were carefully selected to encompass a comprehensive range of major hybrid rice types. Specifically, they consisted of ten intrasubspecific and eight intersubspecific combinations; additionally, they were also categorized into five WA, three TJ, three TI and seven BT hybrids according to cytoplasmic types, providing a diverse representation within the chosen set. The planting standards utilized for $F_2$ individuals were as those employed for $F_1$s. Similar to the procedures employed for $F_1$s, the investigation of physiological feature, morphological characteristics, yield components and relevant factors was also carried out for the $F_2$ lines. No replication was prepared for $F_2$ lines because $F_2$ individuals have different genotypes.

### Sequencing and variation calling

Leaves were collected from plants in vegetative growth phase in the field for each accession, and genomic DNA was then extracted from the fresh leaf tissue using DNeasy Plant Mini Kit (Qiagen). A sequencing library with ~400 bp insert size was constructed by Trueprep Tagment 3 Enzyme (Tn5 transposase, Vazyme). The amplification-free method was used for library construction to accurately judge heterozygous sites by reducing duplicate sequences. Sequencing was performed on the NovaSeq 6000 platform, generating 150 bp paired-end reads.

Quality control was conducted using the software Trimmomatic (version 0.38) (ref. 63), and bases of low quality or from adapter contamination were removed by the tools with parameters 'ILLUMINACLIP:TruSeq3-PE.fa:2:30:10:2:true MAXINFO:50:0.6'. The clean reads were then aligned against the rice reference genome IRGSP 1.0 by software BWA (version 0.7.1) (ref. 64) with parameters of '-R '@RG\tID:SampleID\tPL:Illumina\tSM:SampleName' -M'. Read alignment results were sorted according to their coordination by 'SortSam' functions in Genome Analysis Toolkit v4.1.4.1 (GATK) (ref. 65) with default parameters. Variations were called using 'HaplotypeCaller', 'GenomicsDBImport' and 'GenotypeGVCFs' functions in GATK. High-quality variations were contained by the 'VariantFiltration' function in GATK with parameters of '−cluster-size 3 −cluster-window-size 10 QD < 10.00 FS > 15.000 AC < 3 DP > 200||DP < 5' for SNPs and 'QD < 10.00 FS > 30.000 DP > 200||DP < 5' for InDels.

### Genetic diversity analysis

Nucleotide diversity was calculated with VCFtools (version 0.1.15) (ref. 66) using 5,222,902 high-quality SNPs. Kinship analysis was conducted to identify hybrid pairs sharing close relationships in the same breeding period, allowing for further assessment of genomic diversity of rice hybrids. Kinship coefficients were estimated using the 'emmax-kin' program in EMMAX software (version emmaxbeta-07Mar2010) (ref. 67) with default parameters. The cutoff value of kinship coefficient was set as 0.9. The result was visualized using Cytoscape desktop software version 3.8.2 (ref. 68). A dot represented a hybrid accession and was colored according to its cytoplasm type. Pairs of hybrids with kinship coefficient greater than 0.9 were connected by a beeline. Principal component analysis (PCA) and admixture analysis were based on 223,533 fourfold degenerate (4DTv) sites. SNPs were annotated by a gene model (snpEff_v4_3_ENSEMBL_BFMPP_32_268.zip) in software SnpEff (v 4.3t) (ref. 22), and the 4DTv sites were identified using customized Python script from the annotated SNPs. PCA was performed by software GCTA (Genome-wide Complex Trait Analysis v1.93.2 beta) (ref. 69) with default parameters. The highest-ranked three principal components were chosen for visualizing the population structure. Ancestral components for hybrids were inferred by the ADMIXTURE program (version 1.3.0) (ref. 70) with K values from 2 to 5, and the population admixture result was visualized using the R package pophelper (version 2.3.1) (ref. 71).

### Genome-wide identification of introgression from the *japonica* ancestry

Zhao et al. have reported sequence variants for 66 divergent rice accessions that included 19 *O. sativa indica* and 22 *O. sativa* temperate

*japonica* accessions[72]. An additional accession of *O. sativa* temperate *japonica* sequenced by us was also included for *indica–japonica* differentiated SNP identification. In total, 19 *indica* and 23 temperate *japonica* accessions were used for analysis (Supplementary Fig. 1a). We identified intersubspecific differentiated SNPs according to the following criteria: at the SNP site, (1) ≥17 *indica* varieties were the same genotype; (2) ≥21 *japonica* varieties held the same genotype; (3) the major alleles in *indica* and *japonica* populations were different. We identified 830,245 *indica–japonica* differential SNPs in total.

On the basis of the differentiated SNPs, introgressive *japonica* fragments were detected across the whole genome: First, we analyzed the genotypes (as *indica/indica*, *japonica/japonica* or heterozygous genotype) at the 830,245 differentiated loci in the samples to be tested. Then, a 199-SNP sliding window with 1-SNP step was applied to identify introgressive fragments. Within the 199-SNP length fragment, if there were at least 120 SNPs with homozygous genotypes of *japonica* origin, the segment was labeled as homozygous *japonica* introgression; if there were at least 120 SNPs with heterozygous genotypes (*indica/japonica*), the segment was deemed as heterozygous *japonica*-introgressive fragments. Adjacent fragments in the same state were then combined for further analysis.

To verify the capacity of our method to distinguish *indica*- or *japonica*-origin sequences, representative *O. sativa japonica* cv. Nipponbare and *O. sativa indica* cv. Shuhui498 (ref. [73]) were chosen to scan for the distribution of *indica*- or *japonica*-origin segments across the whole genome, respectively. Out of 830,245 loci, 99.97% SNPs from cv. Nipponbare were judged as of *japonica* origin and all the 199-SNP length fragments across the whole genome were considered to be of *japonica* lineage. For cv. Shuhui498, 95.39% SNPs were judged as of *indica* origin and all the 199-SNP length fragments were identified as of *indica* lineage (Supplementary Fig. 1b,c). These findings indicated that the method based on the 830,245 *indica–japonica* differentiated loci had the capacity to distinguish *indica*- or *japonica*-origin sequence accurately.

## GWAS

GWAS was performed for three cohorts separately, including 2,716 *indica* rice hybrids, 4,497 $F_2$ progenies from ten *indica–indica* rice hybrids and 5,342 $F_2$ progenies from eight *indica–japonica* rice hybrids. Association analysis was conducted with the mixed linear model in the TASSEL software package (version 5.0 Standalone)[74] for all three cohorts. For rice hybrid cohort, the high-quality SNP data were further filtered by software PLINK (v.1.90b6.12 64-bit)[75] to keep variants with a missing rate ≤10% and minor allele frequency ≥5%. For $F_2$ progenies, every individual was genotyped on the basis of high-density SNPs of both parental lines, as described previously[21], and further imputed by the genotyping results and SNP information in both parental lines using a customized Perl script. In total, 1,296,386 and 1,059,427 SNPs were supplied to perform GWAS analysis for *indica–indica* and *indica–japonica* $F_2$ progenies, respectively. PCA was performed using the input genetic markers by the software GCTA[69], and the first two principal components were incorporated as the covariates to effectively account for population structure. Kinship matrix was generated based on the input genetic markers by 'Kinship' function in TASSEL. The significant threshold was set as $1 \times 10^{-6}$ for all three cohorts.

The dominance-effect/additive-effect (*d/a*) for association signal was calculated based on genotype effect estimated by TASSEL. The genotype effects of peak SNP in target association signal were chosen to calculate *d/a* index:

$$a = \left| \frac{A - C}{2} \right|$$

$$d = M - \frac{A + C}{2}$$

where *A* and *C* respectively represented the genotype effects of two homozygous genotypes, and *M* represented the genotype effect of heterozygous genotype. For traits with high observation preferred by breeding practice (for example, gel consistency), the degree of dominance was quantified by calculating the ratio of *d/a*; for traits with low observation preferred by breeding practice (for example, chalkiness), the degree of dominance was quantified by calculating the ratio of −*d/a*. If the number of observations of any homozygous genotype was less than 5, the estimated effect could be influenced by outliers and might be unreliable. Thus, such locus was excluded in the calculation of *d/a* index.

Furthermore, for QTL mapping by $F_2$ population, the index of *d/a* was estimated by IciMapping software (version 4.2.53) (ref. [76]).

## Phenotypic variance explained by association loci

The PVE by the candidate region surrounding the association signal was estimated according to previously report[77]. The candidate regions were the 1 Mb region surrounding the genome-wide association signals. A mixed linear model with multiple random effect was applied to estimate the variance components using the R package sommer (version 4.2.0.1) (refs. [78],[79]). The model can be written as:

$$y = Lb + T_1 u_{a1} + T_2 u_{d1} + T_3 u_{a2} + T_4 u_{d2} + e$$

where **y** is a vector of the phenotypic value, **b** is a vector of fixed effects, and $u_{a1}$, $u_{d1}$, $u_{a2}$ and $u_{d2}$ are vectors of random effect. $L$, $T_1$, $T_2$, $T_3$ and $T_4$ are incidence matrices for **b**, $u_{a1}$, $u_{d1}$, $u_{a2}$ and $u_{d2}$, respectively. **e** represents a matrix of residual effects and $e \sim N(0, I\sigma_e^2)$, in which $I$ is an identity matrix and $\sigma_e^2$ is the residual variance. The variable $u_{ai} \sim N(0, k_{ai}\sigma_{uai}^2)$, in which $k_{ai}$ represents the variance covariance matrix constructed by the additive relationship matrix A based on SNPs within ($i = 1$) and outside ($i = 2$) the candidate region, and $\sigma_{uai}^2$ is the genetic variance of the corresponding random effect; the variable $u_{di} \sim N(0, k_{di}\sigma_{udi}^2)$, in which $k_{di}$ represents the variance covariance matrix constructed by the dominance relationship matrix D based on SNPs within ($i = 1$) and outside ($i = 2$) the candidate region, and $\sigma_{udi}^2$ is the genetic variance of the corresponding random effect. The proportion of genetic variance explained by the candidate region was estimated by formula $\frac{\sigma_{ua1}^2 + \sigma_{ud1}^2}{\sigma_{ua1}^2 + \sigma_{ud1}^2 + \sigma_{ua2}^2 + \sigma_{ud2}^2}$, and the proportion of PVE by the target region was calculated by formula $\frac{\sigma_{ua1}^2 + \sigma_{ud1}^2}{\sigma_{ua1}^2 + \sigma_{ud1}^2 + \sigma_{ua2}^2 + \sigma_{ud2}^2 + \sigma_e^2}$. The equations for constructing the matrices A and D are the same as that shown in the 'Constructing a genomic selection model for hybrid breeding' section.

## The definition of 'breeding-favorable genotype'

For loci with definite QTNs reported previously, the definition of breeding-favorable genotype was in accord with previous report[42,80]. With respect to others, the genotype effects of leading SNP in significant association signal were used to define the breeding-favorable genotype. If a locus associated with multiple phenotypes, only the leading SNP in the association signal with the largest $-\log_{10}(P)$ value was selected for analysis. Genotyping was conducted by the leading SNP, and the average phenotypes of both homozygous genotypes were calculated to help determine the breeding-favorable genotype. For grain yield-related traits, the genotype associated with higher grain yield was considered to be the breeding-favorable genotype. For morphological traits, the genotype associated with larger leaf size or tiller angle (erect plant architecture) was defined as the breeding-favorable genotype. With respect to heading date, short heading date resulted in a yield penalty but long heading date was impractical for agricultural production in most rice planting areas; thus, an appropriate heading date was chosen for hybrid breeding. According to the performance of hybrid cultivars in our collection, almost all accessions completed their life cycle when planting in Hangzhou, and a slightly shortened heading date was observed in Y3 hybrids. Thus, the genotype with

shorter heading date was defined as the breeding-favorable genotype in our work. Definition of the breeding-favorable genotype for grain quality referred to the Chinese national standard (NY/T 593–2013 'cooking rice variety quality'). According to the standard, the value of chalkiness and grain translucency (level) were lower for *O. sativa* varieties with better grain quality, and the value of gel consistency was higher for *O. sativa* varieties with better grain quality. Thus, the genotype leading to lower chalkiness, lower grain translucency (level) or higher gel consistency was deemed as the breeding-favorable genotype in our work. With respect to amylose content, the value range was 13.0–18.0 for first-class *O. sativa indica* and *japonica* rice varieties, was 13.0–20.0 and 13.0–19.0 for second-class *O. sativa indica* and *japonica* rice varieties, and was 13.0–22.0 and 13.0–20.0 for third-class *O. sativa indica* and *japonica* rice varieties, respectively. The value range was narrower for first-class varieties, with a lower upper limit. Thus, for convenience, the genotype associated with lower amylose content within limit (13.0–22.0) was determined as the breeding-favorable genotype.

### Constructing a genomic selection model for hybrid breeding

**Training a model based on the GBLUP method to predict breeding value.** We used the GBLUP method in R package sommer to train the model. The basic equation for our model was as follows:

$$\mathbf{y} = X\boldsymbol{\beta} + Z_1\mathbf{u_a} + Z_2\mathbf{u_d} + \boldsymbol{\epsilon}$$

where **y** is a vector of the phenotypic value, **β** is the vector of fixed effects, $\mathbf{u_a}$ and $\mathbf{u_d}$ are vectors for additive and dominant effects, X, $Z_1$ and $Z_2$ are incidence matrices for fixed and random effects, and **ε** is the residual error. The additive relationship matrix A was constructed with the A.mat() function in sommer, and markers were coded −1, 0 and 1 with respect to the homozygous reference genotype, heterozygous genotype and homozygous alternative genotype, respectively. The dominance relationship matrix D was calculated with the D.mat() function in sommer, and markers were coded as 0 and 1 for homozygotes and heterozygote respectively. The equations for the additive relationship matrix A and the dominance relationship matrix D are as follows:

$$A = \frac{WW'}{2\sum_{j=1}^{m} p_j(1-p_j)}$$

$$D = \frac{HH'}{2\sum_{j=1}^{m} p_j(1-p_j)}$$

with *W* and *H* being the scaled marker matrix for A and D, respectively, and *p* and *q* being the allelic frequencies for the *j*th marker ($j = 1…m$).

We used sixfold cross-validation to test the accuracy of the model. The accuracy was evaluated through calculating the Pearson correlation coefficient between the GEBV and observed value. Three iterations were performed and averaged to determine the final accuracy.

**Defining and calculating the selection index.** All individuals were sorted by their GEBV, and a high score was given to individuals in the best rank. The score of seven traits was comprehensively incorporated by calculating the selection index. For yield per plant, seed setting rate, full grain number per panicle and valid panicle number, the following percentile GEBV ranges and associated scores were respectively assigned: >80%: 40; 60–80%: 30; 40–60%: 20; 20–40%: 10; <20%: 0. With respect to the rest of three traits, the optimal value ranges were 80–90 days for heading date, 115–125 cm for plant height, and 3.0–3.4 for grain shape. A score of 40 was assigned to the GEBV falling within the optimal range, while a score of 20 was allocated to the GEBV falling outside the optimal range.

**Validating the genomic selection model in a new population.** A validation population was derived from a random combination of 58 inbred lines and 19 commercial male sterile lines. Among all possible combinations, we randomly selected 67 combinations and planted them in the field conditions of Shanghai, China in the summer of 2021. On the basis of the selection index, 13 and 10 individuals were scored in the top 20% and bottom 20%, respectively. We investigated grain yield-related traits (yield per plant, valid panicle number, full grain number per panicle and seed setting rate). Two-sided Wilcoxon test was conducted to discern significant differences between high-scoring and low-scoring subsets.

### Statistics and reproducibility

No statistical method was used to predetermine sample size. No data were excluded from the phenotypic comparison analysis. Considering the TASSEL software was sensitive to outliers, the outliers were identified by 'boxplot(data$phenotype, plot = FALSE)$out' script in R-4.1.0 and removed for yield per plant, full grain number per plant, seed setting rate, valid panicle number and chalkiness in *indica*–*indica* hybrids, to better identify loci associated with phenotypic variation in *indica*–*indica* rice hybrids. The field work and phenotypic investigation of randomly selected 67 pseudo-combinations out of all 1,102 combinations were parallel to the model construction and selection index calculation, and the analysis was under double-blind experimental control.

The function of 'aov()' and 'Kruskal.test()' in R 4.1.0 were used to identify significant phenotypic change ($P \le 0.05$) for data with distributions conforming to the assumption of homoscedasticity and data with a heteroscedastic distribution. 'LSD.test' in R package agricolae (version 1.3–5) and 'kwAllPairsNemenyiTest()' in R package PMCMRplus (version 1.9.6) with parameter of 'dist = Chisquare' were further used to determine multiple comparison in data with distributions of homoscedasticity and heteroscedasticity, respectively. Box plot was used to demonstrate the distribution of data. For each box plot, the upper and lower boundaries represent the 25th and 75th percentile, respectively; the middle horizontal lines represent the median; the whiskers represent 1.5× the interquartile range; and the dots beyond the whiskers represent outliers.

### Reporting summary

Further information on research design is available in the Nature Portfolio Reporting Summary linked to this article.

## Data availability

All data supporting the findings reported here are available in the paper and Supplementary Information. The raw DNA sequencing data of the 2,839 rice hybrid genomes used in this study are deposited in the European Nucleotide Archive (ENA) under study accession number PRJEB53225. The publicly available website, incorporating resource applied in this study, is at http://ricehybridresource.cemps.ac.cn/#/.

## Code availability

Custom scripts and codes used in this study are provided in the GitHub repository (https://github.com/zlguu/Rice_Heterosis) and Zenodo (https://doi.org/10.5281/zenodo.8195098) (ref. 81).

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

## Acknowledgements

We thank the China National Rice Research Institute and Win-All High Tech Seed Co., Ltd. for providing publicly available rice male sterile resource and rice hybrid germplasm. We also J. Yang from Westlake University for valuable discussion and suggestions, and appreciate D. Sanders from John Innes Centre for his precious suggestions and great effort in polishing the manuscript. This work was supported by the grants from the National Natural Science Foundation of China (31788103), Chinese Academy of Sciences (XDB27010301), Shanghai Municipal Commission of Science and Technology (18JC1415000) to B.H., National Natural Science Foundation of China (32072049) to J.G., and the China Postdoctoral Science Foundation (2021M703213) to Z.G. The funders had no role in study design, data collection and analysis, decision to publish or preparation of the manuscript.

## Author contributions

B.H. conceived and designed the project. X.H. and Z.G. participated in the whole experiment design and data analysis. Z.G., Z.Z., Y.Z. and C.W. performed data analysis and visualization. J.G. and S.Y. collected experimental samples and were responsible for phenotyping and resource preservation. J.G., Z.L., S.Y., Q. Zhan and A.W. performed field management and phenotyping. Q.F., C.Z., Q.T., D.F. and Y.L. performed sampling, library construction and sequencing. T.H., L.Z. and Q. Zhao were responsible for the dataset management. Z.G. and B.H. wrote the article.

## Competing interests

The authors declare no competing interests.

## Additional information

**Extended data** is available for this paper at https://doi.org/10.1038/s41588-023-01495-8.

**Correspondence and requests for materials** should be addressed to Shihua Yang or Bin Han.

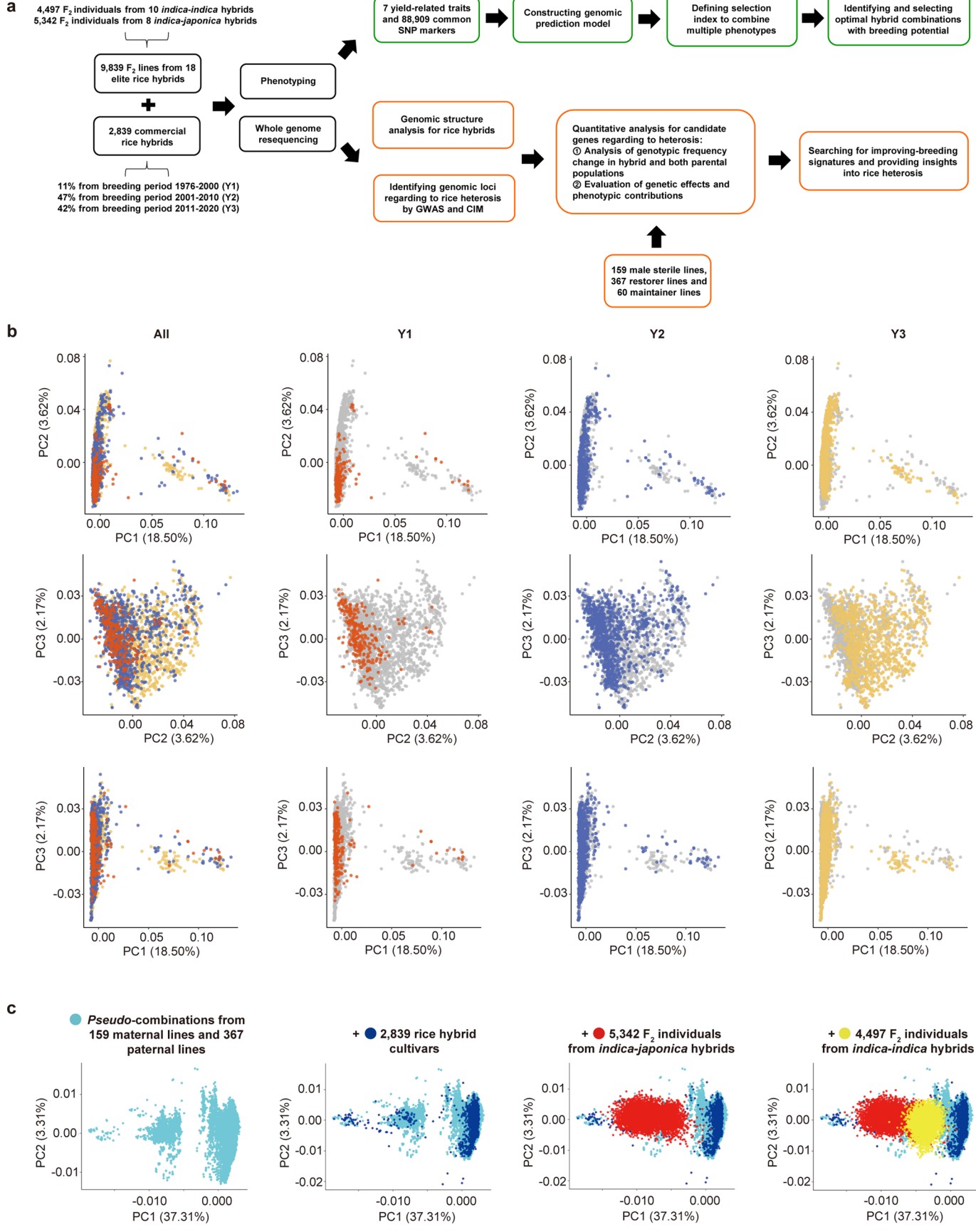

**Extended Data Fig. 1 | See next page for caption.**

**Extended Data Fig. 1 | The analysis procedure and materials used in the study.** **a**, The experimental design and analysis procedure used in this study. **b**, Principal component analysis (PCA) for 2,839 rice hybrids used in this study. The first three PCs were used for plotting. Red, dark blue and yellow dots represent individuals from Y1, Y2 and Y3 breeding periods, respectively. For better demonstration, hybrids from three periods were separately highlighted in three panels. **c**, PCA plot for all materials used for genomic selection model construction. The first two PCs were used for plotting. Cyan-blue, blue, read and yellow dots represent 58,353 *pseudo* combinations, 2,839 cultivar rice hybrids, 5,342 segregating individuals from *indica-japonica* hybrids, and 4,497 segregating individuals from *indica-indica* hybrids. They were successively added to the plots to clearly demonstrate their distribution.

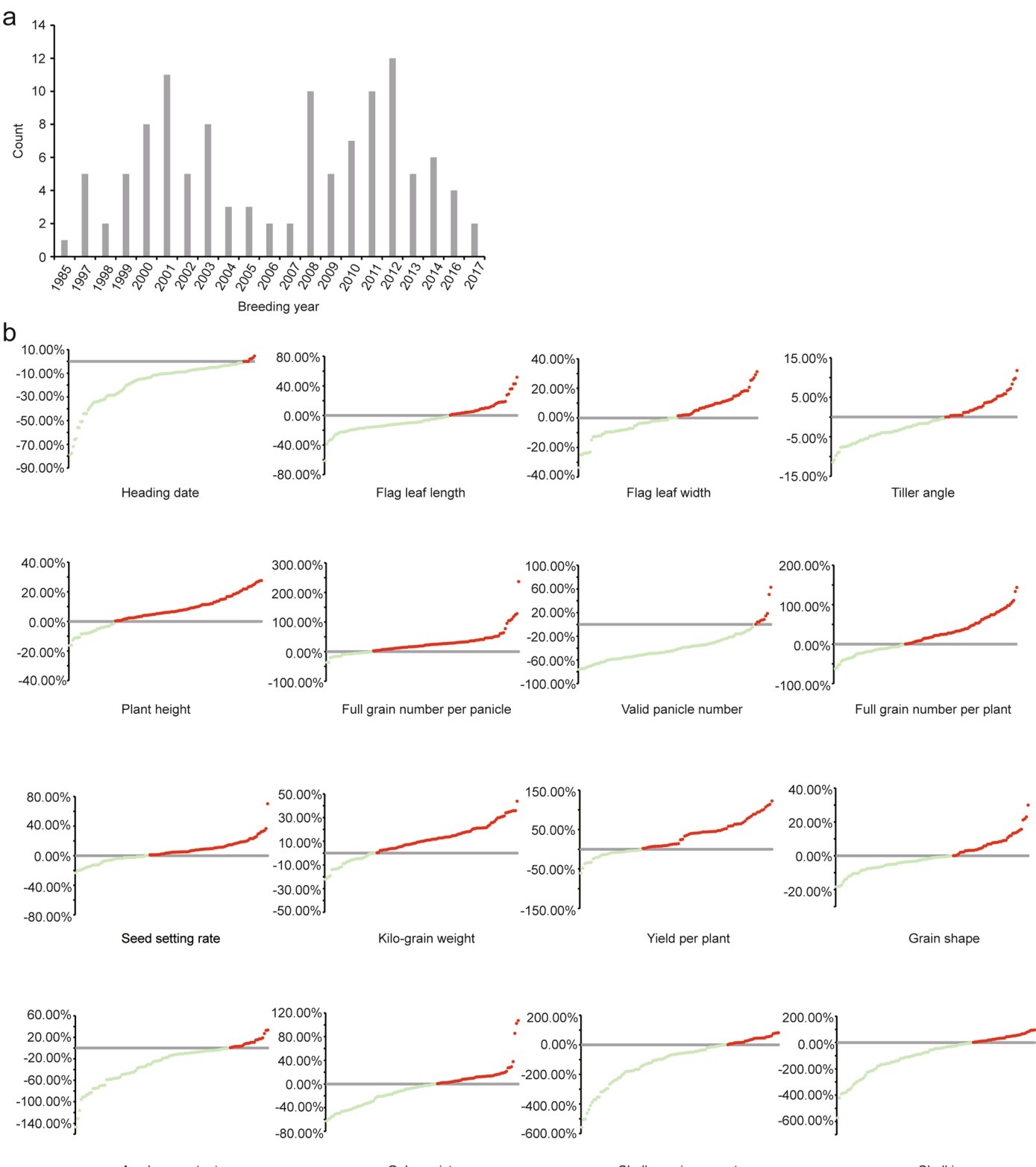

**Extended Data Fig. 2 | Better parent heterosis (BPH) analysis in 123 parent-hybrid trios for heading date, morphological characteristics, grain yield-related and grain quality-related traits.** There were 123 cultivar rice hybrids with both parental lines available, and the 123 parent-hybrid trios were planted in the same conditions and their phenotypes were investigated according to the same standard. **a**, The breeding year of the 123 hybrids. **b**, The BPH value of 123 trios. The index of BPH was estimated as $(F_1-P_1)/P_1$, where $F_1$ and $P_1$ were phenotypic measurements of the hybrid and its parental line with better performance. Because the female lines are sterile, it is difficult to investigate the grain yield-related and quality-related traits. Furthermore, male parents generally have better performance than female parents. Thus, BPH mostly refers to the advantage over male parents. Taking above two points into consideration, the measurements of grain yield-related and quality-related traits from restorer lines were applied to estimate the index of BPH. Red dots represent trios with BPH > 0, and greed dots are trios with BPH ≤ 0.

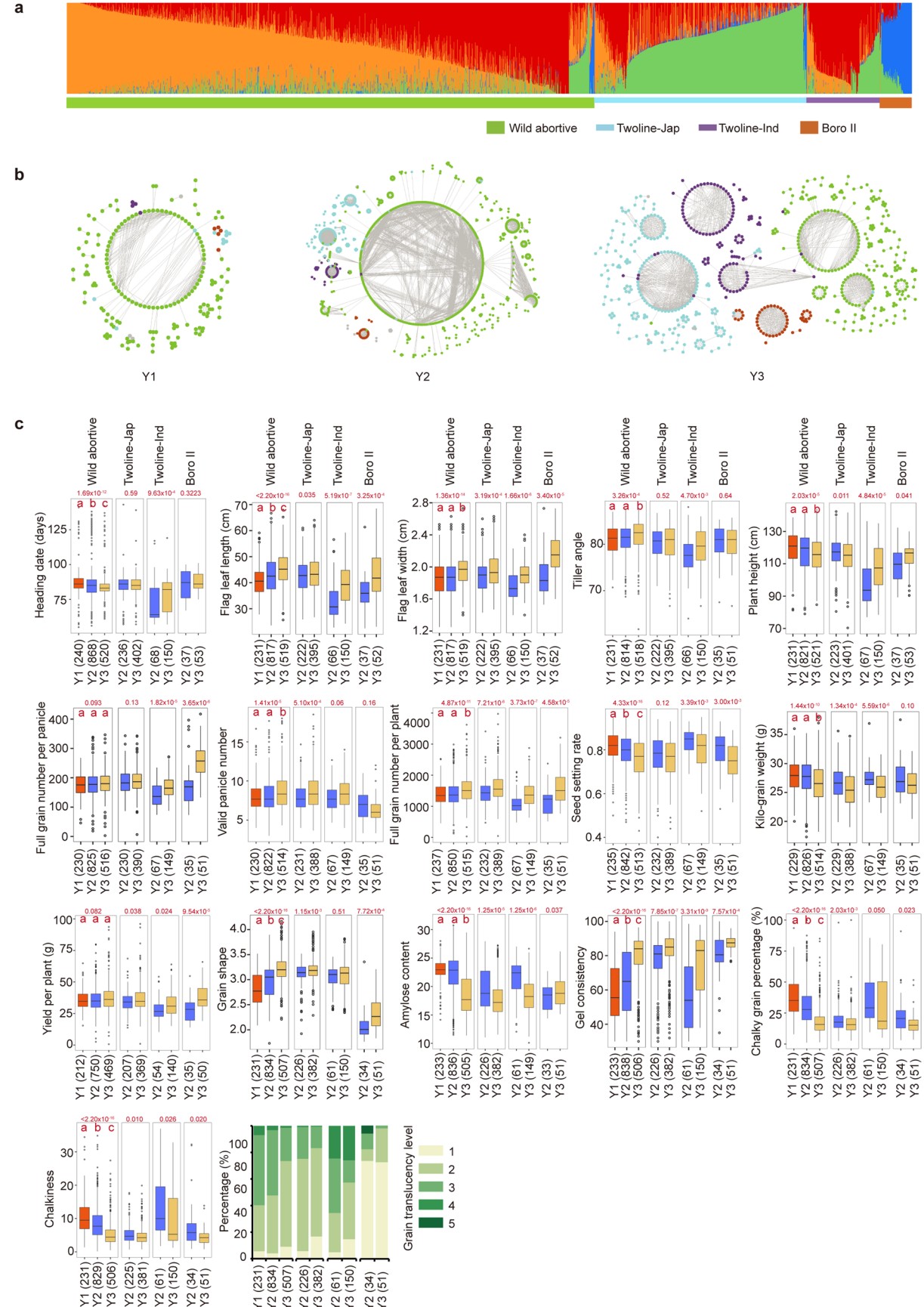

**Extended Data Fig. 3 | See next page for caption.**

**Extended Data Fig. 3 | Population structure analysis and phenotypic change within four subgroups during rice hybrid improvement breeding.** **a**, Admixture analysis with K = 4. Samples were clustered according to their cytoplasm types. The bar on the bottom represents cytoplasm types. The half-height bar represents cytoplasm types from two-line breeding systems, comprising Twoline-Jap and Twoline-Ind organelle types. The full-height bar represents cytoplasm types from three-line breeding system, comprising Wild abortive and Boro II organelle types. **b**, Kinship relationships for hybrids within three breeding periods. A circle indicats a hybrid accession, which is colored according to cytoplasmic type. Pairs of accessions sharing close relationship (kinship coefficients≥0.9) were connected by beelines. **c**, Phenotypic change of hybrids within four subgroups during improvement breeding. Sample size is shown in parenthesis. For WA subgroups, significant test was conducted by one-way ANOVA for data with homoscedasticity distribution or Kruskal-Wallis test for data with heteroscedasticity distribution. And multiple comparisons were further conducted using the least significant difference (LSD) method with 'Bonferroni' correction for homoscedasticity distribution or the Nemenyi test for heteroscedasticity distribution. Different lowercase letters above the boxplots indicate significant phenotype difference ($p \leq 0.05$). For TJ, TI and BT subgroups, significant test was conducted by two-sided t-test for data with homoscedasticity distribution or two-sided wilcoxon test for data with heteroscedasticity distribution.

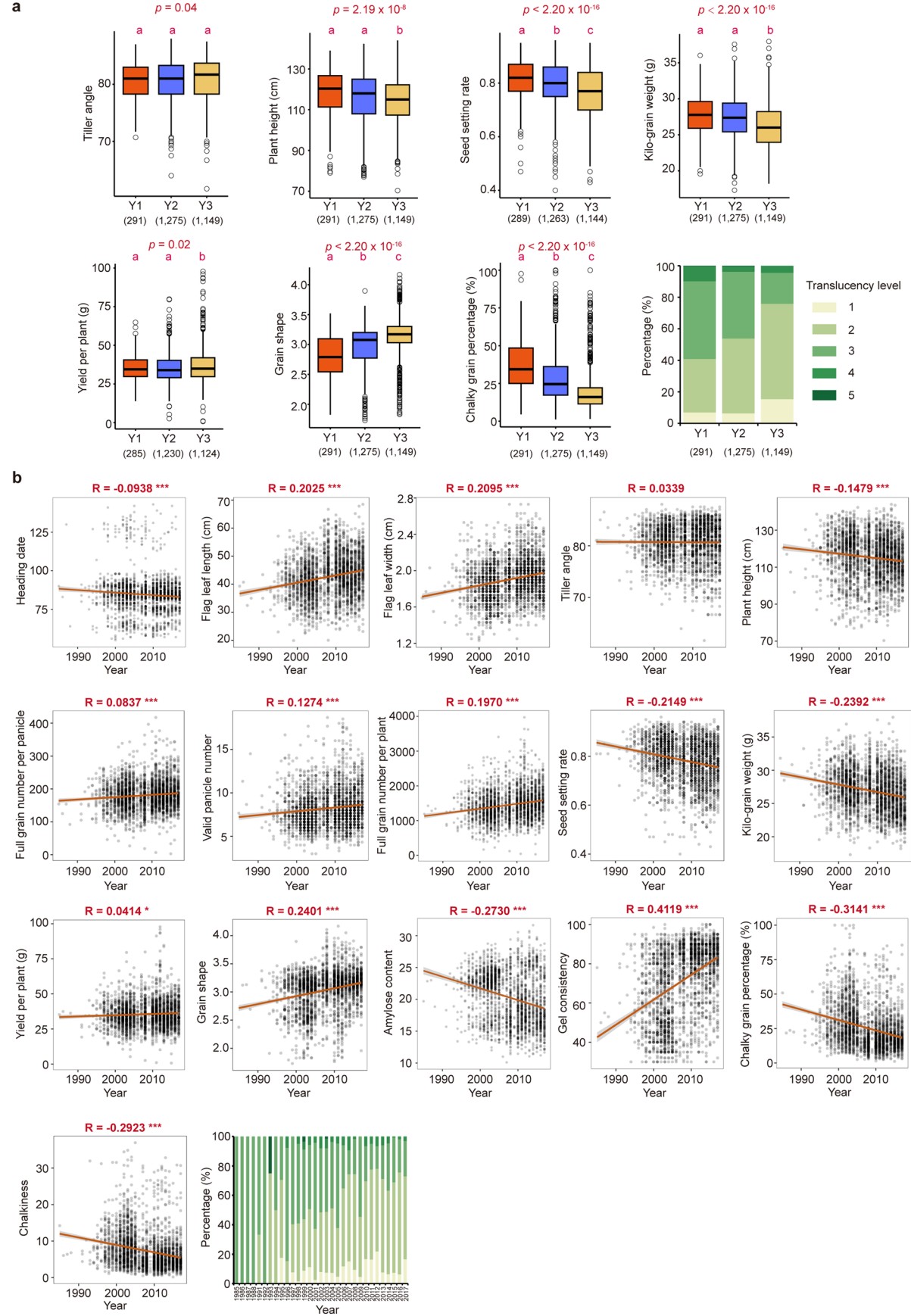

**Extended Data Fig. 4 | See next page for caption.**

**Extended Data Fig. 4 | Phenotypic change during rice hybrid improvement breeding. a**, Phenotypic comparison of hybrids from three breeding periods. Sample size is shown in parenthesis. Significant test was conducted by one-way ANOVA for data with homoscedasticity distribution or Kruskal-Wallis test for data with heteroscedasticity distribution. And multiple comparison was further conducted by the least significant difference (LSD) method with 'Bonferroni' correction for homoscedasticity distribution or Nemenyi test for heteroscedasticity distribution. Different lowercase letters above the boxplots indicates significant phenotype difference ($p \leq 0.05$). **b**, Correlation analysis between phenotype and breeding years of the hybrids. Correlation coefficient and corresponding $p$ value were estimated by cor.test() function in software R. '*': $0.01 < p \leq 0.05$; '**': $0.001 < p \leq 0.01$; '***': $p \leq 0.001$.

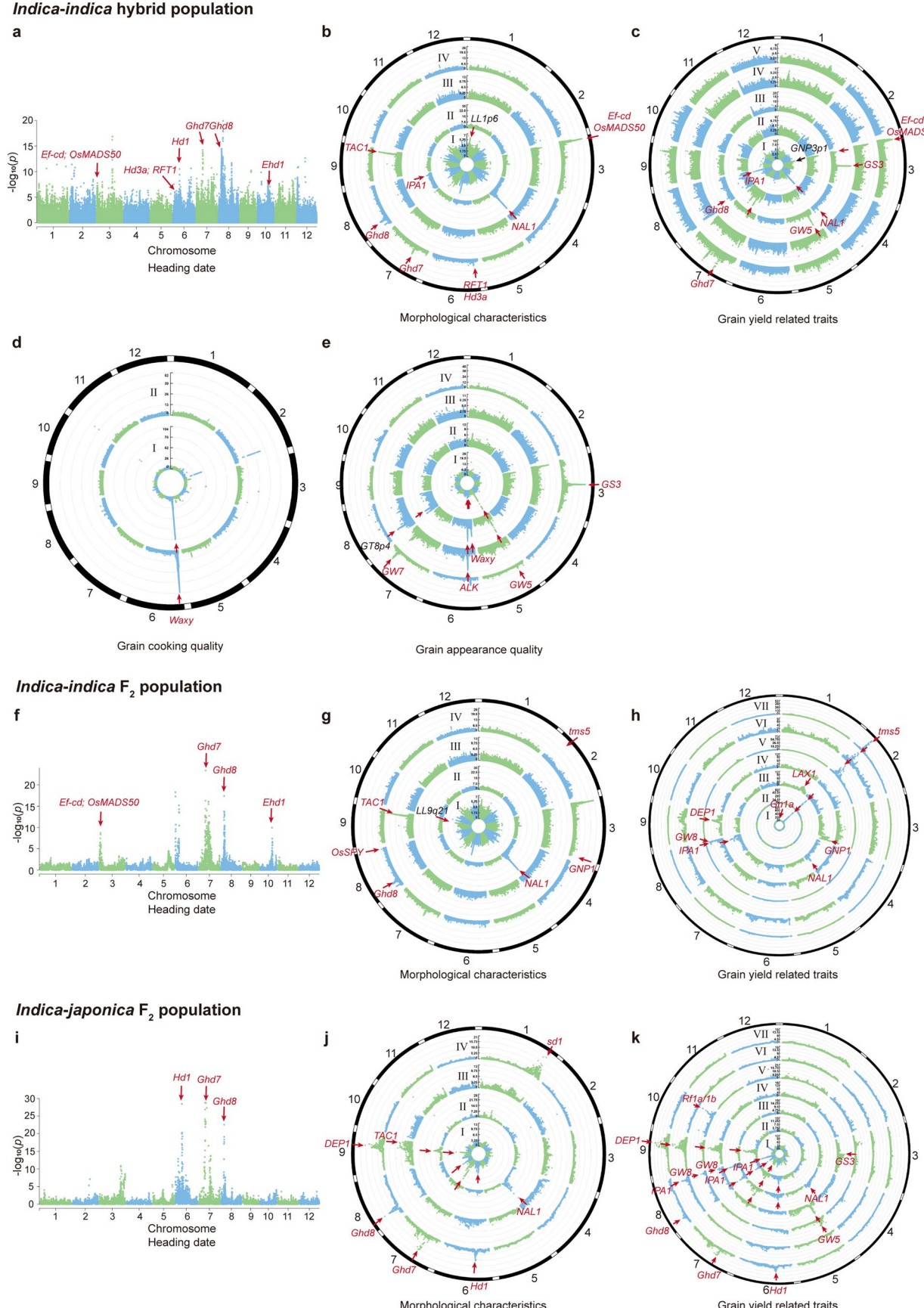

**Extended Data Fig. 5 | See next page for caption.**

**Extended Data Fig. 5 | Genome-wide association analysis (GWAS) of important agronomic traits in *indica-indica* hybrid population and segregating populations. a-e**, GWAS results of key agronomical traits in *indica-indica* hybrid population. (a) Physiological feature (heading date). (b) Morphological traits: I, leaf length; II, leaf width; III, tiller angle; IV, plant height. (c) Grain yield-related traits: I, full grain number per panicle; II, full grain number per plant; III, kilo-grain weight; IV, seed setting rate; V, yield per plant. (d) Grain cooking quality: I, amylose content; II, gel consistency. (e) Grain appearance quality: I, chalky grain percentage; II, chalkiness; III, grain translucency level; IV, grain shape (length to width ratio of polished grain). **f-h**, GWAS results of key agronomical traits in *indica-indica* segregating population. **I-k**, GWAS results of key agronomical traits in *indica-japonica* segregating population. (f and i) Physiological feature (heading date). (g and j) Morphological traits: I, leaf length; II, leaf width; III, tiller angle; IV, plant height. (h and k) Grain yield-related traits: I, full grain number per panicle; II, full grain number per plant; III, total grain number per panicle; IV, kilo-grain weight; V, seed-setting rate; VI, valid panicle number; VII, yield per plant. Negative $\log_{10}(p)$ values (Y-axis) were plotted against SNP positions (X-axis) on each of 12 chromosomes. Association signals with the known genes located nearby were marked in red, and newly identified loci discussed in Fig. 3 were marked in black.

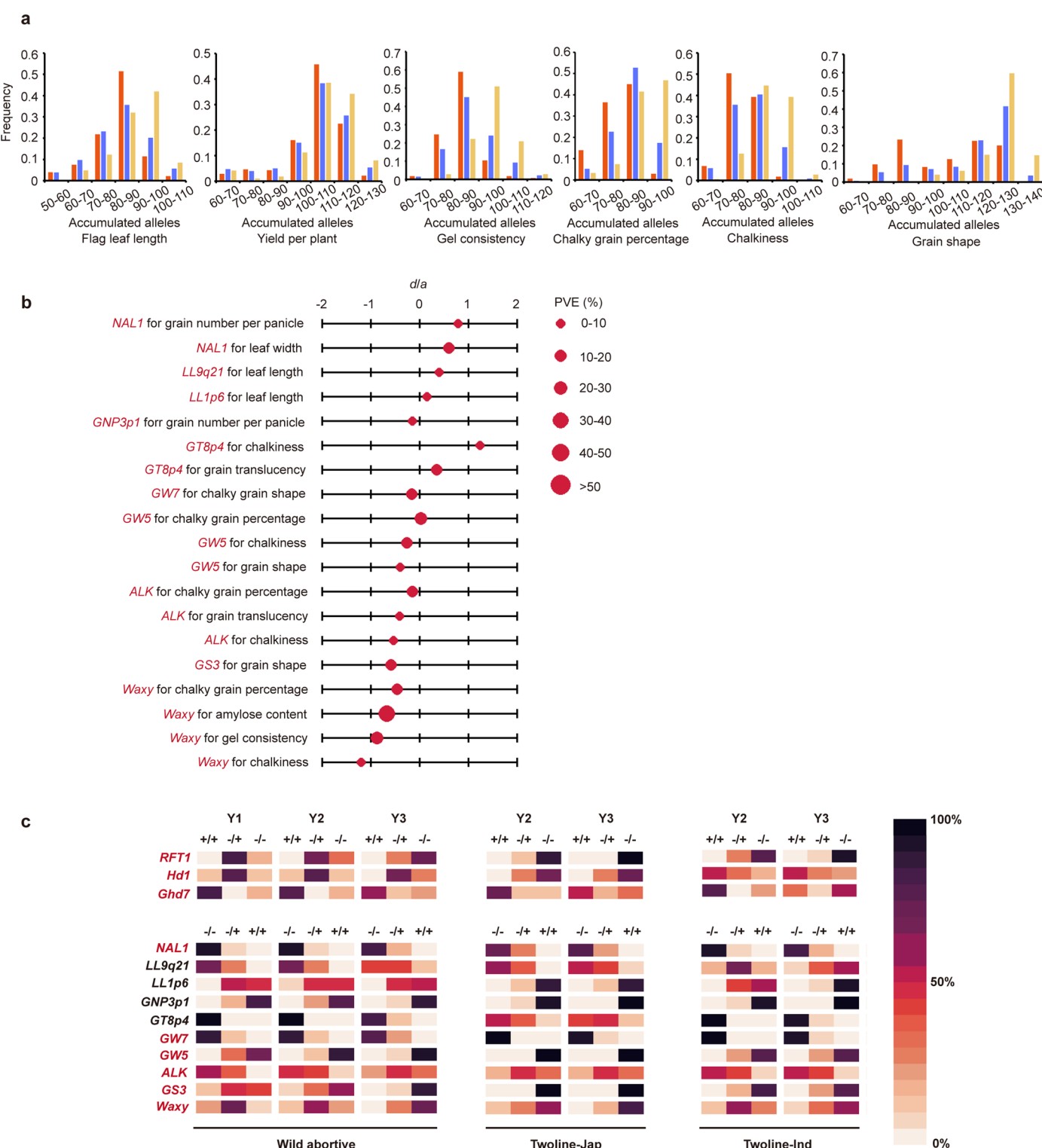

**Extended Data Fig. 6 | Identification and quantitative analysis of breeding signatures based on genome-wide association analysis. a**, The distribution of the count of pyramiding breeding-favorable alleles in *indica-indica* hybrid from three breeding periods. The top-100 GWAS signals in *indica-indica* rice hybrid population were applied for analysis. **b**, The *d/a* index (dominance-effect/additive-effect) and PVE (phenotypic variance explained) for candidate genes involved in improvement breeding. **c**, Allelic frequency change for candidate genes involved in improvement breeding within three subgroups of *indica-indica* hybrids. Because most BT hybrids were *indica-japonica* or *japonica-japonica* hybrids, they were not discussed here. For morphological characteristics, grain yield-related and quality-related traits, symbol '+' denotes breeding-favorable alleles and '-' indicates breeding-unfavorable alleles. As for heading date, symbol '+' represents alleles associated with longer heading date and '-' indicates alleles with shorter heading date.

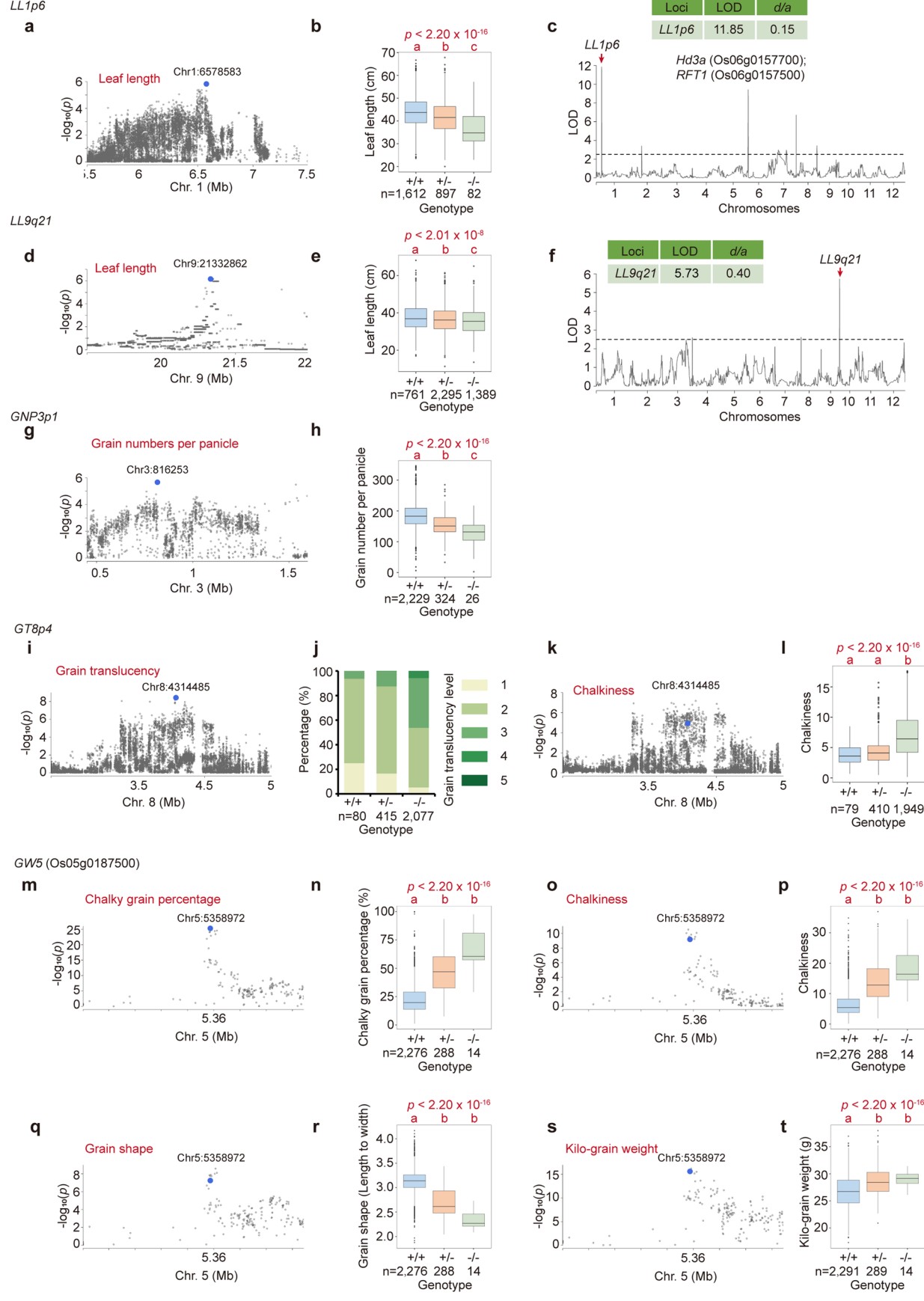

**Extended Data Fig. 7 | See next page for caption.**

**Extended Data Fig. 7 | Identification of candidate genes involved in improvement breeding. a-c**, *LL1p6*. **d-f**, *LL9q21*. **g-h**, *GNP3p1*. **i-l**, *GT8p4*. **m-t**, *GW5*. (a, d, g, i, k, m, o, q, s) Local manhattan plots surrounding the genomic regions where the candidate loci are located. (b, e, h, j, l, n, p, r, t) boxplots or barplot demonstrating the phenotypic observation distribution of 3 genotypes. Peak association signals marked in blue dots were used for genotyping. n = sample size. Significant test was conducted by one-way ANOVA for data with homoscedasticity distribution or Kruskal-Wallis test for data with heteroscedasticity distribution. And multiple comparison was further conducted by the least significant difference (LSD) method with 'Bonferroni' correction for homoscedasticity distribution or Nemenyi test for heteroscedasticity distribution. Different lowercase letters above the boxplots indicate significant phenotype difference ($p \leq 0.05$). (c and f) Two newly-identified loci in GWAS were also identified by the linkage mapping of leaf length in two sets of segregating populations, respectively. *LL1p6* was identified in a $F_2$ population from the *indica-indica* hybrid Longjingyou534 and *LL9q21* was from the *indica-indica* hybrid Quanliangyou2118. LOD (likelihood of odds) values were plotted against the physical positions. The threshold was set as 2.5, and was indicated by a dashed line.

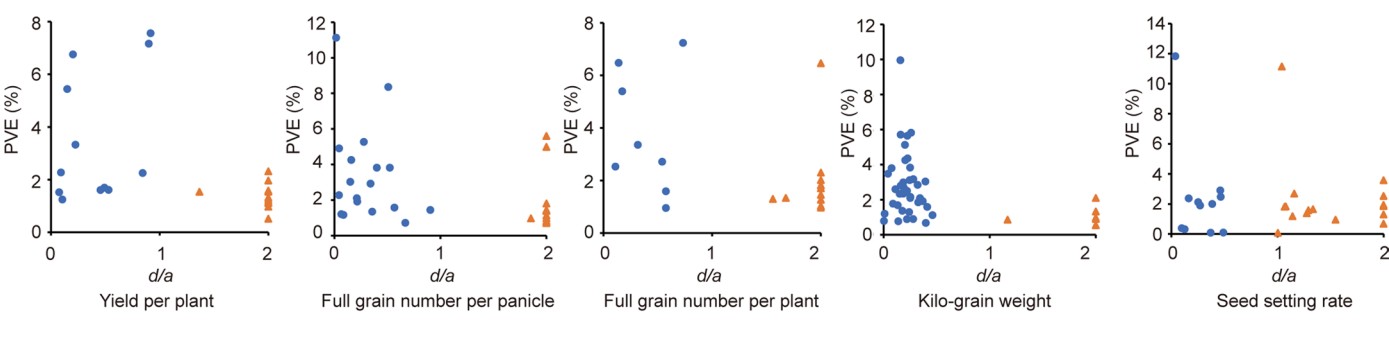

Loci with positive partial dominance effect

Loci with over-dominance or *pesudo* over-dominance effect

**Extended Data Fig. 8 | Evaluating the contribution of dominance and over-dominance effects QTLs identified in *indica-japonica* segregating populations.** In rice hybrids, male parents generally have much better yield performance than female parents, so better parent heterosis mostly refers to the advantage over male parents. Therefore, the over-parent heterosis observed in hybrids is predominantly attributable to the influence of positive partial dominance effect loci contributed by female parents, as well as over-dominance effect loci from both parental lines. Thus, female-contributed QTLs that displayed positive dominance effects and the QTLs that demonstrated over-dominance effects were included for analysis. The *d/a* index is applied to measure the magnitude of dominance effect. *d/a* ≤ 1 and >1 refer to positive partial dominance (blue) and over dominance effect (orange), respectively.

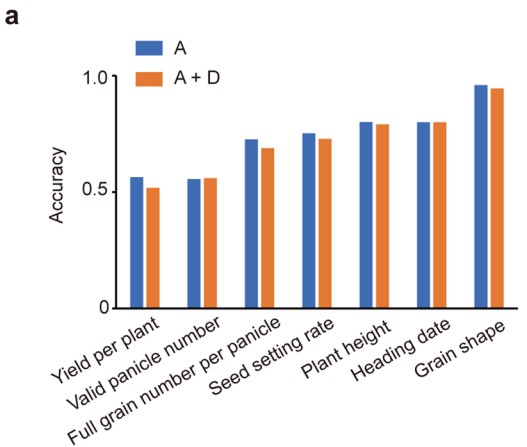

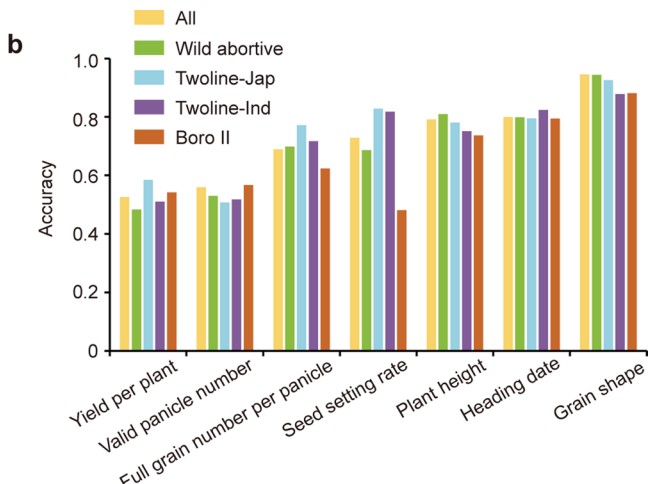

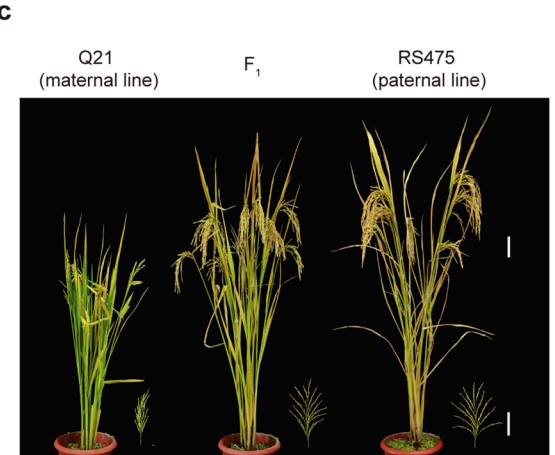

**Extended Data Fig. 9 | The prediction accuracy of genomic selection model and quick selection of a candidate combination by the model. a**, Comparison of prediction accuracy of additive (A) model and additive plus dominance (A + D) model for seven agronomical traits. **b**, Prediction accuracy of A + D model for seven agronomic traits across all mixed samples and within four subgroups. Among all hybrids supplied for model construction, 62.05%, 24.96%, 8.61% and 3.74% were WA, TJ, TI and BT hybrids (the remaining samples were HL hybrids

or accessions with unknown identity). Furthermore, among all the 9,839 $F_2$ individuals, 21.27%, 16.14%, 14.61% and 47.98% were from WA, TJ, TI and BT hybrids. **c**, A combination with high selection score. Scale bar: 10 cm; upper for the whole plant and lower for the panicle. **d**, Genome-wide scanning for QTNs controlling key agronomical traits. The breeding-favorable alleles contributed by maternal and paternal lines were respectively listed, and the breeding-favorable alleles deficient in both parental lines were also listed.

**Extended Data Table 1 | The size of 18 F$_2$ populations from 18 elite rice hybrids and the number of recombination bins identified in each F$_2$ population**

| Ecotype[a] | F$_2$ Population | Size | Recombination bins |
|---|---|---|---|
| *Indica-Indica* | Pop1 | 498 | 3,093 |
| | Pop2 | 340 | 2,716 |
| | Pop3 | 452 | 3,072 |
| | Pop4 | 357 | 2,665 |
| | Pop5 | 357 | 2,676 |
| | Pop6 | 399 | 2,916 |
| | Pop7 | 365 | 2,959 |
| | Pop8 | 725 | 3,155 |
| | Pop9 | 418 | 3,157 |
| | Pop10 | 586 | 2,955 |
| *Indica-Japonica* | Pop11 | 621 | 3,017 |
| | Pop12 | 949 | 3,235 |
| | Pop13 | 529 | 2,964 |
| | Pop14 | 690 | 3,131 |
| | Pop15 | 808 | 3,327 |
| | Pop16 | 659 | 3,329 |
| | Pop17 | 465 | 3,023 |
| | Pop18 | 621 | 3,202 |

a: The ecotypes for F$_{1s}$ used to generate F$_2$ populations. "*Indica-indica*" indicated *indica* intrasubspecific cross; "*Indica-japonica*" represented intersubspecific cross.

| | |
|---|---|
| | Bin Han |

# Reporting Summary

## Statistics

For all statistical analyses, confirm that the following items are present in the figure legend, table legend, main text, or Methods section.

| n/a | Confirmed | |
|---|---|---|
| ☐ | ☒ | The exact sample size (*n*) for each experimental group/condition, given as a discrete number and unit of measurement |
| ☐ | ☒ | A statement on whether measurements were taken from distinct samples or whether the same sample was measured repeatedly |
| ☐ | ☒ | The statistical test(s) used AND whether they are one- or two-sided *Only common tests should be described solely by name; describe more complex techniques in the Methods section.* |
| ☐ | ☒ | A description of all covariates tested |
| ☐ | ☒ | A description of any assumptions or corrections, such as tests of normality and adjustment for multiple comparisons |
| ☐ | ☒ | A full description of the statistical parameters including central tendency (e.g. means) or other basic estimates (e.g. regression coefficient) AND variation (e.g. standard deviation) or associated estimates of uncertainty (e.g. confidence intervals) |
| ☐ | ☒ | For null hypothesis testing, the test statistic (e.g. $F$, $t$, $r$) with confidence intervals, effect sizes, degrees of freedom and $P$ value noted *Give P values as exact values whenever suitable.* |
| ☒ | ☐ | For Bayesian analysis, information on the choice of priors and Markov chain Monte Carlo settings |
| ☒ | ☐ | For hierarchical and complex designs, identification of the appropriate level for tests and full reporting of outcomes |
| ☐ | ☒ | Estimates of effect sizes (e.g. Cohen's *d*, Pearson's *r*), indicating how they were calculated |

*Our web collection on statistics for biologists contains articles on many of the points above.*

## Software and code

Policy information about availability of computer code

| Data collection | No software was used to collect data. |
|---|---|
| Data analysis | The details have been described in Method section.<br>1. Reads quality control was performed using Trimmomatic (version 0.38) with parameters 'ILLUMINACLIP:TruSeq3-PE.fa:2:30:10:2:true MAXINFO:50:0.6'. The clean reads were mapped against the rice genome IRGSP1.0 by BWA (version 0.7.1). Variation was detected by GATK (version 4.1.4.1).<br><br>2. Genome-wide nucleotide diversity was calculated using VCFtools (version 0.1.15) with 200kb sliding window. Kinship coefficient was calculated using EMMAX (version emmaxbeta-07Mar2010) and visualized using Cytoscape (version 3.8.2). Four-fold degenerate (4DTv) sites were identified by SnpEff (version 4.3t). PCA was performed by GCTA (version 1.93.2 beta). Ancestral components for hybrids were inferred by the ADMIXTURE program (version 1.3.0) and visualized using the R package pophelper (version 2.3.1)<br><br>3. GWAS was performed by the mixed linear model in the TASSEL software package (version 5.0 Standalone). The high-quality SNP data were further filtered by software PLINK (v.1.90b6.12 64-bit) to keep variants with a missing rate ≤10% and minor allele frequency ≥5%. Principal component analysis was performed using the input genetic markers by the software GCTA (version 1.93.2 beta), and the first two principal components were incorporated as the covariates to effectively account for population structure. Kinship matrix was generated based on the input genetic markers by "Kinship" function in TASSEL.<br>The dominance-effect/additive-effect (d/a) for association signal was calculated based on genotype effect estimated by TASSEL. The genotype effects of peak SNP in target association signal were chosen to calculate d/a index:<br>a=\|A-C\|/2<br>d=M-(A+C)/2 |

Furthermore, for QTLs mapping by F2 population, the index of d/a was estimated by IciMapping software (version 4.2.53).

4. The PVE by the candidate region surrounding the association signal was estimated according to previously report. A mixed linear model with multiple random effect was applied to estimate the variance components using the R package sommer (version 4.2.0.1).

5. Training a model based on the GBLUP method to predict breeding value. We used the GBLUP method in R package sommer to train the model.

6. Custom scripts and codes used in this study are provided in the GitHub repository (https://github.com/zlguu/Rice_Heterosis).

For manuscripts utilizing custom algorithms or software that are central to the research but not yet described in published literature, software must be made available to editors and reviewers. We strongly encourage code deposition in a community repository (e.g. GitHub). See the Nature Portfolio guidelines for submitting code & software for further information.

# Data

Policy information about availability of data

All manuscripts must include a data availability statement. This statement should provide the following information, where applicable:

- Accession codes, unique identifiers, or web links for publicly available datasets
- A description of any restrictions on data availability
- For clinical datasets or third party data, please ensure that the statement adheres to our policy

Data availability statement is provided in Page 29, Lines 856-861:
All data supporting the findings reported here are available in the paper and supplementary files.
1. The raw DNA sequencing data of the 2,839 rice hybrid genomes used in this study are deposited in the NCBI Sequence Read Archive under study accession no. PRJEB53225.
2. Sample information and phenotype are provided as the Supplementary Information.
3. The publicly available website, incorporating resource applied in this study, is at http://ricehybridresource.cemps.ac.cn/#/.

# Human research participants

Policy information about studies involving human research participants and Sex and Gender in Research.

| Reporting on sex and gender | not involved |
|---|---|
| Population characteristics | not involved |
| Recruitment | not involved |
| Ethics oversight | not involved |

Note that full information on the approval of the study protocol must also be provided in the manuscript.

# Field-specific reporting

Please select the one below that is the best fit for your research. If you are not sure, read the appropriate sections before making your selection.

☒ Life sciences          ☐ Behavioural & social sciences          ☐ Ecological, evolutionary & environmental sciences

For a reference copy of the document with all sections, see nature.com/documents/nr-reporting-summary-flat.pdf

# Life sciences study design

All studies must disclose on these points even when the disclosure is negative.

| Sample size | All 2,839 rice hybrids are from the collections preserved at the China National Rice Research Institute in Hangzhou, China, including 1,495 hybrids rice accessions reported previously. |
|---|---|
| Data exclusions | No data was excluded from the phenotypic comparison analysis. Considering the TASSEL software was sensitive to outliers, the outliers was removed by "boxplot(data$phenotype, plot = FALSE)$out" script in R-4.1.0 for yield per plant, full grain number per plant, seed setting rate, valid panicle number and chalkiness in indica-indica hybrids, to better identify loci associated with phenotypic variation in indica-indica rice hybrids. |
| Replication | For morphological characteristics and yield components and relevant factors, they were investigated with three biological replicates for each sample. The heading date was recorded as the duration in days from the date of sowing to the emergence of first inflorescences above flag leaf sheath of five plants for each accession. With respect to grain quality-related traits, grains from mixed harvest were randomly selected to conduct investigation, with two replicates for each accession. |

| | |
|---|---|
| Randomization | We used all the samples to conduct phenotypic comparison, genomic structure analysis and GWAS (except for the outliers), and randomization was not involved. Furthermore, with respect to model construction, 6-fold cross-validation was used to estimate the accuracy of the model. |
| Blinding | The field work and phenotypic investigation of randomly-selected 67 pseudo-combinations out of all 1,102 combinations were parallel to the model construction and selection index calculation, and the analysis was under double-blind experimental control. |

# Reporting for specific materials, systems and methods

We require information from authors about some types of materials, experimental systems and methods used in many studies. Here, indicate whether each material, system or method listed is relevant to your study. If you are not sure if a list item applies to your research, read the appropriate section before selecting a response.

## Materials & experimental systems

| n/a | Involved in the study |
|---|---|
| ☒ ☐ | Antibodies |
| ☒ ☐ | Eukaryotic cell lines |
| ☒ ☐ | Palaeontology and archaeology |
| ☒ ☐ | Animals and other organisms |
| ☒ ☐ | Clinical data |
| ☒ ☐ | Dual use research of concern |

## Methods

| n/a | Involved in the study |
|---|---|
| ☒ ☐ | ChIP-seq |
| ☒ ☐ | Flow cytometry |
| ☒ ☐ | MRI-based neuroimaging |

