## [Peer Review File · Nature Genetics]

Peer Review Information

Manuscript Title: Structure and function of rice hybrid genomes reveal genetic basis and optimal performance of heterosis

Corresponding author name(s): Professor Shihua Yang, Professor Bin Han

Reviewer Comments & Decisions:

Decision Letter, initial version:

7th Feb 2023

Dear Professor Han,

Your Article, "Structure and function of the hybrid rice genomes reveal genetic basis and optimal performance of heterosis" has now been seen by 3 referees. You will see from their comments copied below that while they find your work of considerable potential interest, they have raised quite substantial concerns that must be addressed. In light of these comments, we cannot accept the manuscript for publication, but would be very interested in considering a revised version that addresses these serious concerns.

Briefly, the reviewers appreciate the richness of your dataset, but currently differ quite widely on the overall advance presented by your manuscript.

Reviewer #1 is the most positive, saying you present "significant new insights" of potential "great impact". They make a number of suggestions for improvement, primarily further analysis.

Conversely, Referees #2 and #3 - while acknowledging the importance of your dataset - are much less supportive at this stage.

Reviewer #2 states that the overall advance in our understanding of heterosis is lacking, but seems to suggest that further analysis of your rich data could present mechanistic insight that would be a novelty worth of publication.

Reviewer #3 provides an in-depth technical critique of your analytic approach, highlighting a broad range of fundamental issues (e.g. the use of the 3 categorical groups as a basis to analyse time-dependent improvement). However, they also provide productive suggestions to address many of their critiques.

Taken together, we think that a substantial revision and expansion of your study to address both the technical (Referee #3) and novelty (Referee #2) concerns will be required to persuade these referees to support publication. However, given that all three reviews suggest the dataset will be a valuable

resource, we have decided to offer you and your co-authors the opportunity to do so.

We note there are several common criticisms across these reports, e.g. the definition of "superior" alleles is unclear. We would also highlight that the reviews also suggest that the English language presentation could be significantly improved, and we'd recommend that this also be addressed in a revision.

We hope you will find the referees' comments useful as you decide how to proceed. If you wish to submit a substantially revised manuscript, please bear in mind that we will be reluctant to approach the referees again in the absence of major revisions.

To guide the scope of the revisions, the editors discuss the referee reports in detail within the team, including with the chief editor, with a view to identifying key priorities that should be addressed in revision and sometimes overruling referee requests that are deemed beyond the scope of the current study. We hope that you will find the prioritised set of referee points to be useful when revising your study. Please do not hesitate to get in touch if you would like to discuss these issues further.

If you choose to revise your manuscript taking into account all reviewer and editor comments, please highlight all changes in the manuscript text file. At this stage we will need you to upload a copy of the manuscript in MS Word .docx or similar editable format.

*2) If you have not done so already please begin to revise your manuscript so that it conforms to our Article format instructions, available [here](http://www.nature.com/ng/authors/article_types/index.html). Refer also to any guidelines provided in this letter.

[redacted]

If you wish to submit a suitably revised manuscript we would hope to receive it within 6 months. If you cannot send it within this time, please let us know. We will be happy to consider your revision so long as nothing similar has been accepted for publication at Nature Genetics or published elsewhere. Should your manuscript be substantially delayed without notifying us in advance and your article is eventually published, the received date would be that of the revised, not the original, version.

Thank you for the opportunity to review your work.

Sincerely,

Michael Fletcher, PhD
Senior Editor, Nature Genetics

ORCID: 0000-0003-1589-7087

Referee expertise:

Referee #1: crop genetics; heterosis.

Referee #2: crop (including rice) genetics; evolution.

Referee #3: crop (rice) genetics; breeding.

Reviewers' Comments:

Reviewer #1:

Remarks to the Author:

Hybrid rice breeding has contributed significantly to world food security. Although a number of heterotic loci has been identified, nevertheless, the genetic basis and mechanisms of rice heterosis still

remain elusive. In particular, prediction of heterosis and thus genomics-guided hybrid breeding remains a challenging task. In this study, the authors performed a comprehensive genome analysis of 2839 elite rice hybrid varieties released in China over the past 40 years. These hybrids were classified into three groups based on their release time: Y1 (1976-2000); Y2 (2001-2010), and Y3 (2011-2020). Phenotyping analyses revealed significant improvement of three important agronomic traits in the hybrids over the breeding history: heading date, grain yield potential and grain quality. In combination with GWAS analysis of 17 agronomic traits in 2724 indica-indica hybrids and 4497 F2 plants from 10 indica-indica hybrids, 89 putative heterotic loci were identified (with 32 containing known genes). The authors showed that during modern hybrid rice breeding, the genetic diversity of hybrids gradually increased, accompanied by increased introgression of japonica genomic components and superior alleles. They also observed a stacking effect of superior alleles from Y1 to Y3 for multiple agronomic traits (heading date, leaf length, grain number per panicle, and grain quality) and purging of inferior alleles. Interestingly (but expectedly), they also found a high level of genome-wide complementarity in indica-japonica intersubspecific hybrids, including numerous known genes controlling important agronomic traits (such as DEP1, NAL1, Ghd7, Ghd8, GW5 and Hd1). The observation supports the notion that genetic complementarity plays a major role in indica/japonica heterosis. At last, the authors constructed a genomic-selection model for future hybrid rice breeding based on the genomic and phenotypic information of 2839 hybrids and 9839 F2 plants derived from indica-indica and indica-japonica hybrids. The prediction accuracy for a set of traits ranged from 0.518 to 0.945. They further developed a customized selection index to improve the accuracy of the prediction model. They demonstrated the potential utility of such a model for genomic selection breeding programs of hybrid rice. Overall, this study provides significant new insights into rice heterosis and established a prototype genomic selection model for future hybrid rice breeding, and this should generate a great impact on hybrid breeding in rice and other crops as well.

I have a number of questions and comments for further improvement of the manuscript:

1. Line 110: I am not sure what is the meaning of "head rice". Please explain.
2. Line 135-136: "The four main subgroups of hybrids had distinguishing genomic components (Extended Data Fig. 5a)". Could the authors elaborate more on this point—meaning and implications?
3. Fig. 2e-2f, I am not sure the meaning and definition of "incomplete (indica/japonica" and complete (japonica/japonica) states of the WA and TJ hybrids---how they are determined and measured? Based on what? Reasons? And what are the implications? Some explanation is warranted.
4. Line 199-200, could the authors elaborate more on the determination of dominant effect/additive effect (d/a) and its implications (such as for the LL1p6 locus)? How in general such loci contribute to heterosis? How to define the superior allele or inferior allele based on such values and the controlled traits?
5. I am wondering whether the developed prediction model could be applied to different subgroups of hybrids with comparable accuracy? Also, the prediction accuracy varied substantially for different agronomic traits, could the authors elaborate on the possible causes and solutions?
6. I suggest the authors to cite the recent studies by Li et al. (Nature Plants, 2022 Jul;8(7):750-763) and Wang et al. (Nature Genetics, 2023, <https://www.nature.com/articles/s41588-022-01283-w>) on maize heterotic groups and maize heterosis and properly discuss them in the context of the findings made in this work.
7. Careful proofreading is recommended to improve the clarity and logics of this manuscript.

Reviewer #2:

Remarks to the Author:

This paper describes a large experiment to examine the genomic basis for heterosis in hybrid rice. Hybrid rice has been a key driver of agricultural productivity, particularly in China, yet the genomic basis for hybrid vigor remains elusive. This is clearly an issue with most examples of heterosis. In this study, they scan the genomes of a very large collection of hybrid rice, identify genome regions/genes associated with increased vigor and develop a predictive framework for hybrid rice vigor that could be applied to breeding efforts.

First, the scope of this study is large and is fairly comprehensive. With such a large data set, it is one of the most comprehensive genome scans of heterosis I am aware of. Moreover, the ability to provide a predictive tool could be of great importance to breeders in the future.

Nevertheless, there are major issues with the paper that makes me less enthusiastic about it.

1. First is the definition of heterosis. The authors frequently invoke "superior" alleles that appear to go with "superior" phenotypes. But defining superiority can be subjective, and so it is unclear how to think about this in a genetic context. For traits that are directly related to yield – grain number, panicle number, biomass – hybrid vigor is a clear phenomenon since in these cases vigor is equivalent to higher yield or size. But for flowering time, grain cooking quality and so forth, superiority is based in human desires. In those cases, it would be better to discuss these types of traits are over- or underdominance.

2. The molecular genetic basis of heterosis continues to be the central issue of hybrid vigor research. Just what is happening behind this phenomenon? The authors in their Introduction point out that there are several models to try to explain heterosis. A good use of their data would be to set up a strong test of the different models so that we can advance our understanding of the genetic basis of heterosis.

3. This comment has to do with point 2 above – the authors identify several genomic regions associated with heterosis in hybrid rice. Identifying heterotic loci is not new – what would be new is if we could understand how it all works – what are the molecular mechanisms that translate from heterotic loci to vigor. Is it just overdominance – and if so, how? Is it masking deleterious recessives?

The paper as written does make a contribution to our understanding of heterosis but the contribution is incremental and does not really go to the heart of the matter.

A final thought – it would help to have a professional writer go through the text as there are some awkward sentence structures.

Reviewer #3:

Remarks to the Author:

This study performed re-sequencing of 2,839 hybrid rice varieties to elucidate genetic mechanism of hybrid performance in rice. Through nucleotide diversity survey and GWAS, the authors found loci associated with rice hybrid breeding. In addition, the authors applied genomic selection approach to

the rice hybrid varieties and proposed a strategy to contribute to future rice hybrid breeding. Since such a large-scale data are rare and valuable in plant science, the data obtained in this study will contribute not only to hybrid breeding but also to genetic study of hybrids. However, I have concerns about the theoretical validity of the analytical methods and the conclusions drawn from the analyses. Some of the concerns will affect key message of the manuscript (especially, #1, #4, #8, #11, #14, and #21). My comments are as follows:

<Major comments>

#1

(Line 85) Please provide theoretical evidence(s) or reason(s) for the division of the hybrid varieties into Y1-Y3. Is the division based on differences in breeding objectives from period to period? Or does it reflect differences in breeding method? Without an explanation to this question, the classification Y1-Y3 seems arbitrary and makes me skeptical of the conclusions about phenotypic changes (e.g., Fig. 1c-k, Line 112-141). Perhaps the authors were trying to show "statistical significance" of phenotypic changes in hybrid rice varieties over time. In that case, statistical tests must be based on a valid null hypothesis, and the classification used in this study (i.e. Y1-Y3) does not seem to be valid for establishing a valid null hypothesis. If the authors want to show phenotypic changes over time, it is recommended to use simple X-Y plot where X is the release year and Y is the phenotypic value (interpolation using LOESS or LOWESS function is also valid). If there is no justification or reasonable explanation for Y1- Y3 classification, a simple X-Y plot is more objective than the classification.

#2

(Line 103) The objective of this analysis is not re-sequencing but genotyping. Therefore, information such as the number of called variant sites and the missing rate is more important than the read depth information.

#3

(Line 128) What does "head rice of the hybrids" mean? I think it is a typo of another word.

#4

(Line 125-126, Line 139-140) The author said that the hybrid rice breeding achieved enhancement of grain yield potential. Indeed, grain number showed significant increase (Fig. 1d, f). However, kilo-grain weight, which I think a phenotype directly related to yield potential, showed decrease in Y3 (Extended Data Fig. 4a). In addition, it was not clear whether yield per plant increased in Y3 or not (Extended Data Fig. 4c). Please provide explanations for this issue.

#5

(Fig. 2b, d) The 3D-plot is difficult to understand the results. Please split the 3D-plot into multiple 2D-plots (e.g., PC1 vs PC2, PC2 vs PC3, and PC1 vs PC3), otherwise it is difficult to check whether the conclusion described in line 145-146 is valid.

#6

(Line 160 and 178) I think "the superior alleles" means "breeding-favorable alleles", but it was difficult to understand. I recommend the authors to provide a brief description about "the superior alleles" in line 160. For example, "the superior (i.e., breeding-favorable) alleles"

#7

(Line 165-168) These messages were difficult for me to understand. Please elaborate on (1)

relationship between “the genetic diversity of the early-bred hybrid rice resources” (line 165) and Figure 2h (line 164) and (2) the basis or data for “the genetic diversity of hybrids gradually increased” (line 167).

#8

(Line 281–288) Please provide comparison of genomic selection model accuracy between additive and additive plus dominant model (i.e., genomic selection model with A and A+D). One of the most interesting points of genomic selection of hybrids is whether inclusion of non-additive genetic variable can increase prediction accuracy. In many cases, additive model shows higher prediction accuracy than additive plus dominant effect model (e.g., reviewed in Varona et al. 2013). If additive plus dominant effect model shows higher prediction accuracy than additive model, it provides evidence that non-additive effects contribute to phenotypic variation in the hybrids and may contribute to hybrid rice breeding.

* Varona et al. 2013. doi: 10.3389/fgene.2018.00078

#9

(Fig. 4d) A more detailed caption is needed for Fig. 4d. What does the gray shaded area mean? Are the definitions of the blue, yellow, and red shaded areas the same as in Figure 4c?

#10

(Fig 4d, line 258-264) Please describe the method to determine paternal and maternal allele of the F2 individuals.

#11

(Line 259 and 261) Does the “dominance effect” means the dominance effect estimated in the linear mixed model implemented in PLINK, or d/a that the authors used to represent impact on dominance effect in elsewhere (e.g. Fig. 3g)?

#12

(Line 265) I have concerns about definition of terms related to genomic selection. The authors used “phenotype prediction” (e.g. line 269), but since the model in line 692 does not include environmental effects, it is not appropriate to use “phenotype prediction”. The authors should use “genomic prediction” or “genomic selection” in this case. Besides, genomic selection model does not predict “the phenotype” but “the genomic estimated breeding value” because phenotypes are affected by environmental conditions. Please confirm definition of the terms.

#13

(Line 327) What does “the predicting phenotype” mean? Does it mean predicted phenotype value from the genomic selection model?

#14

(Line 336-338) Please provide a more detailed theoretical explanation of this message. This message is interesting and attractive, but I could not understand why the authors reached this indication. Is it because the genomic selection model included seed setting rate? Without a statement on this point, the result for rf3 seems a coincidence, and could be misleading to readers of the manuscript.

#15

(Line 565-577) How many biological replicates were prepared for grain related traits?

#16

(Fig. 2b, d) Please provide the PVE of each PC. This information is important for estimating strength of the population structure.

#17

(Line 628) Please provide criteria for selecting 19 indica and 23 japonica landraces to define the indica and japonica loci. Were the landraces selected based on a population structure analysis or results from other studies? Because some studies suggested ancient introgressions between indica and japonica, definition of japonica-indica haplotype is a sensitive problem. I am not asking the authors to perform precise japonica-indica definition. I think adding descriptions explaining the criteria would be sufficient and the readers can judge the validity and the reliability by themselves based on their own knowledge.

#18

(Line 646 and Extended Data Fig. 16) The message seems to be inconsistent between the conclusions and the results. Specifically, Nipponbare genome is not composed of 100% Japonica SNPs, and the same is true for Shuhui498. I agree with validity of the method and the results for this analysis, but use of "100%" (line 646) and "all" (Extended Data Fig. 16) is inappropriate from the results. I ask the authors to remove intuitive description and to include descriptions based on the results.

#19

(Line 658) Is it really covariance matrix? I think it is the covariate matrix. Please confirm.

#20

(Line 664) If the authors used permutation test to define significant threshold, then the value should vary by trait. Therefore, I am bit skeptical about the GWAS significant threshold in this study. Please provide the actual significant threshold value for each trait (e.g., heading date: 6.03, flag leaf length: 6.01, and so on).

#21

(Line 670). Is the calculation method for the index appropriate? Generally, in a population with population structure, average phenotype value is not appropriate for estimating effect of each SNP because the average phenotype includes effect of polygenic background effects. Since the authors used PLINK to perform the GWAS, I recommend the authors to use the estimated SNP effect to account for background polygenic effects.

<Minor comments>

#22

(Line 251-251) "Genetic complementary pyramided superior alleles in intraspecific hybrids" seems unnecessary. Perhaps, it is a subsection title?

#23

(Line 628) Perhaps, "Definition of indica-japonica differential loci" is a subsection title.

#24

(Line 701 -702) The authors used Z to represent marker genotype matrix. However, in line 692, Z is used for incident matrices. To avoid confusion, I recommend the authors to use other alphabets to

represent marker genotype matrix. It is also recommended to use different alphabets for the marker genotype matrix in line 701 and 702 because line 701 is an additive genotype value and line 702 is a heterozygous genotype value. (e.g., W for additive and H for heterozygote genotype value).

#25

(Line 710) Perhaps, "Selection index calculation" is a subsection title.

#26

(Fig. 5g, 6a, 6c, 6e) Alphabets on the box and whisker plots would be unnecessary because there are only two categories in the figures.

Author Rebuttal to Initial comments

Response to referees

Response to Referee #1

Remarks to the Author:

Hybrid rice breeding has contributed significantly to world food security. Although a number of heterotic loci has been identified, nevertheless, the genetic basis and mechanisms of rice heterosis still remain elusive. In particular, prediction of heterosis and thus genomics-guided hybrid breeding remains a challenging task. In this study, the authors performed a comprehensive genome analysis of 2839 elite rice hybrid varieties released in China over the past 40 years. These hybrids were classified into three groups based on their release time: Y1 (1976-2000); Y2 (2001-2010), and Y3 (2011-2020). Phenotyping analyses revealed significant improvement of three important agronomic traits in the hybrids over the breeding history: heading date, grain yield potential and grain quality. In combination with GWAS analysis of 17 agronomic traits in 2724 indica-indica hybrids and 4497 F2 plants from 10 indica-indica hybrids, 89 putative heterotic loci were identified (with 32 containing known genes). The authors showed that during modern hybrid rice breeding, the genetic diversity of hybrids gradually increased, accompanied by increased introgression of japonica genomic components and superior alleles. They also observed a stacking effect of superior alleles from Y1 to Y3 for multiple agronomic traits (heading date, leaf length, grain number per panicle, and grain quality) and purging of inferior alleles. Interestingly (but expectedly), they also found a high level of genome-wide complementarity in indica-japonica intersubspecific hybrids, including numerous known genes controlling important agronomic traits (such as DEPI, NAL1, Ghd7, Ghd8, GW5 and Hd1). The observation supports the notion that genetic complementary plays a major role in indica/japonica heterosis. At last, the authors constructed a genomic-selection model for future hybrid rice breeding based on the genomic and phenotypic information of 2839 hybrids and 9839 F2 plants derived from indica-indica and indica-japonica hybrids. The prediction accuracy for a set of traits ranged from 0.518 to 0.945. They further developed a customized selection index to improve the accuracy of the prediction model. They demonstrated the potential utility of such a model for genomic selection breeding programs of hybrid rice. Overall, this

study provides significant new insights into rice heterosis and established a prototype genomic selection model for future hybrid rice breeding, and this should generate a great impact on hybrid breeding in rice and other crops as well.

Answer: Thanks for the comments! We extend our appreciation for your constructive advice and revised our manuscript under your suggestions. Responses and modifications are presented as follows:

I have a number of questions and comments for further improvement of the manuscript:

#1. *Line 110: I am not sure what is the meaning of “head rice”. Please explain.*

Answer: “Head rice” was grains having its husk and outer brown layers removed. For better understanding, we have used the “polished grain” in place of “head rice” and given an explanation for “polished grain” in Methods section of the revised manuscript (Line 152, 258 and 673).

#2. *Line 135-136: “The four main subgroups of hybrids had distinguishing genomic components (Extended Data Fig. 5a)”. Could the authors elaborate more on this point—meaning and implications?*

Answer: By saying “*The four main subgroups of hybrids had distinguishing genomic components (Extended Data Fig. 5a)*”, we would like to express that “The four main subgroups of rice hybrids had distinguishing genomic structure”. In order to avoid ambiguity, the description for the phylogenetic analysis of hybrid accessions was changed into “According to our previous work, hybrids could be divided into five subgroups based on cytoplasm¹: three subgroups from the three-line breeding system and possessing Wild abortive (WA), Boro II (BT) or Honglian (HL) cytoplasm, and two subgroups from the two-line breeding systems and possessing Twoline-Jap (TJ) or Twoline-Ind (TI) cytoplasm. Phylogenetic analysis of nuclear genotype information implied that WA, TJ and BT hybrids were clearly separated with only a few exceptions and TI hybrids were genetically mixed with WA and TJ hybrids (Extended Data Fig. 7a in the revised manuscript). These findings were in close agreement with previous reports¹ and indicating different genetic resource was exploited for different breeding systems. HL hybrids were excluded for phenotypic analysis due to the small population size (17 HL hybrids in our collection)” at lines 155 to 164.

#3. *Fig. 2e-2f, I am not sure the meaning and definition of “incomplete (indica/japonica” and complete (japonica/japonica) states of the WA and TJ hybrids---how they are determined and measured? Based on what? Reasons? And what are the implications? Some explanation is warranted.*

Answer:**The procedure used for identifying introgression:**

Step1. Based on nineteen representative *indica* and twenty-three typical temperate *japonica* rice accessions reported previously², we identified 830,245 *indica-japonica* differential SNPs according to the following criterions: at the SNP site, ① ≥ 17 *indica* varieties were the same genotype; ② ≥ 21 *japonica* varieties hold the same genotype; ③ *indica* and *japonica* rice possessed different genotypes (Fig. 1 in response)

Step2. We judged the state of every locus (as *indica*, *japonica* or heterozygous genotype) out of all the 830,245 differentiated loci for each accession.

Step3. A 199-SNP sliding window with 1-SNP steps was applied to identify introgressive fragments across the whole genome for each accession. Within each of the 199-SNP length fragments, if there were at least 120 SNPs in homozygous genotype of *japonica*-origin, the segment was labeled as homozygous *japonica*-introgression; if there were at least 120 SNPs of heterozygous genotype (*indica/japonica*), the segment was deemed as heterozygous *japonica*-introgression. Adjacent fragments in the same state were then combined for further analysis.

And the procedure for introgression identification has been described in Methods at lines 730 to 748.

Fig. 1 | the procedure of identifying 830,245 *indica-japonica* differential SNPs.

Thus, “incomplete state” means that over 60% (120/199) of SNPs within the introgressive fragments were in heterozygous genotype (*indica/japonica*), and “complete state” means that over 60% of SNPs within the introgressive fragments were in homozygous genotype (*japonica/japonica*).

And for better understanding, the original sentence has been changed into “Across the whole genome, unequal levels of *japonica* introgression between WA and TJ hybrids were identified for both heterozygous and homozygous introgression (segment with over 60% of SNPs being the *indica/japonica* genotype was defined as heterozygous introgression, and that with more than 60% of SNPs being the *japonica/japonica* genotype was identified as homozygous introgression)” at lines 179 to 183.

#4. Line 199-200, could the authors elaborate more on the determination of dominant effect/additive effect (d/a) and its implications (such as for the *LL1p6* locus)? How in general such loci contribute to heterosis? How to define the superior allele or inferior allele based on such values and the controlled traits?

Answer:

Estimation of dominant effect/additive effect (d/a):

The estimation of index d/a was referred to previous report^{3,4} for loci identified by GWAS:

The average phenotype of individuals with heterozygous genotype and both homozygous genotypes were respectively calculated to estimate the index of d/a . The peak SNPs in association signals were used for genotyping and calculation of the index of d/a according to the relationships:

$$d = F_1 - \frac{P_1 + P_2}{2}$$

$$a = P_1 - \frac{P_1 + P_2}{2}$$

where F_1 , P_1 and P_2 were average phenotypic value of the heterozygous and both homozygous genotypes, respectively, and P_1 was the advantageous phenotypic value. And the SNP site, in which the number of individuals with either heterozygous or homozygous genotypes was fewer than 10, was excluded in the calculation of “ d/a ”. Furthermore, for quantitative trait loci (QTL) mapping by F_2 population, the index of d/a was estimated by IciMapping software⁵.

We have demonstrated the calculating formula of index d/a in the revised manuscript for better understanding (Line 815-827).

The implications of the index of d/a

We used the index to evaluate **the magnitude of dominance effect**. Taking *LL1p6* as an example, it controlled flag leaf length and the d/a index was estimated at 0.15, which indicated that the locus represented positive dominance but the effect was incomplete (The heterozygous genotype did not exhibit comparable performance to those of the advantageous homozygous genotype). The enlarged leaf size positively linked to rice grain yield^{6,7} and enlarged leaf size was observed during improvement breeding, thus the allele associated with longer leaf length was defined as the breeding-favorable allele for *LL1p6* in our study. Because the dominance effect of *LL1p6* was incomplete, the frequency of breeding-unfavorable allele of *LL1p6* decreased during hybrid improvement breeding, and more individuals in Y3 breeding stage hold the advantageous homozygous genotype, especially for TI hybrids (Fig. 2 in response). Based on the magnitude of dominance effect and allelic frequency change tendency of target loci, we could learn how breeding history exploited heterotic loci. The related analysis have been also demonstrated in the manuscript (Line 229-233, Fig. 3e, Extended Data Fig. 12 a, b and Extended Data Fig. 14-16 in the revised manuscript).

Fig. 2 | *LL1p6* was identified to control leaf length and represented positive partial dominance effect. **a**, Local Manhattant plot surrounding the genomic regions where the candidate locus located. **b**, Phenotypic analysis of three genotypes. Symbol “+ / +” and “- / -” represented breeding-favorable and -

unfavorable homozygous genotypes for breeding, respectively. And “+/-” was heterozygous genotype. The leading SNP marked in dark blue in panel a was used for genotyping. c, Frequency of three genotypes in wild abortive (WA), Twoline-Jap and Twoline-Ind hybrids. Genotype “+/+”, “+/-” and “-/-” were marked in blue, orange and green, respectively.

The definition of the superior alleles

A “superior allele” was defined as the breeding-favorable allele in this study. For newly identified loci, the leading SNPs was used for genotyping, and the average phenotype of individuals with heterozygous genotype and both homozygous genotypes were calculated (the method referred to previous work^{3,4}) to determine the breeding-favorable allele. The alleles associated with better performance was defined as the breeding-favorable alleles. And for loci associated with known genes, the definition of the breeding-favorable alleles was in accord with that in the previous report⁸ (<http://www.xhhuanglab.cn/tool/RiceNavi.html>). For grain yield-related traits, comprising yield per plant, full grain number per plant, valid panicle number, full grain number per panicle, seed setting rate and kilo-grain weight, the allele leading to higher observation values was considered to be the breeding-favorable allele. For morphological traits discussed in our work (leaf length and width, tiller angle), the allele leading to larger source organ (longer leaf length and wider leaf width) was defined as the breeding-favorable allele; erect plant architecture permitted high planting density, and allele associated with larger tiller angle (the angle between tiller and the ground level) was determined as the breeding-favorable allele. With respect to heading date, short heading date resulted in yield penalty but long heading date was impractical for agricultural production in most rice planting area, thus an appropriate heading date was chosen for hybrid breeding⁹. The collections in our study comprised commercial hybrids, and almost all accessions completed their life cycle when planting in Hangzhou. We observed a slightly shortened heading date in Y3 hybrids. Thus, allele associated with shorter heading date was defined as the breeding-favorable allele in our study for allelic frequency analysis. Definition of the breeding-favorable allele for grain eating and cooking quality referred to the Chinese national standard (NY/T 593-2013 ‘cooking rice variety quality’). According to the standard, the values of chalkiness and grain translucency (level) were lower for *O. sativa* varieties with better grain quality, and the value of gel consistency was higher for *O. sativa* varieties with better grain quality. Thus, alleles leading to lower chalkiness, lower grain translucency, or higher gel consistency, were deemed as the breeding-favorable allele in our study. With respect to amylose content, the value range was 13.0-18.0 for first-class *O. sativa indica* and *japonica* rice varieties, was 13.0-20.0 for second-class *O. sativa indica* rice varieties, 13.0-19.0 for second-class *O. sativa japonica* rice varieties, was 13.0-22.0 for third-class *O. sativa indica* rice varieties and was 13.0-20.0 for third-class *O. sativa japonica* rice varieties. The value range was narrower for first-class varieties, with a lower upper limit. Thus, for convenience, the allele associated with lower amylose content within limits (13.0-22.0) was determined as the breeding-favorable allele.

For better understanding, a clearer definition of “breeding-favorable allele” has been described in Methods (Line 852-883) and the “superior” was replaced by “breeding-favorable” and “inferior” was replaced by “breeding-unfavorable” in the revised manuscript.

#5. I am wondering whether the developed prediction model could be applied to different subgroups of hybrids with comparable accuracy? Also, the prediction accuracy varied substantially for different agronomic traits, could the authors elaborate on the possible causes and solutions?

Answer: According to our previous work, hybrids could be divided into five subgroups based on cytoplasm¹: three hybrid subgroups from the three-line breeding system and possessing WA, BT, HL cytoplasm, and two subgroups from the two-line breeding system and possessing TJ and TI cytoplasm. Phylogenetic analysis based on nuclear genotype information implied the differentiated genomic structure among four hybrid subgroups (Extended Data Fig. 7a in the revised manuscript) (due to the small population size, hybrids holding HL cytoplasm were not discussed in our study). And the four subgroups of hybrids were named as WA, BT, TJ and TI hybrids for convenience hereafter. Among all rice hybrids supplied for model construction, 62.05%, 24.96%, 8.61% and 3.74% were WA, TJ, TI and BT hybrids. Furthermore, among all the 9,839 F₂ individuals, 21.27%, 16.14%, 14.61% and 47.98% were from WA, TJ, TI and BT hybrids. Thus, abundant hybrids from each of the four subgroups were included in the training set, and the sample set was representative. As advised, we further proceeded to assess the prediction accuracy within subgroup, and found the prediction accuracy for four subgroups was comparable to that of all hybrids for all seven traits (Fig. 3 in response), except for the seed setting rate in BT hybrids. The relatively low accuracy of seed setting rate in BT hybrids might be explained by the intricate genetic underpinnings of fertility in interspecific rice hybrids (most *indica-japonica* hybrids were BT type). The prediction accuracies for four subgroups were added to the revised manuscript (Line 345-351 and Extended Data Fig. 21 in the revised manuscript).

Fig. 3 | Prediction accuracy of seven selected agronomic traits for all rice hybrids and four subgroups.

The variation in prediction accuracy for different agronomic traits may be due to a number of factors:

i) The complexity of the trait itself. For example, yield per plant was influenced by valid panicle number, seed setting rate, grain number per panicle, kilo-grain weight and other traits, and controlled by multiple genetic loci and complex physiological pathways. Considering the variable prediction accuracy for different traits, selection index comprehensively considering multiple traits was applied to conduct decision-making model in our study.

ii) Heritability difference. Quantitative traits were substantially affected by environment, thus phenotypic variations of them could result from genetic factors, environmental factors, and environmental and genetic interaction factors. Different traits had different sensitivity to environment. For example, grain shape was more stable under different environments, while heading date varied substantially. One possible solution is to collect data from multiple environments, and construct model accounting for genotype-environment interaction¹⁰.

#6. I suggest the authors to cite the recent studies by Li et al. (*Nature Plants*, 2022 Jul;8(7):750-763) and Wang et al. (*Nature Genetics*, 2023, <https://www.nature.com/articles/s41588-022-01283-w>) on maize heterotic groups and maize heterosis and properly discuss them in the context of the findings made in this work.

Answer: I read the recent studies by Li¹¹ and Wang¹² carefully, and Wang's work helped me to get a deeper understanding of genetic complementation underlying heterosis and Li's work inspired me to think about the importance of improving parental lines for hybrid breeding:

Wang found that “the significantly positive correlation between the SV numbers in bi-parents and better-parent heterosis in hybrid maize provided the support for a prevalent role of genetic complementarity underlying heterosis”, and genetic complementarity (dominance) were also identified as a contributing factor to heterosis in both Arabidopsis and rice. We cited it to introduce the advance in heterosis research in crops (Line 57-59).

In our study, we found that both parental lines simultaneously acquired breeding-favorable alleles in improvement breeding, especially for major loci controlling grain quality and representing negative dominance effect. In Li's work¹¹, convergent changes of several agronomic traits were observed in both parents of hybrid maize, and co-selection of genomic loci were also identified in both parental lines; hybrid improvement was accompanied with improvement in both parental lines of hybrids. We cited their work to support our opinion: parental improvement was vital to hybrid breeding, and discussed it in the Discussion section of the revised manuscript (Line 427-439). Due to prevalent genetic complementarity^{9,12}, multiple genomic loci combined in hybrid led to heterosis. In agricultural practice, heterosis was not always observed in randomly crossed combinations, and the loci representing negative dominance effect, might be responsible. Thus, we reasoned that improving the parental lines by simultaneously equipping both parental lines with breeding-favorable genotypes for those loci, could avoid the loci with negative dominance effect dragging down the performance of F₁ generations and help to increase the probability to generate heterosis in F₁ generation.

#7. Careful proofreading is recommended to improve the clarity and logics of this manuscript.

Answer: Following your advice, the manuscript underwent a comprehensive review by a professional writer to improve its quality and enhance its readability.

Response to Referee #2

Remarks to the Author:

This paper describes a large experiment to examine the genomic basis for heterosis in hybrid rice. Hybrid rice has been a key driver of agricultural productivity, particularly in China, yet the genomic basis for

hybrid vigor remains elusive. This is clearly an issue with most examples of heterosis. In this study, they scan the genomes of a very large collection of hybrid rice, identify genome regions/genes associated with increased vigor and develop a predictive framework for hybrid rice vigor that could be applied to breeding efforts.

First, the scope of this study is large and is fairly comprehensive. With such a large data set, it is one of the most comprehensive genome scans of heterosis I am aware of. Moreover, the ability to provide a predictive tool could be of great importance to breeders in the future.

Answer: Thanks for the comments. To facilitate access to the resource, we have constructed a web-based tool (<https://www.bic.ac.cn/HybridRice/#>). Users can rapidly access the sample, phenotype, variant and genome-wide analysis information of interest. The tool also demonstrates the allele frequency and potential impact of variants (according to annotation by SnpEff¹³), and conducts haplotype analysis for a small subset of variants.

Nevertheless, there are major issues with the paper that makes me less enthusiastic about it.

#1. First is the definition of heterosis. The authors frequently invoke “superior” alleles that appear to go with “superior” phenotypes. But defining superiority can be subjective, and so it is unclear how to think about this in a genetic context. For traits that are directly related to yield – grain number, panicle number, biomass – hybrid vigor is a clear phenomenon since in these cases vigor is equivalent to higher yield or size. But for flowering time, grain cooking quality and so forth, superiority is based in human desires. In those cases, it would be better to discuss these types of traits are over- or underdominance.

Answer:

Thanks for the comments. We have used “breeding-favorable alleles” to replace “superior alleles” in the revised manuscript. For the loci associated with known genes, the definition of the **breeding-favorable alleles** was in accord with that in the previous report⁸ (<http://www.xhhuanglab.cn/tool/RiceNavi.html>). For those newly identified loci, the leading SNPs was used for genotyping, and the average phenotype of individuals with heterozygous genotype and both homozygous genotypes were calculated (the method referred to previous work^{3,4}) to determine the breeding-favorable allele. The alleles associated with better performance was defined as the breeding-favorable alleles.

We defined the “breeding-favorable” phenotypes based on the following standards:

i) With respect to grain yield-related traits, comprising yield per plant, full grain number per plant, valid panicle number, full grain number per panicle, seed setting rate and kilo-grain weight, the allele leading to higher observation values was considered to be the breeding-favorable allele.

ii) With respect to morphological traits discussed in our work (leaf length and width, tiller angle), the allele leading to larger source organ (longer leaf length and wider leaf width) was defined as the breeding-favorable allele; erect plant architecture permitted high planting density, and allele associated with larger tiller angle (the angle between tiller and the ground level) was determined as the breeding-favorable allele.

iii) With respect to heading date, short heading date resulted in yield penalty but long heading date was impractical for agricultural production in most rice planting area, thus an appropriate heading date was chosen for hybrid breeding⁹. The collections in our study comprised commercial hybrids, and almost all accessions completed their life cycle when planting in Hangzhou. We observed a slightly shortened heading date in Y3 hybrids. Thus, allele associated with shorter heading date was defined as the breeding-favorable allele in our study for allelic frequency analysis.

iv) Definition of the breeding-favorable allele for grain eating and cooking quality referred to the Chinese national standard (NY/T 593-2013 ‘cooking rice variety quality’). According to the standard, the values of chalkiness and grain translucency (level) were lower for *O. sativa* varieties with better grain quality, and the value of gel consistency was higher for *O. sativa* varieties with better grain quality. Thus, alleles leading to lower chalkiness, lower grain translucency, or higher gel consistency were deemed as the breeding-favorable allele in our study. With respect to amylose content, the value range was 13.0-18.0 for first-class *O. sativa indica* and *japonica* rice varieties, was 13.0-20.0 for second-class *O. sativa indica* rice varieties, 13.0-19.0 for second-class *O. sativa japonica* rice varieties, was 13.0-22.0 for third-class *O. sativa indica* rice varieties and was 13.0-20.0 for third-class *O. sativa japonica* rice varieties. The value range was narrower for first-class varieties, with a lower upper limit. Thus, for convenience, the allele associated with lower amylose content within limits (13.0-22.0) was determined as the breeding-favorable allele.

The definition of superiority has been added to the revised manuscript (Line 852-883).

Then, as advised, we evaluated the better-parent heterosis (BPH) for all traits based on the definition of “superiority” above. Because the parental lines were not available for most commercial rice hybrids, it was impractical to collect parental lines for all F₁s. Among our collections, 123 hybrids had both their

parental lines collected. The subset of hybrids was released from the years 1985 to 2017 and was representative (Fig. 4 in response). The parental lines were planted in the same condition as F₁s and phenotypic investigation was conducted according to the standard applied for F₁s. Based on the parents-hybrid traits, we evaluated the BPH for all traits (Fig. 5 in response). Most hybrids had longer heading date than the early-heading parental line (Fig. 5a in response). With respect to morphological traits (Fig. 5b-e in response), over one-third of hybrids demonstrated BPH. For yield-related traits (Fig. 5f-k in response), most hybrids showed BPH, except for the valid panicle number. With respect to grain cooking and appearance quality, partial hybrids exhibited BPH (Fig. 5i-p in response). Most rice hybrids did not have heterosis in quality. It might result from that pursuit for yield heterosis has been prioritized through the whole breeding history of rice hybrid, while a growing emphasis has been placed on enhancing grain quality more recently.

We added the observations in the revised manuscript (Line 115-122 and Extended Data Fig. 4).

Table 1a. The grade of grain quality of *O. sativa indica* rice variety (from NY/T 593-2013)

Item	first class	second class	third class
Amylose content (%)	13.0-18.0	13.0-20.0	13.0-22.0
Gel consistency (mm)	≥60	≥60	≥50
Chalkiness (%)	≤1	≤3	≤5
Grain translucency (level)	≤1	≤2	≤2

Table 1b. The grade of grain quality of *O. sativa Japonica* rice variety (from NY/T 593-2013)

Item	first class	second class	third class
Amylose content (%)	13.0-18.0	13.0-19.0	13.0-20.0
Gel consistency (mm)	≥70	≥70	≥60
Chalkiness (%)	≤1	≤3	≤5
Grain translucency (level)	≤1	≤2	≤2

Fig. 4 | The breeding year for the 123 hybrids with both parental lines available.

Fig. 5 | Better parent heterosis (BPH) for heading date, morphological characteristics, grain yield-related traits and grain quality-related traits in 123 parents-hybrid trios. Each dot represented a parents-hybrid trio. Red dots represented trios with $BPH > 0$ while green dots were trios with $BPH \leq 0$. The

index of BPH was estimated as $\frac{F_1 - P_1}{P_1}$, where F_1 and P_1 respectively were corresponding phenotypic measurements of the hybrid and parental line with better performance. Because the female lines were sterile, it was difficult to investigate the grain yield-related and quality-related traits. And in rice hybrids, male parents generally have much better yield performance than female parents, so BPH of yield-related traits mostly refers to the advantage over male parents⁹. Taking above two points into consideration, the phenotypic observations from restorer lines were used for estimating the index of BPH for grain yield-related and quality-related traits (Full grain number per plant, full grain number per panicle, seed setting rate, kilo-grain weight, yield per plant, grain shape, amylose content, gel consistency, chalkiness and chalky grain percentage).

#2. The molecular genetic basis of heterosis continues to be the central issue of hybrid vigor research. Just what is happening behind this phenomenon? The authors in their Introduction point out that there are several models to try to explain heterosis. A good use of their data would be to set up a strong test of the different models so that we can advance our understanding of the genetic basis of heterosis.

#3. This comment has to do with point 2 above – the authors identify several genomic regions associated with heterosis in hybrid rice. Identifying heterotic loci is not new – what would be new is if we could understand how it all works – what are the molecular mechanisms that translate from heterotic loci to vigor. Is it just overdominance – and if so, how? Is it masking deleterious recessives?

Answer:

Thanks for the suggestions. By taking these suggestions, we conducted QTL mapping using eighteen F_2 populations for five yield related traits. The five yield-related traits, which exhibited BPH in most parents-hybrid trios (Fig. 5 in response), comprised yield per plant, full grain number per panicle, full grain number per plant, kilo-grain weight and seed setting rate. Based upon the QTLs identified, we further evaluated the contribution of different models to heterosis. In rice hybrids, male parents generally have much better yield performance than female parents, so BPH mostly refers to the advantage over male parents⁹. Therefore, the over-parent heterosis observed in F_1 hybrids was predominantly attributable to the influence of positive partial dominance effect loci contributed by female parents, as well as over-dominance effect loci. Thus, we assessed the contributions of dominance and overdominance effect to heterosis, based on female-contributed QTLs that displayed positive dominance effects and the QTLs that demonstrated over-dominance effects, respectively. We also estimated phenotypic variance explained (PVE) by each QTL, as the magnitude of its effect.

We found that, in *indica-indica* hybrids, both positive partial dominance effect and over-dominance effect made comparable contributions for four yield-related traits (yield per plant, full grain number per panicle, full grain number per plant and seed setting rate), and the QTLs with over-dominance effect had larger

PVE (Fig. 6 in response). For kilo-grain weight, QTLs with positive partial dominance effect were predominant. In *indica-japonica* hybrids, a relatively balanced distribution of positive partial dominance and over-dominance QTLs was also observed for four yield-related traits (yield per plant, full grain number per panicle, full grain number per plant and seed setting rate), and partial dominance QTLs had higher PVE than over-dominance QTLs for yield per plant and full-grain number. With respect to kilo-grain weight, most QTLs demonstrated partial dominance effects.

The collective findings indicated that both dominance complementation and over-dominance effects played important roles in both intra- and inter-specific heterosis for grain yield-related traits, except for kilo-grain weight. Over-dominance accounted for a relatively larger proportion of PVE in intraspecific rice hybrids, while dominance complementation accounted for a relatively larger proportion of PVE in interspecific rice hybrids. It was worth to note that the over-dominance loci referred to in our study may also include the *pseudo*-overdominance loci, which was conferred by tightly linked genes in opposite phase. With respect to kilo-grain weight, dominance complementation played the leading role.

Because the genetic architecture for *indica-indica* hybrids has been well elucidated in previous report⁹, we added the observations in *indica-japonica* hybrids in the revised manuscript (Line 314-323 and Extended Data Fig. 19).

a *Indica * Indica* populations

b *Indica * Japonica* populations

Fig. 6 | Evaluating dominance complementation and over-dominance effects for QTLs in eighteen F₂ populations from elite hybrids. a, The phenotypic variance explained by both partial dominance and over-dominance QTLs mapped in ten *indica-indica* F₂ populations. The ratio of d/a was used for measuring the magnitude of dominance effect. $d/a \leq 1$ and >1 referred to positive partial dominance and over-dominance effect, respectively. **b,** The phenotypic variance explained by both partial dominance and over-dominance QTLs in eight *indica-japonica* F₂ populations.

In current work, we focused more on how heterotic loci were exploited by hybrid improvement breeding, which advanced the understanding of rice heterosis and its applications. Elucidating the molecular mechanisms for heterotic genes, will be our diligent direction, since we have mapped several heterotic loci by both GWAS and linkage mapping. We will further narrow down the heterotic QTLs, and investigate their molecular mechanisms through cloning the heterotic genes. However, this process will be long, and it is difficult to discuss the molecular mechanisms of heterotic genes in this manuscript.

The paper as written does make a contribution to our understanding of heterosis but the contribution is incremental and does not really go to the heart of the matter.

Answer: As advised, we have evaluated the contribution of dominance and over-dominance effects to heterosis for traits exhibiting over-parent heterosis in most commercial hybrids (Fig. 6 in response).

A final thought – it would help to have a professional writer go through the text as there are some awkward sentence structures.

Answer: Following the advice, the manuscript has undergone a thorough professional review to improve its quality and enhance its readability by a professional writer. All necessary changes have been made in the revised manuscript.

Response to Referee #3

Remarks to the Author:

This study performed re-sequencing of 2,839 hybrid rice varieties to elucidate genetic mechanism of hybrid performance in rice. Through nucleotide diversity survey and GWAS, the authors found loci associated with rice hybrid breeding. In addition, the authors applied genomic selection approach to the rice hybrid varieties and proposed a strategy to contribute to future rice hybrid breeding. Since such a

large-scale data are rare and valuable in plant science, the data obtained in this study will contribute not only to hybrid breeding but also to genetic study of hybrids. However, I have concerns about the theoretical validity of the analytical methods and the conclusions drawn from the analyses. Some of the concerns will affect key message of the manuscript (especially, #1, #4, #8, #11, #14, and #21). My comments are as follows:

Answer: We sincerely appreciate for your invaluable and constructive suggestions, which we believe will significantly enhance the quality and readability of our manuscript. Our replies for your advices and corresponding modifications are described as follows:

<Major comments>

#1. (Line 85) Please provide theoretical evidence(s) or reason(s) for the division of the hybrid varieties into Y1-Y3. Is the division based on differences in breeding objectives from period to period? Or does it reflect differences in breeding method? Without an explanation to this question, the classification Y1-Y3 seems arbitrary and makes me skeptical of the conclusions about phenotypic changes (e.g., Fig. 1c-k, Line 112-141). Perhaps the authors were trying to show “statistical significance” of phenotypic changes in hybrid rice varieties over time. In that case, statistical tests must be based on a valid null hypothesis, and the classification used in this study (i.e. Y1-Y3) does not seem to be valid for establishing a valid null hypothesis. If the authors want to show phenotypic changes over time, it is recommended to use simple X-Y plot where X is the release year and Y is the phenotypic value (interpolation using LOESS or LOWESS function is also valid). If there is no justification or reasonable explanation for Y1- Y3 classification, a simple X-Y plot is more objective than the classification.

Answer: Thanks a lot for your constructive suggestions. We conducted Y1-Y3 classification according to following reasons:

i) Proliferation of hybrids from the two-line breeding system in 2000s. The first commercial hybrid from the two-line breeding system was generated in **1994**. The boom and widely cultivation of two-line hybrid varieties were since **2000** according to previous report¹⁴, what was also observed in our work (Fig. 7 in response);

Fig. 7 | Yearly statistics of released varieties and cultivated area for two-line hybrids in China. a, Yearly statistics of two-line hybrid varieties released from 1996 to 2013. **b,** Yearly statistics of the cultivated area of two-line hybrid in 1996 to 2013. Data and figures shown in a and b were from Hu’s work¹⁴. **c,** The yearly statistics of released two-line hybrids in our collections. Twoline-Jap and Twoline-Ind hybrids were marked in blue and purple, respectively.

ii) Policy steering to breed “super rice” in late 1990s and 2000s. China Ministry of Agriculture established a nationwide mega project entitled “Breeding and cultivation system of super rice in China” in **1996**. The project encouraged combined use of heterosis and ideal plant type to increase yield of rice, and it sets standards for ideal plant type of super rice: erect panicle model, long erect leaves with delayed senescence model, early vigorous growth and heavy panicle model. The policy led to major advance in breeding and production in China and elsewhere¹⁵. Considering the breeding cycle and time interval, it could be inferred that the initial variety responding to policy regulation might be generated around year **2000**;

iii) Progress of breeding technology in 2010s. The draft sequence of the rice genome (both *O. sativa* L. ssp. *japonica* and *indica*) was reported in 2002^{16,17}, and the finished quality sequence was released in 2005¹⁸. With the reference sequence available and development of high-throughput sequencing and genotyping¹⁹, rice molecular biology and functional genomics made great progress, including identification of key functional genes (Fig. 8 in response), which further accelerated the rational design and molecular breeding of rice hybrid by marker assisted selection and marker assisted gene pyramiding **in the past decade**^{20,21};

Fig. 8 | The yearly statistics of newly cloned functional genes in 1995 to 2020. The data was from Wei's report⁸.

Taking the above points into consideration, we performed Y1-Y3 classification, and analyzed the phenotypic and allelic frequency change according to the classification. We provided the explanation for Y1-Y3 classification in the revised manuscript for better understanding (Line 129-131 and 635-646).

Furthermore, followed your advice, X-Y plots were also drawn to show the phenotypic changes over time (Fig. 9 in response). And correlation analysis between release year and phenotypic observations was also conducted to exhibit the significant change. And the tendency of phenotypic change was corresponding to the analysis conducted in Y1-Y3 classification. The result based on X-Y plot was incorporated in the revised manuscript to better demonstrate the achievement of improvement breeding (Line 139-140 and Extended Data Fig. 6 in the revised manuscript).

Fig. 9 | Phenotypic change during improvement breeding for rice hybrids. Correlation coefficient and corresponding p value were estimated by `cor.test()` function in R. *: $0.01 < p \leq 0.05$, **: $0.001 < p \leq 0.01$, ***: $p \leq 0.001$.

#2. (Line 103) The objective of this analysis is not re-sequencing but genotyping. Therefore, information such as the number of called variant sites and the missing rate is more important than the read depth information.

Answer: We did not call variations for F_2 individuals directly due to the low sequencing depth (0.2× on average). We inferred the genotype of each F_2 individual referring to previous report¹⁹:

The identification of high-quality SNPs for the parental lines, which were used to generate the F_2 population, was conducted using the pipeline described in “Sequencing and variation calling” section of Methods. We then created the *pseudo*-reference sequence for both parental lines by replacing the Nipponbare bases with the corresponding bases at SNPs identified in both parents. We aligned the paired-end reads of F_2 individuals to the *pseudo*-sequence of both parents using BWA (version 0.7.1)²², and implemented a customized Perl script to determine the origin (paternal or maternal) of reads according to the SNPs within the reads. Then, we took a sliding-window approach to call genotypes and determine recombination breakpoints. Within each window, the numbers of maternal-origin and paternal-origin reads were counted. The genomic region covered by the sliding window was called as homozygous maternal genotype when over 80% reads were judged as maternal origin; called as homozygous paternal genotype when over 80% reads were paternal origin; otherwise called as heterozygous genotype. As the window sliding, genotypes were called and recombination breakpoints were determined.

Thus, the called variant sites and the missing rate for parental lines were key factors for the resolution and accuracy of genotyping. In total, we identified 2,218,510 high-quality SNPs for F_2 genotyping in our study. And the average per-site missing rate was estimated at 4.56%. Meanwhile, the average per-individual missing rate was 4.56%, with the minimum missing rate being 0.94% and maximal missing rate being 7.68%. And the number of differentiated SNPs for genotyping in each F_2 population shown in Table 2 in response.

Table 2. Counts of differentiated SNPs used for genotyping for F_2 populations

F_2 Populations	Ecotype ^a	Counts of SNPs used for genotyping
Pop1	Indica-Indica	461,093
Pop2	Indica-Indica	350,300
Pop3	Indica-Indica	438,034
Pop4	Indica-Indica	492,683
Pop5	Indica-Indica	411,300

Pop6	Indica-Indica	495,721
Pop7	Indica-Indica	489,616
Pop8	Indica-Indica	480,675
Pop9	Indica-Indica	382,633
Pop10	Indica-Indica	690,169
Pop11	Indica-Japonica	658,151
Pop12	Indica-Japonica	1,138,158
Pop13	Indica-Japonica	833,387
Pop14	Indica-Japonica	863,026
Pop15	Indica-Japonica	1,011,741
Pop16	Indica-Japonica	776,013
Pop17	Indica-Japonica	675,829
Pop18	Indica-Japonica	737,867

a, The Ecotype for F_{1s} used to generate F_2 populations. “*Indica-indica*” indicated *indica* intraspecific cross; “*Indic-japonica*” represented interspecific cross.

#3. (Line 128) What does “head rice of the hybrids” mean? I think it is a typo of another word.

Answer: Sorry for the unclear description. “Head rice” was the grain with its husk and outer brown layers removed. For better understanding, we have used the “**polished grain**” in place of “head rice” and given an explanation for the “polished grain” in Methods section of the revised manuscript (Line 152, 258 and 673).

#4. (Line 125-126, Line 139-140) The author said that the hybrid rice breeding achieved enhancement of grain yield potential. Indeed, grain number showed significant increase (Fig. 1d, f). However, kilo-grain weight, which I think a phenotype directly related to yield potential, showed decrease in Y3 (Extended Data Fig. 4a). In addition, it was not clear whether yield per plant increased in Y3 or not (Extended Data Fig. 4c). Please provide explanations for this issue.

Answer: We did observe the significant increase of grain numbers (Fig. 1d and f in the revised manuscript) but decrease of kilo-grain weight (Extended Data Fig. 5a in the revised manuscript) in Y3 rice hybrids. And the yield per plant slightly increased (Extended Data Fig. 5c in the revised manuscript). In previous manuscript, the description about “enhancement of grain yield potential in breeding process” might be ambiguity. Thus, in the revised manuscript, we replaced the description of “enhancement of grain yield potential” by “enlarged source and sink organ” (Line 138, 167 and 418). We further analyzed the possible reasons for significant increase of source and sink organ but modest increase of yield and significant decrease of kilo-grain weight (Line 278-286 and 418-426):

i) Heading date become slightly shorter in Y3 (Figure 1c in the revised manuscript). Practically, the climatic condition was changeable in the late growth period of cultivated rice, thus slightly shortening the growth period could aid stable production. However, shorter growth period also brought challenge for grain yield maintaining.

ii) *GW5* (Os05g0187500) was identified as major locus controlling kilo-grain weight in *indica-indica* rice hybrid by GWAS (Extended Data Fig. 10e and Supplementary Information 4 in the revised manuscript). Allele related with higher grain weight for gene *GW5* was almost omitted in Y3 hybrids (Fig. 10a in response). It has been reported that, *GW5* regulated grain width and grain weight²³⁻²⁵ and the allele with higher length-to-width ratio showed a lower level of grain chalkiness²⁶. In our study, we found that the allele of *GW5* associated with higher kilo-grain weight had negative effect on length to width ratio and grain appearance quality, including chalkiness and chalky grain percentage (Fig. 10b-e in response). Thus, the *GW5* allele with higher grain weight was discarded in favor of grain-quality improvement.

As part of the improvement breeding efforts, source and sink organs, which have previously been shown to be closely linked to rice grain yield, significantly enlarged^{6,7}. We believe that this has helped to maintain a balance between grain yield and quality, particularly in instance of slightly shortened growth period.

iii) We also found decreased frequency of alleles with high nitrogen use efficiency in Y3 hybrids (Fig. 11 in response). The loss of high nitrogen use efficiency alleles might also be responsible for the modest yield increase because normal agricultural management, not high nitrogen management, were applied in our study.

Fig. 10 | Allelic frequency change across three breeding periods and phenotypic distribution of three genotypes for *GW5*. a, Allelic frequency change of *GW5* across three breeding periods. b-e, Boxplot demonstrated the phenotypic distribution of three genotypes for *GW5*. The peak associated SNP for kilo-grain weight was used for genotyping. Homozygous genotypes associated with high (Geno A in panel b to e) and low (Geno B in panel b to e) kilo-grain weight were marked by blue and green, respectively; and heterozygous genotype (Geno H in panel b to e) was indicated by orange.

Fig. 11 | Allelic frequency change of three loci related with nitrogen use efficiency in three breeding periods. a, *OsNPF6.1*; b, *ARE1*; c, *OsNR2*. The reported causative variations⁸ of the three loci were chosen to do genotyping. Homozygous genotypes leading to high and low nitrogen use efficiency were respectively marked in blue and green, while heterozygous genotype was marked in orange.

#5. (Fig. 2b, d) The 3D-plot is difficult to understand the results. Please split the 3D-plot into multiple 2D-plots (e.g., PC1 vs PC2, PC2 vs PC3, and PC1 vs PC3), otherwise it is difficult to check whether the conclusion described in line 145-146 is valid.

Answer: We have split the 3D-plot into multiple 2D-plots in the revised manuscript (Fig 2b, d and Extended Data Fig. 8 in the revised manuscript). *O.sativa indica* and *japonica* varieties could be separated by PC1 and PC2 (Fig. 12 in response), and PC2 and PC3 further distinguished the genomic differentiation between the wild abortive and Twoline-Jap hybrids (Fig. 13p-r in response). Due to the large quantity of dots, the display effect was suboptimal when all dots were presented together. We further successively highlighted dots from three breeding periods in three separate panels to clearly demonstrate the population structure of individuals (Fig. 13a-l in response). And dots for WA and TJ hybrids were also highlighted separately for three pairs of PCs (Fig. 13m-u in response). The set of figures was also incorporated into the revised manuscript (Extended Data Fig. 8).

Fig. 12 | Principal component analysis for rice hybrid. Colored dots indicated hybrid ecotypes.

Fig. 13 | Principal component analysis for rice hybrids. a-l, Dots were colored according to breeding period. Red, dark blue and yellow dots represented individuals from Y1, Y2 and Y3 breeding periods, respectively. (a-d), Plot of PC1 and PC2. Dots for hybrids from three periods were separately highlighted in panels b to d. (e-h), Plots of PC2 and PC3. Dots for hybrids from three periods were separately highlighted in panels f to h. (i-l), Plots of PC1 and PC3. Dots for hybrids from three periods were separately highlighted in panels j to l. **m-u,** Dots were colored according to cytoplasm type. Green and light blue indicated individuals possessing wild abortive and Twoline-Jap cytoplasm, respectively. (m-o), Plot of PC1 and PC2. Dots for wild abortive and Twoline-Jap hybrids were separately highlighted in panel n and o. (p-r), Plots of PC2 and PC3. Dots for wild abortive and Twoline-Jap hybrids were separately highlighted in panel q and r. (s-u), Plots of PC1 and PC3. Dots for wild abortive and Twoline-Jap hybrids were separately highlighted in panel t and u.

#6. (Line 160 and 178) I think “the superior alleles” means “breeding-favorable alleles”, but it was difficult to understand. I recommend the authors to provide a brief description about “the superior alleles” in line 160. For example, “the superior (i.e., breeding-favorable) alleles”.

Answer: Yes, “superior alleles” means “breeding-favorable alleles”. For better understanding, the “superior” was replaced by “breeding-favorable” and “inferior” was replaced by “breeding-unfavorable” in the revised manuscript. Thanks for your advice.

#7. (Line 165-168) These messages were difficult for me to understand. Please elaborate on (1) relationship between “the genetic diversity of the early-bred hybrid rice resources” (line 165) and Figure 2h (line 164) and (2) the basis or data for “the genetic diversity of hybrids gradually increased” (line 167).

Answer: The description in previous manuscript was somehow misleading, and there was no correlation between line 164 and line 165. The description at lines 165 to 168 (in previous manuscript: *The above results demonstrated that the genetic diversity of the early-bred hybrid rice resources was relatively narrow. With the generation and wide cultivation of various types of hybrids, such as TJ hybrids, the genetic diversity of hybrids gradually increased. Specially, the superior alleles were also introduced for improvement breeding during the process*) was the summary for Figure 2 in the revised manuscript. The data shown in Figure 2a-c and Extended Data Figure 9 in the revised manuscript (nucleotide diversity, PCA, admixture and kinship analysis) indicated that the genetic diversity increased during improvement breeding of rice hybrid. Furthermore, we also observed the number of Twoline-Jap (TJ) hybrids increased from Y1 to Y3 (Fig. 2c in the revised manuscript; TJ hybrids contained *japonica* cytoplasm but most of them were from *indica-indica* crosses of two-line system). Figure 2d in the revised manuscript indicated the Wild-abortive (WA) and TJ hybrids had differentiated genetic structure. And Figure 2e-h in the

revised manuscript demonstrated the TJ hybrids had higher *japonica*-introgression levels, and the introgression events introduced breeding-favorable alleles widely-used in *japonica* subspecies, such as *NAL1*(Os04g0615000)^{6,7} and *GW3p6* (Os03g0215400)²⁷⁻²⁹. Thus, we concluded that the genetic diversity of the early-bred rice hybrids was relatively small. With generation and wide cultivation of additional types of hybrids, such as TJ hybrids, the genetic diversity increased. During the process, the advantageous genetic resource was also introduced. We reorganized the text for better presentation of our view in the revised manuscript (Line 194-197).

And description at lines 163-164 (in previous manuscript: *The ratio of introgression was not completely equal to superior allele frequency due to that not all of the indica/japonica introgression associated with superior allele introduction*) was an explanation for the inequality between the frequencies of *indica/japonica* introgression and *favorable/unfavorable* allele in the pie and bar plots shown in Figure 2h. And we have placed the explanation in Figure legend of Figure 2h in the revised manuscript to avoid misunderstanding (Line 1038-1041).

#8. (Line 281–288) *Please provide comparison of genomic selection model accuracy between additive and additive plus dominant model (i.e., genomic selection model with A and A+D). One of the most interesting points of genomic selection of hybrids is whether inclusion of non-additive genetic variable can increase prediction accuracy. In many cases, additive model shows higher prediction accuracy than additive plus dominant effect model (e.g., reviewed in Varona et al. 2013). If additive plus dominant effect model shows higher prediction accuracy than additive model, it provides evidence that non-additive effects contribute to phenotypic variation in the hybrids and may contribute to hybrid rice breeding.*

* Varona et al. 2013. doi: 10.3389/fgene.2018.00078

Answer: It is good suggestion. Actually, we conducted a comparison of genomic selection model accuracy between additive and additive plus dominant model as you suggested. There was no obvious improvement of accuracy for additive plus dominant model in comparison with additive model. In our previous⁹ and current works (Fig. 6 in response), we found the support for positive partial dominance and over-dominance effects for heterosis of yield traits. Consequently, we thought non-additive effects contributed to hybrid rice breeding. The major reasons for no obvious improvement of accuracy when inclusion of non-additive genetic variable in model construction, might be as follows:

i) The magnitude of dominance effect varied continuously across different loci. When using d/a to measure the magnitude of dominance effect (Fig. 14 in response), the d/a value for complete dominance effect was 1 or -1, ranged from -1 to 1 for the partial dominance effect, and was larger than 1 or smaller than -1 for the over- or under-dominance effect, respectively. In contrast, the additive effect, with the d/a value of 0 (the phenotypic value for heterozygote was equal to the average of both homozygotes), was easier to capture.

ii) Furthermore, including non-additive genetic effects might increase the complexity of the model and it was more difficult to accurately estimate the weight of markers with non-additive effects.

Current models for genomic selection were inadequate in handling the non-dominance effect, and the community will need more time to construct a new model to address this issue.

Fig. 14 | The partition of various dominance effect measured by d/a value.

#9. (Fig. 4d) A more detailed caption is needed for Fig. 4d. What does the gray shaded area mean? Are the definitions of the blue, yellow, and red shaded areas the same as in Figure 4c?

Answer: The gray shaded area represented chromosomes for Fig. 4d. In each gray shaded region, the lower stripes represented SNPs along the hypothetical chromosomal region, and higher bar indicated genomic sequence with its origin judged by the SNP markers distributed in it (Details seeing “Genome-wide identification of introgression from the *japonica* ancestry” section in Method). In Figure 4 in the revised manuscript, SNP and genomic sequence judged as homozygous *indica*-origin were marked in red, judged as homozygous *japonica*-origin were indicated by blue, and judged as heterozygous genotypes (*indica/japonica*) were represented by yellow.

For better understanding, detailed caption (Line 1069-1072) for Figure 4d as well as figure labels for Figure 4c were supplied in the revised manuscript.

#10. (Fig 4d, line 258-264) Please describe the method to determine paternal and maternal allele of the F₂ individuals

Answer: As you advised, the pipeline about determining the genotype of F₂ individuals was described in the revised manuscript (Line 762-782). Genotyping of F₂ individuals was conducted largely as described by us previously¹⁹. The average sequencing depth for F₂ individuals was 0.2×, and for both parental lines of F₂ population was 30×. High-quality SNPs for paternal lines were detected according to the pipeline described in the “Sequencing and variation calling” section above. We then created the *pseudo*-reference sequence for both parental lines by replacing the Nipponbare bases with the corresponding bases at SNPs identified in both parents. We aligned the paired-end reads of F₂ individuals to the *pseudo*-sequence of both parents using BWA (version 0.7.1)²², and implemented a customized Perl script to determine the origin (paternal or maternal) of reads according to the SNPs within the reads. Then we took a sliding-window approach to call genotypes and determine recombination breakpoints: a self-adaptation sliding window with step size 1 (read) moved from left to right. The window size was adjusted according to the marker density: the sliding window was composed of 15 reads when the genome-wide SNPs were <5000; 25 reads for genome-wide SNPs of 5000-10000; 39 reads for genome-wide SNPs >10000-20000; 59 reads for genome-wide SNPs >20000-100000; 99 reads for genome-wide SNPs >100000. Within each window, the numbers of maternal-origin and paternal-origin reads were counted. The genomic region covered by the sliding window was called as homozygous maternal genotype when >80% reads were judged as maternal origin; called as homozygous paternal genotype when >80% reads were paternal origin; otherwise called as heterozygous genotype. Thus, as the window slid, genotypes were called and recombination breakpoints were determined. Finally, we could obtain a bin map for each F₂ individual, the results could be exhibited as below:

Fig. 15 | The recombination map of a single F₂ line.

In the picture, the gray area represented hypothetical chromosomes. In each gray shaded region, the lower stripe represented sequenced reads along the hypothetical chromosomal region, and the higher bar indicated genomic sequence with its genotype called by sliding window. Reads and genomic sequence judged as homozygous maternal origin were marked in red, as homozygous paternal origin were marked by blue, and as heterozygous genotype were marked by yellow.

Due to that we judged the origin (paternal or maternal) of genomic blocks for F₂ lines, we could also infer the genotype of target loci according to the information from both parental lines.

Figure 4d in the revised manuscript illustrated the distribution of *indica*- or *japonica*-origin segments across the whole genome of the interspecific hybrid Quanjingyou No.1. The 830,245 *indica-japonica* differentiated SNPs were used for judging the *indica*- or *japonica*-origin of genome-wide segments in accordance with the pipeline described above with several modifications: i) There was no need to construct *pseudo*-reference and align sequencing reads back to the *pseudo*-reference; instead, the SNPs from the hybrid were judged as *indica/indica*, heterozygous (*indica/japonica*), or *japonica/japonica* genotype.

ii) The *indica*- or *japonica*-origin of segments was identified by a SNP-based sliding window rather than the read-based window. More details were described in “Genome-wide identification of introgression from the *japonica* ancestry” section of the revised manuscript (Line 729-761)

#11. (Line 259 and 261) Does the “dominance effect” means the dominance effect estimated in the linear mixed model implemented in PLINK, or *d/a* that the authors used to represent impact on dominance effect in elsewhere (e.g. Fig. 3g)?

Answer: The “dominance effect” means the magnitude of dominance effect (*d/a* value), which was estimated according to a previously reported method^{3,4} for associated loci identified by GWAS: The average phenotype of individuals with heterozygous genotype and both homozygous genotypes were calculated to estimate the *d/a* value. The peak SNPs in association signals were used for genotyping and calculation of the index of *d/a* according to the relationships:

$$d = F_1 - \frac{P_1 + P_2}{2}$$

$$a = P_1 - \frac{P_1 + P_2}{2}$$

where F_1 , P_1 and P_2 were average phenotypic value for the heterozygous and both homozygous genotypes, respectively, and P_1 was the advantageous phenotypic value. And the SNP site, in which the number of individuals with either heterozygous or homozygous genotypes was fewer than 10, was excluded in the calculation of “ d/a ”.

For quantitative trait loci (QTLs) mapped in F_2 population using a composite interval-mapping method, the d/a value was estimated by the software IciMapping⁵.

To avoid ambiguity, the “dominance effect” indicating the “magnitude of dominance effect” was replaced by “**dominance effect/additive effect (d/a)**” or “the magnitude of dominance effect” in the revised manuscript (Line 229, 271, 815, 818, 1055 and 1073). For better understanding, we also added detailed description for “partial dominance effect” and “negative partial dominance effect” at the point of their initial introduction (Line 63 and 252).

#12. (Line 265) *I have concerns about definition of terms related to genomic selection. The authors used “phenotype prediction” (e.g. line 269), but since the model in line 692 does not include environmental effects, it is not appropriate to use “phenotype prediction”. The authors should use “genomic prediction” or “genomic selection” in this case. Besides, genomic selection model does not predict “the phenotype” but “the genomic estimated breeding value” because phenotypes are affected by environmental conditions. Please confirm definition of the terms.*

Answer: Very good suggestions. Thanks a lot for your reminder. We have confirmed the definition of the terms, and replaced “phenotype prediction” by “genomic selection (GS)” in the revised manuscript (Line 81, 326, 338, 339, 385, 399, 885, 933, 1078 and 1080). Furthermore, “the phenotype” or “the predicting phenotype” was also replaced by “the genomic estimated breeding value (GEBV)” as you suggested (Line 342, 362, 394, 911, 913, 914, 918, 919, 927 and 1090).

#13. (Line 327) *What does “the predicting phenotype” mean? Does it mean predicted phenotype value from the genomic selection model?*

Answer: Yes, “The predicting phenotype” means the predicted phenotype value from the genomic selection model. And we have replaced it by GEBV as you suggested in Question #12. We aimed to show the association between the accumulated breeding-favorable alleles and the GEBV value in *pseudo* hybrid combinations (Fig. 6 in the revised manuscript), and further illustrated the pyramiding more breeding-

favorable alleles contributed to better performance of hybrid combinations, which was also observed in improvement breeding of rice hybrids (Fig. 3 in the revised manuscript).

#14. (Line 336-338) Please provide a more detailed theoretical explanation of this message. This message is interesting and attractive, but I could not understand why the authors reached this indication. Is it because the genomic selection model included seed setting rate? Without a statement on this point, the result for *rf3* seems a coincidence, and could be misleading to readers of the manuscript.

Answer: Thanks for your suggestions. The ability of the genomic selection model to screen out combinations with crossing incompatibility was indeed due to the integration of seed setting rate (Fig. 6e, f in the revised manuscript). One of the reasons for crossing incompatibility was the absence of the pollen-fertility-restoring genes in paternal line, which resulted in the failure to restore the fertility of maternal line. Thus, F₁ combinations had low seed setting rate. In the revised manuscript, we analyzed the proportion of combinations containing sterile genotypes of *rf3* (Os01g0201700) and *rf4* (Os10g0495200) responsible to wild abortive sterility as well as *tms5* (Os02g0214300) contributing to thermosensitive male sterility in high-scoring, random and low-scoring subgroups (Fig. 16 in response). Fewer high-scoring F₁ combinations contained the homozygous genotypes causing pollen sterility, in comparison to randomly selected and bottom-scoring combinations. The analysis for *rf4* and *tms5* as well as a clearer explanation were included in the revised manuscript (Fig. 6g and Line 399-408 in the revised manuscript) to further show the ability of the model to screen out fertility-unrestorable combinations.

Fig. 16 | The proportion of combinations possessing genotypes causing pollen abortion for genes *rf3*, *rf4* and *tms5* in top-scoring, randomly selected and bottom-scoring subgroups. The causative variations of *rf3* (Os01g0201700), *rf4* (Os10g0495200) and *tms5* (Os02g0214300) reported by Wei⁸ were used for analysis.

In the revised manuscript, all 58,353 possible *pseudo*-combinations generated by 367 paternal lines and 159 maternal lines were used for analysis demonstrated in Figure 6.

#15. (Line 565-577) How many biological replicates were prepared for grain related traits?

Answer: For six grain yield-related traits, comprising full grain number per plant, valid panicle number, full grain number per panicle, kilo-grain weight, seed setting rate and yield per plant, three biological replicates were prepared for each F₁ accession. With respect to grain quality-related traits, comprising amylose content, gel consistency, chalkiness, chalky grain percentage, grain translucency and grain shape (length to width ratio of polished grain), grains from mixed harvest were randomly selected to conduct phenotype investigation, with two replicates prepared for each sample.

With respect to F₂ individuals, there was no replicate for all traits because F₂ individuals have different genotypes.

As advised, we have provided a clearer description for replicates prepared for different traits in Methods (Line 664 and 674-676).

#16. *(Fig. 2b, d) Please provide the PVE of each PC. This information is important for estimating strength of the population structure.*

Answer: Thanks for your reminder. The variances explained by three PCs were marked in the labels of x and y axis for Figure 2b and d in the revised manuscript.

#17. *(Line 628) Please provide criteria for selecting 19 indica and 23 japonica landraces to define the indica and japonica loci. Were the landraces selected based on a population structure analysis or results from other studies? Because some studies suggested ancient introgressions between indica and japonica, definition of japonica-indica haplotype is a sensitive problem. I am not asking the authors to perform precise japonica-indica definition. I think adding descriptions explaining the criteria would be sufficient and the readers can judge the validity and the reliability by themselves based on their own knowledge.*

Answer: The standards of landrace selection were based on the population structure analysis from Zhao's work². They reported the ecotypes of sixty-six accessions. Among them, nineteen accessions of *O. sativa indica* and twenty-two of *O. sativa temperate japonica* were chosen. Furthermore, an additional accession of *O. sativa temperate japonica* (named V1) sequenced by us was also included for *indica-japonica* differentiated SNP identification (Fig. 17 in response). In total, nineteen *indica* and twenty-three temperate *japonica* accessions were selected for analysis (Table 3 in response). The criteria for selecting

the samples were provided in Methods of the revised manuscript (Line 730-735 and Extended Data Fig. 23).

Fig. 17 | The Neighbor-joining tree of the 42 rice accessions. The tree was constructed using whole-genome data. The accessions belonging to *O. sativa indica* and temperate *japonica* subgroups were respectively indicated by red and blue.

Table 3. The sample ID and corresponding ecotypes for the 42 samples used for *indica-japonica* differentiated SNP identification²

Sample ID	Ecotype	Sample ID	Ecotype
GP3	O. sativa indica	HP362-2	O. sativa indica
GP22	O. sativa indica	HP383	O. sativa indica
GP51	O. sativa indica	HP396	O. sativa indica
GP72	O. sativa indica	HP407	O. sativa indica
GP772-1	O. sativa indica	HP486	O. sativa indica
HP119	O. sativa indica	HP492	O. sativa indica
HP263	O. sativa indica	HP577	O. sativa indica
HP274	O. sativa indica	GLA4	O. sativa indica
HP327	O. sativa indica	W0128	O. sativa indica
HP517-1	O. sativa indica	V1	O. sativa temperate japonica
GP551	O. sativa temperate japonica	HP98	O. sativa temperate japonica
GP567	O. sativa temperate japonica	HP103	O. sativa temperate japonica
GP669	O. sativa temperate japonica	HP314	O. sativa temperate japonica

GP677	O. sativa temperate japonica	HP390	O. sativa temperate japonica
HP13-2	O. sativa temperate japonica	WYG7	O. sativa temperate japonica
HP14	O. sativa temperate japonica	KY131	O. sativa temperate japonica
HP38	O. sativa temperate japonica	LG31	O. sativa temperate japonica
HP44	O. sativa temperate japonica	DHX2	O. sativa temperate japonica
HP45	O. sativa temperate japonica	IL9	O. sativa temperate japonica
HP48	O. sativa temperate japonica	Koshihikari	O. sativa temperate japonica
HP91-2	O. sativa temperate japonica	UR28	O. sativa temperate japonica

#18. (Line 646 and Extended Data Fig. 16) The message seems to be inconsistent between the conclusions and the results. Specifically, Nipponbare genome is not composed of 100% Japonica SNPs, and the same is true for Shuhui498. I agree with validity of the method and the results for this analysis, but use of “100%” (line 646) and “all” (Extended Data Fig. 16) is inappropriate from the results. I ask the authors to remove intuitive description and to include descriptions based on the results.

Answer: As you said, there were 0.03% SNPs (out of the 830,245 differentiate SNPs identified in our study) from *japonica* cultivar Nipponbare in *indica/indica* state, and 0.03% of SNPs from *indica* cultivar Shuhui498 in *japonica/japonica* state. We stated it in the figure legend of Extended Data Figure 24 in the revised manuscript. We used “all” and “100%” to describe that all blocks or 199-SNP length segments in Nipponbare were judged as *japonica*-origin and all blocks in Shuhui498 were *indica*-origin according to our method. To facilitate understanding, original text was changed into “Among all the 830,245 loci, 99.97% of SNPs from cultivar Nipponbare were judged as of *japonica*-origin and all the 199-SNP length fragments across the whole genome were considered to be of *japonica*-lineage. For cultivar Shuhui498, 95.39% of SNPs were judged as of *indica*-origin and all the 199-SNP length fragments were identified as of *indica*-lineage” at lines 751 to 755. Furthermore, figure legend of Extended Data Figure 24 was also modified for better understanding.

#19. (Line 658) Is it really covariance matrix? I think it is the covariate matrix. Please confirm.

Answer: Thanks for your careful review. It was the covariate matrix, and we have corrected it in the revised manuscript (Line 792).

#20. (Line 664) If the authors used permutation test to define significant threshold, then the value should vary by trait. Therefore, I am bit skeptical about the GWAS significant threshold in this study. Please provide the actual significant threshold value for each trait (e.g., heading date: 6.03, flag leaf length: 6.01, and so on).

Answer: Thanks for your reminder. We have not described clearly how we set the significant threshold in the method section. The significant cutoff was defined according to previous report⁴: permutation test was used to help define the genome-wide significant threshold. For the rice hybrid cohort, we randomly picked ten traits (comprising heading date, tiller angle, leaf width, plant height, full grain number per panicle, kilo-grain weight, amylose content, gel consistency, grain translucency and grain shape), reshuffled the original phenotype data, and then performed association analysis. A total of 100 permutation tests was performed and four association signals passing through the significant threshold 10^{-6} were detected. This suggested a feasible FDR (false discovery rate) level of <0.05 when the whole-genome significant cutoff was set as 10^{-6} . With respect to the F_2 cohort, all seven traits of F_2 individuals from *indica-indica* hybrids (heading date, leaf length, leaf width, tiller angle, full grain number per panicle, full grain number per plant and total grain number per panicle) were taken into consideration when we conducted permutation tests. A total of 70 permutation tests was performed, and only one association signal passed through the threshold 10^{-6} , indicating FDR level of <0.05 . Thus, significant threshold was set as 10^{-6} for both F_1 and F_2 cohorts. We described the procedure clearly in Methods of the revised manuscript (Line 798-811).

#21. (Line 670). *Is the calculation method for the index appropriate? Generally, in a population with population structure, average phenotype value is not appropriate for estimating effect of each SNP because the average phenotype includes effect of polygenic background effects. Since the authors used PLINK to perform the GWAS, I recommend the authors to use the estimated SNP effect to account for background polygenic effects.*

Answer: The method for estimating the dominance-effect/additive-effect (d/a) was referred to a previously reported method^{3,4}. The index was used for evaluating the magnitude of dominance effect. A locus with index d/a estimated at 0 indicated that the heterozygous genotype showed additive effect; a locus with index $d/a \geq 1$ indicated that the heterozygous genotype had overdominance effect; a locus with index $d/a \leq -1$ indicated that the heterozygous genotype represented underdominance effect; a locus with index $0 < d/a < 1$ indicated the heterozygous genotype had positive partial dominance effect; a locus with index $-1 < d/a < 0$ indicated the heterozygous genotype showed negative partial dominance effect. The average phenotypes of individuals with a heterozygous genotype and both homozygous genotypes were calculated to estimate the index. The peak SNPs in significant association signals were used for genotyping and calculation of the index of d/a according to the relationships:

$$d = F_1 - \frac{P_1 + P_2}{2}$$

$$a = P_1 - \frac{P_1 + P_2}{2}$$

where F_1 , P_1 and P_2 were average phenotypic value for the heterozygous and both homozygous genotypes, respectively, and P_1 was the advantageous phenotypic value. The peak SNP, in which the

number of individuals with either heterozygous or homozygous genotypes was fewer than 10, was excluded for further analysis.

Genome-wide association analysis was conducted by the mixed model implemented in EMMAX software package (version emmax-beta-07Mar2010)³⁰. To effectively account for population structure, principal component analysis was performed for the hybrid population and the first two principal components were incorporated as the covariates.

Because we focused on the magnitude of dominance effect for target loci but not the single peak SNPs, the effect of target loci was of interest. Furthermore, a single SNP may not capture the overall amount of variance at this locus according to previous report³¹⁻³³. Thus, we calculated the percent of phenotypic variance explained (PVE) by candidate regions surrounding the association signals in replace of estimating the single peak SNP effect. The procedure about calculating the variance explained by target region was referred to previous report³⁴:

The candidate regions were the 1Mb region surrounding the genome-wide association signals. A mixed linear model with multiple random effect was applied to estimate the variance components using R package *sommer*^{35,36}. The model can be written as:

$$y = Lb + T_1u_1 + T_2u_2 + e$$

where y was the phenotypic value, b was a vector of fixed effects, and u_1 and u_2 were both vectors of random effect. L , T_1 , and T_2 were incidence matrices for b , u_1 and u_2 , respectively. e represented a matrix of residual effects and $e \sim N(0, I\sigma_e^2)$, in which I was an identity matrix and σ_e^2 was the residual variance. The variable $u_1 \sim N(0, k_1\sigma_{u_1}^2)$, in which k_1 represented the variance covariance matrix constructed by the SNPs within the candidate region and $\sigma_{u_1}^2$ was the variance of the candidate region. The variable $u_2 \sim N(0, k_2\sigma_{u_2}^2)$, in which k_2 represented the variance covariance matrix constructed by all SNPs across the whole genome except for the SNP set in target region and $\sigma_{u_2}^2$ was the variance of the remaining region of the whole genome. The genetic variance explained by the target region was estimated by formula $\frac{\sigma_{u_1}^2}{\sigma_{u_1}^2 + \sigma_{u_2}^2}$, and the proportion of PVE by the target region was the ratio between $\sigma_{u_1}^2$ and the phenotypic variance.

The method for PVE estimation has been described in Methods of the revised manuscript (Line 828-845). And loci with index d/a estimated also had their PVE evaluated (Line 226, 244, 248 and 264-269).

<Minor comments>

#22 (Line 251-251) *“Genetic complementary pyramided superior alleles in intraspecific hybrids” seems unnecessary. Perhaps, it is a subsection title?*

Answer: Thanks. We have removed the unnecessary sentence to improve the readability of the manuscript under your suggestion.

#23. (Line 628) *Perhaps, “Definition of indica-japonica differential loci” is a subsection title.*

Answer: Yes, it was a subsection title. And we have removed it to improve the readability of our manuscript.

#24. (Line 701 -702) *The authors used Z to represent marker genotype matrix. However, in line 692, Z is used for incident matrices. To avoid confusion, I recommend the authors to use other alphabets to represent marker genotype matrix. It is also recommended to use different alphabets for the marker genotype matrix in line 701 and 702 because line 701 is an additive genotype value and line 702 is a heterozygous genotype value. (e.g., W for additive and H for heterozygote genotype value).*

Answer: Thanks a lot for your suggestions. In the revised manuscript, *W* was used for the scaled marker matrix when we calculated the additive relationship matrix *A*, and *H* was used for the scaled marker matrix when calculated dominance relationship matrix *D* (Line 901-904).

#25. (Line 710) *Perhaps, “Selection index calculation” is a subsection title.*

Answer: Yes, it was a subsection. There were three subsections for the section “Construction of decision-making model for hybrid breeding”. To improve the readability, we rewrote the subsection title as:

1. Training genomic selection model based on GBLUP (genomic best linear unbiased prediction) method. (Line 885).
2. Defining and calculating selection index. (Line 913).
3. Validating the design-making model in a new population. (Line 929).

#26. (Fig. 5g, 6a, 6c, 6e) Alphabets on the box and whisker plots would be unnecessary because there are only two categories in the figures.

Answer: Thanks a lot for your nice reminder. The alphabets on the box and whisker plots were removed for Fig. 5g, 6a, 6c and 6e in the revised manuscript.

And we have to apologize for the mistake that we gave the wrong p -value for comparison between the high-scoring and low-scoring combinations in Figure 5g (the boxplot was correct in previous manuscript). We confirmed the p -value and corrected the mistake in the revised manuscript.

Reference:

1. Gu, Z. et al. Cytoplasmic and nuclear genome variations of rice hybrids and their parents inform the trajectory and strategy of hybrid rice breeding. *Mol Plant* **14**, 2056-2071 (2021).
2. Zhao, Q. et al. Pan-genome analysis highlights the extent of genomic variation in cultivated and wild rice. *Nat Genet* **50**, 278-284 (2018).
3. Chen, J. et al. Genome-wide association analyses reveal the genetic basis of combining ability in rice. *Plant Biotechnol J* **17**, 2211-2222 (2019).
4. Huang, X. et al. Genomic analysis of hybrid rice varieties reveals numerous superior alleles that contribute to heterosis. *Nat Commun* **6**, 6258 (2015).
5. Meng, L., Li, H., Zhang, L. & Wang, J. QTL IciMapping: Integrated software for genetic linkage map construction and quantitative trait locus mapping in biparental populations. *The Crop Journal* **3**, 269-283 (2015).
6. Zhang, G.H. et al. LSCHL4 from Japonica Cultivar, which is allelic to NAL1, increases yield of indica super rice 93-11. *Mol Plant* **7**, 1350-1364 (2014).
7. Fujita, D. et al. NAL1 allele from a rice landrace greatly increases yield in modern indica cultivars. *Proc Natl Acad Sci U S A* **110**, 20431-20436 (2013).
8. Wei, X. et al. A quantitative genomics map of rice provides genetic insights and guides breeding. *Nat Genet* **53**, 243-253 (2021).
9. Huang, X. et al. Genomic architecture of heterosis for yield traits in rice. *Nature* **537**, 629-633 (2016).
10. Crossa, J. et al. Genomic Selection in Plant Breeding: Methods, Models, and Perspectives. *Trends Plant Sci* **22**, 961-975 (2017).

11. Li, C. et al. Genomic insights into historical improvement of heterotic groups during modern hybrid maize breeding. *Nat Plants* **8**, 750-763 (2022).
12. Wang, B. et al. De novo genome assembly and analyses of 12 founder inbred lines provide insights into maize heterosis. *Nat Genet* **55**, 312-323 (2023).
13. Cingolani, P. et al. A program for annotating and predicting the effects of single nucleotide polymorphisms, SnpEff: SNPs in the genome of *Drosophila melanogaster* strain w1118; iso-2; iso-3. *Fly* **6**, 1-13 (2012).
14. Hu, Z., Tian, Y. & Xu, Q. The analysis of history and current situations for Chinese hybrid rice development. *Hybrid Rice* **31**, 1-8 (2016).
15. Qian, Q., Guo, L., Smith, S.M. & Li, J. Breeding high-yield superior quality hybrid super rice by rational design. *National Science Review* **3**, 283-294 (2016).
16. Yu, J. et al. A Draft Sequence of the Rice Genome (*Oryza sativa* L. ssp. indica). *Science* **296**, 79-91 (2002).
17. Goff, S.A. et al. A Draft Sequence of the Rice Genome (*Oryza sativa* L. ssp. japonica). *Science* **296**, 92-100 (2002).
18. International Rice Genome Sequencing, P. The map-based sequence of the rice genome. *Nature* **436**, 793-800 (2005).
19. Huang, X. et al. High-throughput genotyping by whole-genome resequencing. *Genome Res* **19**, 1068-1076 (2009).
20. Zeng, D. et al. Rational design of high-yield and superior-quality rice. *Nat Plants* **3**, 17031 (2017).
21. Guo, L.-b. & Ye, G.-y. Use of Major Quantitative Trait Loci to Improve Grain Yield of Rice. *Rice Science* **21**, 65-82 (2014).
22. Li, H. & Durbin, R. Fast and accurate long-read alignment with Burrows-Wheeler transform. *Bioinformatics* **26**, 589-595 (2010).
23. Duan, P. et al. Natural Variation in the Promoter of GSE5 Contributes to Grain Size Diversity in Rice. *Mol Plant* **10**, 685-694 (2017).
24. Shomura, A. et al. Deletion in a gene associated with grain size increased yields during rice domestication. *Nat Genet* **40**, 1023-1028 (2008).
25. Liu, J. et al. GW5 acts in the brassinosteroid signalling pathway to regulate grain width and weight in rice. *Nat Plants* **3**, 17043 (2017).
26. Gong, J. et al. Dissecting the Genetic Basis of Grain Shape and Chalkiness Traits in Hybrid Rice Using Multiple Collaborative Populations. *Mol Plant* **10**, 1353-1356 (2017).

27. Wang, C. et al. Dissecting a heterotic gene through GradedPool-Seq mapping informs a rice-improvement strategy. *Nat Commun* **10**, 2982 (2019).
28. Liu, Q. et al. G-protein betagamma subunits determine grain size through interaction with MADS-domain transcription factors in rice. *Nat Commun* **9**, 852 (2018).
29. Yu, J. et al. Alternative splicing of OsLG3b controls grain length and yield in japonica rice. *Plant Biotechnol J* **16**, 1667-1678 (2018).
30. Kang, H.M. et al. Variance component model to account for sample structure in genome-wide association studies. *Nat Genet* **42**, 348-354 (2010).
31. Galarneau, G. et al. Fine-mapping at three loci known to affect fetal hemoglobin levels explains additional genetic variation. *Nat Genet* **42**, 1049-1051 (2010).
32. Sanna, S. et al. Fine mapping of five loci associated with low-density lipoprotein cholesterol detects variants that double the explained heritability. *PLoS Genet* **7**, e1002198 (2011).
33. Yang, J. et al. Conditional and joint multiple-SNP analysis of GWAS summary statistics identifies additional variants influencing complex traits. *Nat Genet* **44**, 369-375 (2012).
34. Tang, Z. et al. Genome-Wide Association Study Reveals Candidate Genes for Growth Relevant Traits in Pigs. *Front Genet* **10**, 302 (2019).
35. Covarrubias-Pazaran, G. Genome-assisted prediction of quantitative traits using the R package sommer. *PLOS ONE* **11**, e0156744 (2016).
36. Covarrubias-Pazaran, G. Software update: Moving the R package sommer to multivariate mixed models for genome-assisted prediction. doi: <https://doi.org/10.1101/354639> (2018).

Decision Letter, first revision:

3rd May 2023

Dear Bin,

Your Article, "Structure and function of rice hybrid genomes reveal genetic basis and optimal performance of heterosis" has now been seen by the original 3 referees. You will see from their comments below that while they continue to find your work of interest, there is still an important point that requires resolution. We remain interested in the possibility of publishing your study in Nature Genetics, but would like to consider your response to these concerns in the form of a revised manuscript before we make a final decision on publication.

In brief, Reviewers #1 and #2 are now satisfied and supportive of publication. Referee #1 has a few minor comments that do not require further review.

Reviewer #3 acknowledges the improvement, but has a few comments remaining; in our reading of these reports, the comment on the dominance effect estimation used (not taking into account the genetic background) is especially concerning given the importance of dominance to a study of heterosis. However, the referee also provides some guidance, so we believe this should be addressable.

To guide the scope of the revisions, the editors discuss the referee reports in detail within the team, including with the chief editor, with a view to identifying key priorities that should be addressed in revision and sometimes overruling referee requests that are deemed beyond the scope of the current study. We hope that you will find the prioritized set of referee points to be useful when revising your study. Please do not hesitate to get in touch if you would like to discuss these issues further.

We therefore invite you to revise your manuscript taking into account all reviewer and editor comments. Please highlight all changes in the manuscript text file. At this stage we will need you to upload a copy of the manuscript in MS Word .docx or similar editable format.

*2) If you have not done so already please begin to revise your manuscript so that it conforms to our Article format instructions, available [here](http://www.nature.com/ng/authors/article_types/index.html). Refer also to any guidelines provided in this letter.

[redacted]

We hope to receive your revised manuscript within four to eight weeks. If you cannot send it within this time, please let us know.

Sincerely,

Michael Fletcher, PhD
Senior Editor, Nature Genetics

ORCID: 0000-0003-1589-7087

Reviewers' Comments:

Reviewer #1:

Remarks to the Author:

This reviewer appreciates the careful and thorough responses to my previous comments and suggestions. All of my comments are satisfactorily addressed. I believe that this work will be a great contribution to our improved understanding of heterosis and more efficient hybrid rice breeding.

I have two minor suggestions for the authors to consider (no need for re-review):

1. The paragraph describing the five groups of hybrids (line 155-165) might be better moved to in front of the paragraph describing phenotyping results (line 137-154)?
2. In line 184-194, please specify the introgression of japonica favourable alleles into WA and TJ hybrids heterozygous or homozygous introgression (hard to tell from Figure 2h). It will be interesting to know if the increased introgression of japonica segments (genes) into WA and TJ contributed to heterosis in the respective hybrids from Y1 to Y3, particularly in the inter-specific hybrids? What could be their action modes? Just out of curiosity, some explanation will be suffice.

Haiyang Wang

Reviewer #2:

Remarks to the Author:

The revised manuscript is much improved, and addresses my previous concerns. The authors have done quite a lot of work to address both my comments and those of the other referees. I still wish that there was more molecular genetic characterization of heterosis and heterotic loci, but I appreciate that this very large study does advance our understanding of this important phenomena.

Reviewer #3:

Remarks to the Author:

I acknowledge the effort that has gone into this revised manuscript. This version of the manuscript reads much better than the previous one. I have some additional comments on the revised manuscript. The numbers on the following comments (#n) correspond to comment number from my previous comments.

< #2 (Genotyping methods for F2. Line 107-108 in the revised manuscript.) >

I understand the genotyping method and its reliability. However, saying "F2 individuals were sequenced at an average depth of 0.2-fold" gives the impression that the experiment was insufficient. I suggest the authors to modify this description. E.g., "F2 individuals were sequenced at an average depth of 0.2-fold, and then the genotype of each F2 individual were determined as in previous study¹⁹" (Since I am not an English native, please check if it is grammatically correct by yourself).

< #8 (Genomic selection models. Fig. 5.) >

I agree the comment from the authors. However, even if the additive plus dominance model did not improve the prediction accuracy, I recommend the authors to show the results for dominance model elsewhere (e.g., Supplementary Figure) and provide brief description for the discussion because i) this is a fundamental question in genomic selection work for hybrids, and ii) the message included in the authors' discussion was important for the question (i.e. increasing accuracy with dominance model is difficult even if we use a big data such as the data provided in this study).

< #11 and #21 (Dominance effect estimation) >

Now I understand that the method for dominance effect estimation was same with the previous study, and this time I accept the authors' opinion. However, in genetically heterogeneous populations, estimation of a locus effect without background effect correction results in biased estimates. This is the reason why most of GWAS apply linear mixed model (LMM), and therefore, the results of the previous study may also contain bias. More specifically, to estimate dominance effects, LMM must be as follow:

$$Y = XB + Aa + Dd + g + e$$

where Y is a vector of phenotype, A is a vector of marker genotype values for additive effect, D is a vector of marker genotype values for dominant effect, and g is a vector of random effect for background genetic effects. As far as I read the manuscript, dominance effect estimation method used in this study can be explained as

$$Y = Aa + Dd + e.$$

Clearly, as the influence of background genetic effect (g) increases, the estimation of dominance effect (d) is biased. As far as I know, EMMAX only considers additive effect, while TASSEL provide estimation of dominance effects.

As for the authors' comment on #21 "Furthermore, a single SNP may not capture the overall amount of variance at this locus according to previous report³¹⁻³³", if it is possible to define haplotype or allelic state of each locus, GWAS based on haplotypes will provide accurate dominance effect estimation. If it is not possible to define haplotype, Sequential Kernel Association Test (SKAT) will provide more appropriate dominance effect estimation. For the above approaches, more flexible platforms such as LIMMIX and BGLR are helpful. I recommend the authors to use the above approaches in their future activities.

As for dominance effect estimation in F2 QTL mapping, I think there is no problem.

<Additional comments>

(Line 828-845) If the authors intended to focus on the magnitude of dominance effect for the target loci, please provide method to construct covariance matrix k1 and k2. A.mat considers additive effect while D.mat considers dominance. If the authors used only A.mat, please provide rationale for that because the main target of this study is "hybrid" and therefore, ignoring D.mat seems strange.

(Fig. 3g) Please provide caption for x-axis on Fig. 3g.

Author Rebuttal, first revision:

Response to referees:

Response to Reviewer #1

Remarks to the Author:

This reviewer appreciates the careful and thorough responses to my previous comments and suggestions. All of my comments are satisfactorily addressed. I believe that this work will be a great contribution to our improved understanding of heterosis and more efficient hybrid rice breeding.

Answer: Thanks for your comments.

I have two minor suggestions for the authors to consider (no need for re-review):

1. The paragraph describing the five groups of hybrids (line 155-165) might be better moved to in front of the paragraph describing phenotyping results (line 137-154)?

Answer: As advised, the paragraph describing the classification of hybrids (Lines 134-143) was moved to in front of the paragraph describing phenotypic change during improvement breeding (Lines 144-161). And we think this modification augments the lucidity and logical consistency inherent of the text.

2. In line 184-194, please specify the introgression of japonica favourable alleles into WA and TJ hybrids heterozygous or homozygous introgression (hard to tell from Figure 2h).

Answer:

We displayed the frequency of homozygous and heterozygous genotypes for four candidate genes (*GW3p6*¹⁻³, *NALI*^{4,5}, *GS6*⁶ and *Waxy*^{7,8}) involved in introgression events in the bar plots of Figure 2h. The figure legend might be ambiguous for Figure 2h in the previous manuscript, and we have rewritten it to facilitate understanding (Lines 1041-1045): the bar plots demonstrated the genotypic frequencies of the four candidate genes, and the putative causal polymorphisms marked at gene body were used to define three genotypes, comprised breeding-favorable (blue) and -unfavorable (green) homozygous genotypes as well as heterozygous (orange) genotype.

In addition, we provided a detailed explanation of the occurrence of breeding-favorable allele introduction through heterozygous or homozygous introgression for four candidate genes in both WA and TJ populations (Lines 189-194):

For *GW3p6* and *NALI*, majority WA and TJ hybrids exploited breeding-favorable alleles by heterozygous introgression. For *GS6*, both heterozygous and homozygous introgression with favorable allele introduction were common in TJ hybrids, while heterozygous introgression was dominant in WA hybrids. For *waxy*, the occurrence of homozygous introgression with favorable allele introduction was frequent in TJ hybrids while heterozygous introgression was dominant in WA hybrids.

It will be interesting to know if the increased introgression of japonica segments (genes) into WA and TJ contributed to heterosis in the respective hybrids from Y1 to Y3, particularly in the inter-specific hybrids? What could be their action modes? Just out of curiosity, some explanation will be suffice.

Answer: Because *Waxy* was resided at the genomic region with high introgression level in both WA and TJ populations (Fig. 2h in revised manuscript) and had its breeding-favorable alleles originated from *japonica* subspecies⁹, we took it as an example and analyzed the frequency of three genotypes in WA and TJ hybrids from three breeding periods. As shown in **Figure 1 in response**, the frequency of homozygous breeding-favorable genotype was increased from Y1 to Y3 periods in both WA and TJ hybrids. Upon

examination of Figure 3 in revised manuscript, the heterozygous genotype of *Waxy* exhibited a negative dominance effect for grain cooking and appearance quality, and using heterozygous genotypes of such loci in F₁ generations could compromise the performance. Thus, we think the strategic exploitation of breeding-favorable homozygous genotype of *Waxy* by introgression contributed to improvement breeding in both WA and TJ hybrids.

The analysis of introgression of *japonica* segments (genes) was based on *indica-indica* WA and TJ hybrids, and inter-specific (*indica-japonica*) hybrids was not included for analysis. As for inter-specific hybrids, genetic complementation was prevalent across the whole genome, with *indica*- and *japonica*-lineage loci accumulated in F₁ generations. And those loci exhibiting positive partial dominance and over-dominance effects, contributed to heterosis (Fig. 4 and Extended Data Fig. 16 in revised manuscript).

Fig 1 | Genotypic frequencies of *Waxy* across three breeding periods in both WA and TJ hybrids. The reported causal polymorphism (chr06:1765761-1765761) widely used in cultivated rice was used for genotyping. Breeding-favorable genotype was marked in blue, unfavorable genotype was in green and heterozygous genotype was indicated by orange.

Haiyang Wang

Response to Reviewer #2

Remarks to the Author:

The revised manuscript is much improved, and addresses my previous concerns. The authors have done quite a lot of work to address both my comments and those of the other referees. I still wish that there was

more molecular genetic characterization of heterosis and heterotic loci, but I appreciate that this very large study does advance our understanding of this important phenomena.

Answer: Many thanks for your comments. In our future work, we will make our effort to clone heterotic loci for dissecting molecular mechanisms of heterosis and quantifying genetic effects of heterotic loci.

Response to Reviewer #3

Remarks to the Author:

I acknowledge the effort that has gone into this revised manuscript This version of the manuscript reads much better than the previous one. I have some additional comments on the revised manuscript. The numbers on the following comments (#n) correspond to comment number from my previous comments.

Answer: Thanks a lot for your comments, which are very helpful for us to improve the quality of our manuscript. We have considered your suggestions seriously, and our replies for your advices and corresponding modifications are described as follows:

< #2 (Genotyping methods for F₂. Line 107-108 in the revised manuscript.) >

I understand the genotyping method and its reliability. However, saying “F₂ individuals were sequenced at an average depth of 0.2-fold” gives the impression that the experiment was insufficient. I suggest the authors to modify this description. E.g., “F₂ individuals were sequenced at an average depth of 0.2-fold, and then the genotype of each F₂ individual were determined as in previous study¹⁹” (Since I am not an English native, please check if it is grammatically correct by yourself).

Answer: As advised, original text was modified into “F₂ individuals were sequenced at an average depth of 0.2-fold, and then genotyping of each F₂ individuals was conducted largely as described by previous work¹⁰ (further details in Methods)” at lines 106-109. Furthermore, we provided the number of recombination bins identified in each F₂ population in Extended Data Table 1, to show the feasibility of genotyping method based on 0.2-fold sequencing data. The average bin size was estimated at $\sim 123.5\text{kb}$ ($\frac{373,245,519 \text{ base pairs (the genome size of reference IRGSP-1.0)}}{\text{total recombination bins of each population}}$).

Table 1. The size of 18 F₂ populations from 18 elite rice hybrids and the number of recombination bins identified in each F₂ population

Ecotype ^a	F ₂ Population	Size	Recombination bins
Indica-Indica	Pop1	498	3,093
	Pop2	340	2,716
	Pop3	452	3,072
	Pop4	357	2,665
	Pop5	357	2,676
	Pop6	399	2,916
	Pop7	365	2,959
	Pop8	725	3,155
	Pop9	418	3,157
	Pop10	586	2,955
Indica-Japonica	Pop11	621	3,017
	Pop12	949	3,235
	Pop13	529	2,964
	Pop14	690	3,131
	Pop15	808	3,327
	Pop16	659	3,329
	Pop17	465	3,023
	Pop18	621	3,202

a: Ecotypes for F_{1s} used to generate F₂ populations. “*Indica-indica*” indicated *indica* intraspecific cross; “*Indica-japonica*” represented interspecific cross.

< #8 (*Genomic selection models. Fig. 5.*) >

I agree the comment from the authors. However, even if the additive plus dominance model did not improve the prediction accuracy, I recommend the authors to show the results for dominance model elsewhere (e.g., Supplementary Figure) and provide brief description for the discussion because i) this is a fundamental question in genomic selection work for hybrids, and ii) the message included in the authors' discussion was important for the question (i.e. increasing accuracy with dominance model is difficult even if we use a big data such as the data provided in this study).

Answer: I appreciate for your constructive suggestions. I believe that there is a clerical error of “*show the results for **dominance model** elsewhere (e.g., Supplementary Figure)*”. I feel that you would like to ask us to show the prediction accuracy of additive model in the revised manuscript.

The comparison of prediction accuracy of additive model and additive plus dominance model was conducted and the result was added to the revised manuscript (Extended Data Fig. 18a in revised manuscript). And we provided a brief description of the result, and further discussed potential factors that

may elucidate the lack of improvement in the predictive accuracy of the genomic selection model upon the inclusion of dominant effect at lines 453-464:

We constructed a genomic selection model and set a selection index to predict the performance of hybrid rice and help to select hybrid combination with breeding potential. Given the significance of non-additive effects in hybrid rice heterosis¹¹, we used additive plus dominance model in genomic selection. However, the additive plus dominance model did not yield an improvement in prediction accuracy in comparison to the additive only model (Extended Data Fig. 18a). It has been reported that it is difficult to increase prediction accuracy by adding dominance in genomic selection models¹², and our work further support the opinion by using a big training data. The major cause might be: i) The magnitude of dominance effect varied across different loci, posing a challenge for reliable estimation and capture of the dominance effect; ii) inclusion of the dominance effect could elevate the model' complexity, hindering precise estimation of marker weights. Current models for genomic selection, including GBLUP model used in our work, were inadequate in handling the non-dominance effect, and the community will need more time to construct a new model to address this issue.

< #11 and #21 (Dominance effect estimation) >

Now I understand that the method for dominance effect estimation was same with the previous study, and this time I accept the authors' opinion. However, in genetically heterogeneous populations, estimation of a locus effect without background effect correction results in biased estimates. This is the reason why most of GWAS apply linear mixed model (LMM), and therefore, the results of the previous study may also contain bias. More specifically, to estimate dominance effects, LMM must be as follow:

$$Y = XB + Aa + Dd + g + e$$

where Y is a vector of phenotype, A is a vector of marker genotype values for additive effect, D is a vector of marker genotype values for dominant effect, and g is a vector of random effect for background genetic effects. As far as I read the manuscript, dominance effect estimation method used in this study can be explained as

$$Y = Aa + Dd + e.$$

Clearly, as the influence of background genetic effect (g) increases, the estimation of dominance effect (d) is biased. As far as I know, EMMAX only considers additive effect, while TASSEL provide estimation of dominance effects.

Answer: Thanks a lot for your constructive advice. In previous manuscript, the method employed for the estimation of *d/a* index did not incorporate a correction for background effects. As advised, we employed TASSEL software (version 5.0 Standalone)¹³ to perform GWAS analysis using a mixed linear model

(MLM). Subsequently, we calculated the d/a value based on the marker effects estimated by TASSEL. Thus, in the revised manuscript, the method for the estimation of d/a index was described as follows:

The dominance-effect/additive-effect (d/a) was calculated based on genotype effect estimated by TASSAL. The genotype effect of peak SNP in target locus was chosen to calculate d/a index:

$$a = \left| \frac{A - C}{2} \right|$$

$$d = M - \frac{A + C}{2}$$

where A and C respectively represented the genotype effects of two homozygous genotypes, and M represented the genotype effect of heterozygous genotype. For traits, where a high observed value was preferred in breeding practice (e.g., gel consistency), the index was estimated as d/a ; for traits, where a low observed value was preferred in breeding practice (e.g., chalkiness), the index was estimated as $-d/a$. If the number of observations of one of the homozygous genotype classes was fewer than 5, the estimation could be influenced by outliers and might be unreliable. Thus, a locus with the number of any genotype's observation less than 5 was excluded in the calculation of d/a index.

Due to the replacement of EMMAX with TASSEL for conducting GWAS in all three cohorts, the results based on GWAS analysis were subsequently updated based on the output from TASSEL, including Fig. 3, Fig. 4c, Fig. 6b, Fig. 6d, Fig. 6f, Extended Data Fig. 9, Extended Data Fig. 10, Extended Data Fig. 11, Extended Data Fig. 12, Extended Data Fig. 14 and Extended Data Fig. 15. The conclusions based on TASSEL was consistent with that from EMMAX.

In addition, it was important to note that in the absence of a permutation test, a significant threshold of 10^{-6} was set based on empirical values due to the substantial time required by TASSEL to complete a round of analysis for a single trait, typically lasting one week.

The updated methods for GWAS analysis and estimation for d/a index were described at lines 780 to 812 in the revised manuscript.

As for the authors' comment on #21 "Furthermore, a single SNP may not capture the overall amount of variance at this locus according to previous report31-33", if it is possible to define haplotype or allelic state of each locus, GWAS based on haplotypes will provide accurate dominance effect estimation. If it is not possible to define haplotype, Sequential Kernel Association Test (SKAT) will provide more appropriate dominance effect estimation. For the above approaches, more flexible platforms such as LIMMIX and BGLR are helpful. I recommend the authors to use the above approaches in their future activities.

Answer: Because a single SNP may not capture the overall amount of variance at a locus, we estimated the phenotypic variance explained by the 1Mb genomic region containing the target locus in current work.

Furthermore, as advised, we explored the method of Sequential Kernel Association Test and employed “fastBAT” module in GCTA software (version 1.26.0) to execute gene-based association test. We utilized gene annotation information from Nipponbare Reference IRGSP-1.0 to generate a gene list file, and the gene structure consisted of CDS, introns and UTRs. We conducted gene-based association test for several traits, comprising grain number per panicle, kilo-grain weight and plant height. As shown in Figure 2 in response, several known genes controlling corresponding traits were included in the top-ranking genes. Thus, we think that utilizing a multi-omics approach, combining genome-wide association analysis, gene-based association analysis, causal polymorphism analysis and expression analysis, will facilitate rapid identification of candidate genes in our future work. With candidate gene identified, it will enable us to precisely estimate effects of target genes using related genetic materials, such as NILs or transgenic materials.

Fig. 2 | Manhattan plots of gene-based association analysis. **a**, Grain number per panicle. **b**, Kilo-grain weight. **c**, Plant height. Known genes associated with corresponding trait were marked in Red. Association signal related with the newly identified locus *GNP3p1* was indicated by black arrow.

As for dominance effect estimation in F2 QTL mapping, I think there is no problem.

(Line 828-845) If the authors intended to focus on the magnitude of dominance effect for the target loci, please provide method to construct covariance matrix k_1 and k_2 . $A.mat$ considers additive effect while $D.mat$ considers dominance. If the authors used only $A.mat$, please provide rationale for that because the main target of this study is “hybrid” and therefore, ignoring $D.mat$ seems strange.

Answer: Thank you very much for your meticulous review. The dominance genetic variable was certainly ignored when we constructed the linear model to estimate the proportion of phenotypic variance explained (PVE) by candidate regions surrounding the association signals. We have incorporated the dominance variables of target and non-target regions as random effect T_2u_{d1} and T_4u_{d2} in the revised manuscript (Lines 815-841):

The model can be written as:

$$y = Lb + T_1u_{a1} + T_2u_{d1} + T_3u_{a2} + T_4u_{d2} + e$$

where y is the phenotypic value, b is a vector of fixed effects, and u_{a1} , u_{d1} , u_{a2} and u_{d2} are vectors of random effect. L , T_1 , T_2 , T_3 and T_4 are incidence matrices for b , u_{a1} , u_{d1} , u_{a2} and u_{d2} , respectively. e represents a matrix of residual effects and $e \sim N(0, I\sigma_e^2)$, in which I is an identity matrix and σ_e^2 is the residual variance. The variable $u_{a1} \sim N(0, k_1\sigma_{ua1}^2)$, in which k_1 represents the variance covariance matrix constructed by the additive relationship matrix A based on SNPs within the candidate region (the 1Mb region surrounding the genome-wide association signals), and σ_{ua1}^2 is the genetic variance of the corresponding random effect; the variable $u_{d1} \sim N(0, k_2\sigma_{ud1}^2)$, in which k_2 represents the variance covariance matrix constructed by the dominance relationship matrix D based on SNPs within the candidate region, and σ_{ud1}^2 is the genetic variance of the corresponding random effect; the variable $u_{a2} \sim N(0, k_3\sigma_{ua2}^2)$, in which k_3 represents the variance covariance matrix constructed by the additive relationship matrix A based on all SNPs across the whole genome except for the SNP set in target region, and σ_{ua2}^2 is the genetic variance of the corresponding random effect; the variable $u_{d2} \sim N(0, k_4\sigma_{ud2}^2)$, in which k_4 represents the variance covariance matrix constructed by the dominance relationship matrix D based on all SNPs across the whole genome except for the SNP set in target region, and σ_{ud2}^2 is the genetic variance of the corresponding random effect. The proportion of genetic variance explained by the target region was estimated by formula $\frac{\sigma_{ua1}^2 + \sigma_{ud1}^2}{\sigma_{ua1}^2 + \sigma_{ud1}^2 + \sigma_{ua2}^2 + \sigma_{ud2}^2}$, and the proportion of PVE by the target

region was calculated by formula $\frac{\sigma_{ua1}^2 + \sigma_{ud1}^2}{\sigma_{ua1}^2 + \sigma_{ud1}^2 + \sigma_{ua2}^2 + \sigma_{ud2}^2 + \sigma_e^2}$. The equations for the matrices A and D are the same as what shown in the “Construction of a decision-making model for hybrid breeding” section.

In addition, in the revised manuscript, the original PVE value has been substituted with the recently computed value using the updated equation (Lines 225, 240, 244 and 248; Fig. 3g and Extended Data Fig. 14 in revised manuscript).

(Fig. 3g) Please provide caption for x-axis on Fig. 3g.

Answer: Thanks for your reminder. The caption for x-axis on Fig. 3g was provided in the revised manuscript. Furthermore, we adjusted Fig. 3g: in addition to the *d/a* value, we also demonstrated the PVE value (%) of the grain quality related loci by the size of dots.

Reference

1. Wang, C. et al. Dissecting a heterotic gene through GradedPool-Seq mapping informs a rice-improvement strategy. *Nat Commun* **10**, 2982 (2019).
2. Liu, Q. et al. G-protein betagamma subunits determine grain size through interaction with MADS-domain transcription factors in rice. *Nat Commun* **9**, 852 (2018).
3. Yu, J. et al. Alternative splicing of OsLG3b controls grain length and yield in japonica rice. *Plant Biotechnol J* **16**, 1667-1678 (2018).
4. Fujita, D. et al. NAL1 allele from a rice landrace greatly increases yield in modern indica cultivars. *Proc Natl Acad Sci U S A* **110**, 20431-20436 (2013).
5. Zhang, G.H. et al. LSCHL4 from Japonica Cultivar, which is allelic to NAL1, increases yield of indica super rice 93-11. *Mol Plant* **7**, 1350-1364 (2014).
6. Sun, L. et al. GS6, a member of the GRAS gene family, negatively regulates grain size in rice. *J Integr Plant Biol* **55**, 938-949 (2013).
7. Tian, Z. et al. Allelic diversities in rice starch biosynthesis lead to a diverse array of rice eating and cooking qualities. *Proc Natl Acad Sci U S A* **106**, 21760-21765 (2009).

8. Wang, Z. et al. The amylose content in rice endosperm is related to the post - transcriptional regulation of the waxy gene. *Plant J* **7**, 613-622 (1995).
9. Zhang, C. et al. Wx(lv), the Ancestral Allele of Rice Waxy Gene. *Mol Plant* **12**, 1157-1166 (2019).
10. Huang, X. et al. High-throughput genotyping by whole-genome resequencing. *Genome Res* **19**, 1068-1076 (2009).
11. Huang, X. et al. Genomic architecture of heterosis for yield traits in rice. *Nature* **537**, 629-633 (2016).
12. Varona, L., Legarra, A., Toro, M.A. & Vitezica, Z.G. Non-additive Effects in Genomic Selection. *Front Genet* **9**, 78 (2018).
13. Bradbury, P.J. et al. TASSEL: software for association mapping of complex traits in diverse samples. *Bioinformatics* **23**, 2633-5 (2007).

Decision Letter, second revision:

14th Jun 2023

Dear Bin,

Thank you for submitting your revised manuscript "Structure and function of rice hybrid genomes reveal genetic basis and optimal performance of heterosis" (NG-A61585R1). It has now been seen by the original referee #3 and their comments are below. The reviewer finds that the paper has improved in revision, and therefore we'll be happy in principle to publish it in *Nature Genetics*, pending minor revisions to satisfy the referees' final requests and to comply with our editorial and formatting guidelines.

Thank you again for your interest in *Nature Genetics* Please do not hesitate to contact me if you have any questions.

Sincerely,

Michael Fletcher, PhD
Senior Editor, Nature Genetics

ORCID: 0000-0003-1589-7087

Reviewer #3 (Remarks to the Author):

I appreciate the authors for their revisions, which greatly improved the quality of the manuscript. I believe this manuscript has a great impact on hybrid breeding research. I have no further comment except for the following minor one.

Line 58: SV -> structural variation (SV) ?

Author Rebuttal, second revision:

Response to referees:

Response to Reviewer #3:

Remarks to the Author:

I appreciate the authors for their revisions, which greatly improved the quality of the manuscript. I believe this manuscript has a great impact on hybrid breeding research. I have no further comment except for the following minor one.

Answer: Thanks for your comments.

Line 58: SV -> 65structural variation (SV) ?

Answer: Yes, it is. The abbreviation ‘SV’ stands for structural variation, and we have provided its full name when it first appeared in the manuscript (Line 78).

Final Decision Letter:

2nd Aug 2023

Dear Bin,

I am delighted to say that your manuscript "Structure and function of rice hybrid genomes reveal genetic basis and optimal performance of heterosis" has been accepted for publication in an upcoming issue of Nature Genetics.

Your paper will be published online after we receive your corrections and will appear in print in the next available issue. You can find out your date of online publication by contacting the Nature Press Office (press@nature.com) after sending your e-proof corrections. Now is the time to inform your Public Relations or Press Office about your paper, as they might be interested in promoting its publication. This will allow them time to prepare an accurate and satisfactory press release. Include your manuscript tracking number (NG-A61585R2) and the name of the journal, which they will need when they contact our Press Office.

Please note that *Nature Genetics* is a Transformative Journal (TJ). Authors may publish their research with us through the traditional subscription access route or make their paper immediately open access through payment of an article-processing charge (APC). Authors will not be required to make a final decision about access to their article until it has been accepted. [Find out more about Transformative Journals](https://www.springernature.com/gp/open-research/transformative-journals)

Authors may need to take specific actions to achieve [a](https://www.springernature.com/gp/open-research/funding/policy-compliance-)

faqs"> compliance with funder and institutional open access mandates. If your research is supported by a funder that requires immediate open access (e.g. according to Plan S principles) then you should select the gold OA route, and we will direct you to the compliant route where possible. For authors selecting the subscription publication route, the journal's standard licensing terms will need to be accepted, including https://www.nature.com/nature-portfolio/editorial-policies/self-archiving-and-license-to-publish. Those licensing terms will supersede any other terms that the author or any third party may assert apply to any version of the manuscript.

Please note that Nature Portfolio offers an immediate open access option only for papers that were first submitted after 1 January, 2021.

An online order form for reprints of your paper is available at https://www.nature.com/reprints/author-reprints.html. Please let your coauthors and your institutions' public affairs office know that they are also welcome to order reprints by this method.

If you have not already done so, we invite you to upload the step-by-step protocols used in this manuscript to the Protocols Exchange, part of our on-line web resource, natureprotocols.com. If you complete the upload by the time you receive your manuscript proofs, we can insert links in your article that lead directly to the protocol details. Your protocol will be made freely available upon publication of your paper. By participating in natureprotocols.com, you are enabling researchers to more readily reproduce or adapt the methodology you use. [Natureprotocols.com](https://natureprotocols.com) is fully searchable, providing your protocols and paper with increased utility and visibility. Please submit your protocol to <https://protocolexchange.researchsquare.com/>. After entering your [nature.com](https://www.nature.com) username and password you will need to enter your manuscript number (NG-A61585R2). Further information can be found at <https://www.nature.com/nature-portfolio/editorial-policies/reporting-standards#protocols>

Sincerely,

Michael Fletcher, PhD
Senior Editor, Nature Genetics

ORCID: 0000-0003-1589-7087